# Safe Multi-task Pretraining with Constraint Prioritized Decision Transformer

## Abstract

Learning a safe policy from offline data without interacting with the environment is crucial for deploying reinforcement learning (RL) policies. Recent approaches leverage transformers to address tasks under various goals, demonstrating a strong generalizability for broad applications. However, these methods either completely overlook safety concerns during policy deployment or simplify safe RL as a dual-objective problem, disregarding the differing priorities between costs and rewards, as well as the additional challenge of multi-task identification caused by cost sparsity. To address these issues, we propose **S**afe **M**ulti-t**a**sk Pretraining with **Co**nstraint Prioritized Decision **T**ransformer (SMACOT), which utilizes the Decision Transformer (DT) to accommodate varying safety threshold objectives during policy deployment while ensuring scalability. It introduces a Constraint Prioritized Return-To-Go (CPRTG) token to emphasize cost priorities in the Transformer's inference process, effectively balancing reward maximization with safety constraints. Additionally, a Constraint Prioritized Prompt Encoder is designed to leverage the sparsity of cost information for task identification. Extensive experiments on the public OSRL dataset demonstrate that SMACOT achieves exceptional safety performance in both single-task and multi-task scenarios, satisfying different safety constraints in over 2x as many environments compared with strong baselines, showcasing its superior safety capability.

## 1 Introduction

Deep reinforcement learning (RL), a machine learning method that optimizes decision-making by maximizing cumulative rewards, has gained significant attention (Wang et al., 2022) and demonstrated remarkable potential in applications like autonomous driving (Zhao et al., 2024), industrial robotics (Haarnoja et al., 2024), healthcare (Yu et al., 2021), and the value alignment or reasoning of large language models (Ouyang et al., 2022; Shinn et al., 2023). However, real-world applications often require policies to adhere to additional safety constraints, such as speed and lane restrictions in autonomous driving to prevent accidents (Krasowski et al., 2020; Wang, 2022), as well as fuel limitations to ensure the vehicle remains operational (Lin et al., 2023). Furthermore, the trial-and-error nature of online RL becomes impractical when addressing safety concerns (Xu et al., 2022b). Consequently, offline safe RL (Le et al., 2019), which learns safe policies using pre-collected offline data without interacting with the real environment, has emerged as a major research focus.

Previous works in offline safe RL typically model the problem as a Constrained Markov Decision Process (CMDP) (Altman, 2021) with fixed elements, and ensure policy safety by solving constrained optimization problems (Wachi et al., 2024). However, such modeling restricts the policy to handling only one safety threshold under the specific task, reducing flexibility in real-world applications where multiple tasks and varying safety thresholds are often required. For instance, different roadways may have distinct speed limits, and vehicles may need to adjust behavior based on fuel levels to maintain safety performance. Therefore, multi-task safe RL, aiming to solve the mentioned problems, should be considered a significant topic for RL deployment. Leveraging the strong expressive power and scalability of Transformers (Vaswani et al., 2017; Lin et al., 2022), several works have attempted to apply them to multi-task decision-making. For instance, MGDT (Lee et al., 2022) simply models all visual tasks in a consistent sequential format, leveraging the Decision Transformer (DT) (Chen et al., 2021a) framework for multi-task learning. Prompt-DT (Xu et al., 2022c) introduces additional expert trajectory segments as prompts, enhancing multi-task identification in

non-visual tasks. These approaches showcase the powerful ability of Transformer-based sequential decision models to quickly adapt their behavior based on different token inputs to achieve various goals. Consequently, recent works have also explored applying Transformers to offline safe RL for decision-making under diverse safety thresholds. These methods either employ Cost-To-Go (CTG) tokens (Zhang et al., 2023; Liu et al., 2023) or logic tokens (Guo et al., 2024) to integrate safety constraints into the Transformer inputs, allowing the policy to adjust its conservatism based on different safety-related tokens, thus facilitating varied decision-making across numerous safety thresholds.

Despite their progress in multi-task or multi-safety threshold decision-making, these methods have not effectively utilized Transformers for multi-task safe RL. Firstly, they overlook the differing priorities between satisfying safety constraints and maximizing rewards by treating safety and rewards equally in model inputs. As a result, during deployment, if safety constraints conflict with reward maximization, the policy may prioritize rewards, potentially ignoring safety limits and leading to unsafe decisions. Additionally, these methods fail to address the extra challenges of task identification posed by multi-task safe scenarios, where differences between tasks mostly come from cost variations. Due to the sparsity of cost signals, short trajectory segments may lack sufficient distinguishing information, leading to failures in task identification and subsequently poorer safety performance. In conclusion, applying Transformers to multi-task safe RL still presents two key challenges that need to be addressed:

- How to model the higher priority of cost compared to reward within the Transformer?
- How to design prompts to extract information from sparse costs for task identification?

To address the aforementioned challenges, we propose a novel algorithm **S**afe **M**ulti-t**a**sk Pretraining with **Co**nstraint Prioritized Decision **T**ransformer (SMACOT) built upon the DT framework. Firstly, to prioritize costs over rewards, SMACOT introduces a novel Constraint Prioritized Return-To-Go (CPRTG) token by explicitly modeling RTG conditioned on CTG. Next, to efficiently extract task-related information from sparse costs, SMACOT introduces a Constraint Prioritized Prompt Encoder, which segments samples in the trajectory into safe and unsafe patches based on cost information, and then encodes them separately. This enables effective task identification based on the varying safe (or unsafe) transition distributions for different tasks. SMACOT effectively resolves the conflict between reward maximization and safety constraints satisfaction, while efficiently overcoming the challenge of task identification caused by sparse characteristics of costs. Extensive experiments on the open-source OSRL (Liu et al., 2024b) dataset demonstrate that SMACOT achieves exceptional safety performance across various safety thresholds in both single-task and multi-task scenarios. Compared to the previous SOTA sequence modeling method, it meets safety constraints in more than 2x tasks, and is currently the only algorithm to surpass the Oracle baseline BC-Safe.

## 2 RELATED WORK

**Safe RL and Offline Safe RL**  Safe RL ensures the safe deployment of policies by requiring them to meet additional safety constraints besides maximizing rewards, which is often modeled as constrained optimization problems (Garcıa & Fernández, 2015; Wachi et al., 2024), and Lagrangian multiplier methods are used as foundational techniques to solve it (Wachi et al., 2024). Lagrangian multiplier-based algorithms typically learn a cost value function and a parameterized Lagrangian multiplier, adjusting the multiplier based on the policy's cumulative cost to enhance safety (Chow et al., 2018; Stooke et al., 2020; Tessler et al., 2019; Chen et al., 2021b). However, these algorithms fail to consider the unsafe interactions with real-world environments during training, making them impractical. To make safe RL conform to reality, some works recently explore to learn safe policies using only pre-collected data, avoiding unsafe exploration, and hasten the offline safe RL. These approaches evaluate the safety performance of policies in a conservative manner, treating out-of-distribution (OOD) samples as unsafe to reduce visits to these regions (Le et al., 2019; Xu et al., 2022a; Zheng et al., 2024; Yao et al., 2024), thus mitigating the negative impacts of extrapolation errors (Fujimoto et al., 2019).

**Policy Learning as Sequence Modeling**  Due to the impressive performance of Transformers in complex sequential tasks such as large language models (Zhao et al., 2023) and time series analysis (Nie et al., 2023), many works aim to leverage their expressive power in offline RL, which can also be modeled a sequential task. For instance, DT (Chen et al., 2021a) uses a GPT-like Transformer

architecture (Achiam et al., 2023) with historical sequences and RTG tokens to infer optimal actions, breaking traditional RL paradigms and circumventing extrapolation errors directly. Variants like ODT (Zheng et al., 2022) and QDT (Yamagata et al., 2023) enhance DT's performance with online fine-tuning and Q-Learning (Watkins & Dayan, 1992), respectively. MGDT (Lee et al., 2022) and Prompt-DT (Xu et al., 2022c) extend DT to multi-task scenarios by using visual inputs or adding additional expert trajectory segments as prompts. For the safe problem, SaFormer (Zhang et al., 2023) and CDT (Liu et al., 2023) introduce CTG tokens to apply DT first in safe RL. SDT (Guo et al., 2024) utilizes signal temporal logic tokens to incorporate more information about safety constraints into DT. However, they all fail to account for the differing priorities between RTG and safety-related tokens, resulting in an incomplete resolution of the conflict between reward maximization and safety constraints, which motivates our work. More related work will be discussed in App. B.

## 3 PRELIMINARIES

### 3.1 SAFE RL AND MULTI-TASK SAFE RL

Safe RL can be modeled as a Constrained Markov Decision Process (CMDP), which is defined as a tuple $\langle S, A, r, c, P, \gamma, b \rangle$, where $S$ and $A$ represent the state space and the action space, respectively, $r : S \times A \to [-R_{\max}, R_{\max}]$ and $c : S \times A \to \{0, 1\}$ denote the reward and cost functions, respectively. $P : S \times A \times S \to [0, 1]$ is the transition probability function, $\gamma \in (0, 1)$ is the discount factor, and $b$ represents the safety threshold. A policy $\pi : S \to \Delta(A)$ maps states to action distributions. Under a given policy $\pi$, the reward return can be expressed as $R(\pi) = \mathbb{E}_{\tau \sim P_\pi} [\sum_{t=0}^{\infty} \gamma^t r(s_t, a_t)]$, where $\tau = (s_0, a_0, s_1, a_1, \dots)$ denotes a trajectory and $\tau \sim P_\pi$ indicates that the distribution of trajectories induced by policy $\pi$ and the environment dynamics $P$. Similarly, the cost return is given by $C(\pi) = \mathbb{E}_{\tau \sim P_\pi} [\sum_{t=0}^{\infty} \gamma^t c(s_t, a_t)]$. Thus, the objective of solving a CMDP is to learn a policy that maximizes reward return while adhering to safety constraints, which can be represented as:

$$\max_\pi R(\pi), \quad s.t. \ C(\pi) \le b. \tag{1}$$

In multi-task safe RL, the policy $\pi$ needs to be trained across multiple tasks to develop the capability to handle them simultaneously. Each task is defined as a unique CMDP, and differences between tasks may arise from changes of any element in $(S, A, P, r, c, b)$. Specifically, different tasks with the same $(S, A, P)$ are referred to as same-domain, otherwise the cross-domain tasks. In this paper, both scenarios are considered. We refer to the "domain" in this context as the "environment", which is uniquely determined by a group of $(S, A, P)$, and "tasks" in an environment only differ in $(r, c, b)$. For simplicity, we assume that the environment ID is known during both training and deployment.

During training, the policy is provided with a set of environments $\{\mathcal{E}_i\}_{i=1}^N$ and tasks $\{\mathcal{T}_j\}_{j=1}^M$, where each task belongs to a specific environment, i.e. an injection that maps task ID to environment ID is known. During deployment, the policy is required to make decisions for a task $\mathcal{T}$ which belongs to environment $\mathcal{E}_i$. If $\mathcal{T} \in \{\mathcal{T}_j\}_{j=1}^M$, then the policy can only utilize one expert trajectory for task identification. Otherwise, it is provided with $L$ expert trajectories of $\mathcal{T}$ to achieve efficient transfer.

### 3.2 SAFE RL VIA DECISION TRANSFORMER

Decision Transformer (DT) is one of the most prominent methods that apply sequence modeling to decision-making. It uses a Transformer network framework, modeling RL's reward maximization problem as a sequence prediction task. When applied to safe RL, DT models the trajectory as the following sequence to support training and generation with Transformers:

$$\tau = (\hat{C}_1, \hat{R}_1, s_1, a_1, \hat{C}_2, \hat{R}_2, s_2, a_2, \dots, \hat{C}_T, \hat{R}_T, s_T, a_T), \tag{2}$$

where $\hat{R}_t = \sum_{i=t}^T r_i$ is the Return-To-Go (RTG) token at time step $t$, and $\hat{C}_t = \sum_{i=t}^T c_i$ is the Cost-To-Go (CTG) token. Let $\tau_{-K:t} = (\hat{C}_{t-K}, \hat{R}_{t-K}, s_{t-K}, a_{t-K}, \dots, \hat{C}_{t-1}, \hat{R}_{t-1}, s_{t-1}, a_{t-1})$, DT's policy can be expressed as $\pi_{\text{DT}}(\hat{a}_t | \tau_{-K:t}, \hat{C}_t, \hat{R}_t, s_t)$, inferring the current action based on the previous K-step trajectory, the current RTG, CTG and state. In the trajectory $\tau$, only actions $a_t$ are generated auto-regressively, while the other elements are externally provided. The policy is trained by minimizing the difference between the inferred actions and actual actions. During deployment, DT in safe RL requires an initial RTG token $\hat{R}_1$, CTG token $\hat{C}_1$, and state $s_1$ to generate actions, with the RTG and CTG updated using $\hat{R}_{t+1} = \hat{R}_t - r_t$ and $\hat{C}_{t+1} = \hat{C}_t - c_t$, respectively.

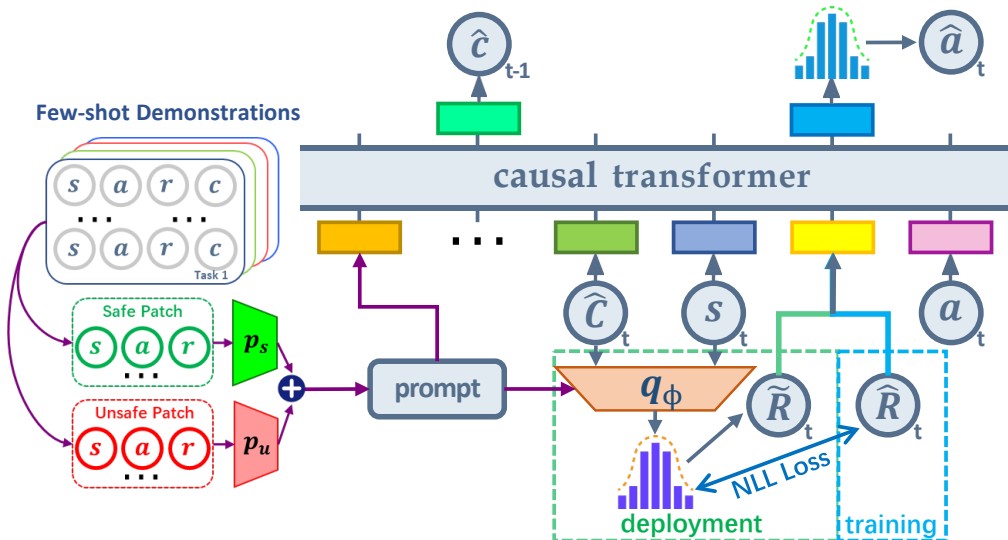

Figure 1: Structure of SMACOT. SMACOT introduces a CPRTG token $\tilde{R}_t$ during testing, which is generated based on the state and CTG token of the current step. For multi-task scenarios, SMACOT utilizes the Constraint Prioritized Prompt Encoder to generate a prompt for each task. This prompt will be used in both action inference and CPRTG generation to distinguish between tasks.

## 4 METHOD

This section gives a detailed description of our proposed SMACOT, a novel algorithm for offline safe and multi-task reinforcement learning (As visually depicted in Fig. 1). Sec. 4.1 illustrates the process of CPRTG token generation, Sec. 4.2 presents SMACOT's procedure for prompt encoding, while Sec. 4.3 introduces SMACOT's overall algorithm.

### 4.1 CPRTG TOKEN GENERATION

In safe RL, the policy optimization objective is represented by Eqn. 1. This constrained optimization problem implicitly prioritizes constraint satisfaction over reward maximization, as solutions that fail to meet constraints cannot be considered valid (Heath, 2018). In DT, these objectives are typically expressed through the RTG token $\hat{R}_t$ and the CTG token $\hat{C}_t$. To ensure safety performance, CTG should be given higher priority when the policy falls short of meeting both reward maximization and safety constraint objectives.

**Modeling RTG Conditioned on CTG**  To achieve the prioritization of CTG in SMACOT, a straightforward approach is to model RTG as conditioned on CTG, i.e., learning the model $p(\hat{R}_t|\hat{C}_t)$ from the offline dataset. Since the relationship between RTG and CTG is primarily derived from offline trajectories, where multiple RTG values might correspond to the same CTG, we further constrain the generation process by incorporating state information at each time step, and model $p$ as a non-deterministic normal distribution $\mathcal{N}$ approximated by a neural network $q_\phi$. Formally, given the offline dataset $\mathcal{D} = \{(s_t, a_t, s'_t, r_t, c_t, \hat{R}_t, \hat{C}_t, t)_k\}_{k=1}^{|\mathcal{D}|}$, we have:

$$q_\phi(\cdot|\hat{C}_t, s_t) = \mathcal{N}(\mu_\phi(\hat{C}_t, s_t), \Sigma_\phi(\hat{C}_t, s_t)), \tag{3}$$

where $\mu_\phi$ and $\Sigma_\phi$ are the mean and standard deviation networks, respectively, and $\hat{C}_t$ and $s_t$ represent the CTG and state at step $t$. To maximize the probability of generating $\hat{R}_t$ conditioned on the given $\hat{C}_t$ and $s_t$, the model $q_\phi$ is optimized by the following negative log-likelihood objective:

$$\min_{q_\phi} \mathbb{E}_{s_t, \hat{R}_t, \hat{C}_t \sim \mathcal{D}}[-\log q_\phi(\hat{R}_t|\hat{C}_t, s_t)]. \tag{4}$$

**CTG-based $\beta$-quantile Sampling for Safe And Expert Inference**  However, such modeling only prioritizes CTG without considering the need for expert-level inferences after ensuring safety.

Therefore, in addition to maximizing $p(\hat{R}_t|\hat{C}_t, s_t)$, we also aim to maximize $p(\hat{R}_t|\text{expert}_t, \hat{C}_t, s_t)$ by introducing a variable $\text{expert}_t$ that indicates the trajectory is expert after time step $t$. Similar to MGDT (Lee et al., 2022), we apply Bayes' theorem to obtain the following:

$$p(\hat{R}_t|\text{expert}_t, \hat{C}_t, s_t) \propto p(\hat{R}_t|\hat{C}_t, s_t)p(\text{expert}_t|\hat{R}_t, \hat{C}_t, s_t). \tag{5}$$

In Eqn. 5, $p(\text{expert}_t|\hat{R}_t, \hat{C}_t, s_t)$ represents the probability that the future trajectory is expert given the current RTG, CTG, and state. Intuitively, when fixing $\hat{C}_t$, a higher probability is attributed to $p(\text{expert}_t|\hat{R}_t, \hat{C}_t, s_t)$ if $\hat{R}_t$ possesses a larger value. Therefore, this term could be maximized by sampling large $\hat{R}_t$. In practice, we first sample $X$ possible values from the distribution $q_\phi(\cdot|\hat{C}_t, s_t)$, and then select the $\beta$-quantile from the candidates as the final input RTG token. Although larger $\beta$ brings about higher probability of $p(\text{expert}_t|\hat{R}_t, \hat{C}_t, s_t)$, it can lead to decrease of $p(\hat{R}_t|\hat{C}_t, s_t)$, necessitating an adjustment of $\beta$ to find a suitable balance. We propose the CTG-based $\beta$ decay technique:

$$\beta_t = \min(\beta_{\text{start}} + (\beta_{\text{start}} - \beta_{\text{end}})\frac{\hat{C}_t - \hat{C}_1}{\hat{C}_1}, \beta_{\text{end}}), \tag{6}$$

where $\hat{C}_1$ is the initially given safety threshold, $\beta_{\text{start}}$ and $\beta_{\text{end}}$ are two hyperparameters. When CTG is large—indicating more room for potential future safety violations—a larger $\beta_t$ for more aggressive decision-making is acceptable. Conversely, when CTG is small, the policy should be more conservative, resulting in a smaller $\beta_t$.

**CPRTG Token Generalization**    In conclusion, at time step $t$, we sample $X$ candidate values from $q_\phi(\cdot|\hat{C}_t, s_t)$, and chose the $\beta_t$-quantile value as the CPRTG token, denoted as $\tilde{R}_t$. This token provides a simple but efficient method for adjusting policy conservatism while attaining high-rewarding behaviors during deployment. If the policy does not meet safety requirements, lowering $\beta_{\text{start}}$ or $\beta_{\text{end}}$ can increase conservatism without altering model parameters. Similarly, adjustments can be made to improve reward return when the policy is too conservative. In practice, we typically fix $\beta_{\text{start}}$ as 0.99 and adjust $\beta_{\text{end}}$ only. More interpretations and theoretical results are provided in App. A.

### 4.2 CONSTRAINT PRIORITIZED PROMPT ENCODER LEARNING

The use of the CPRTG token successfully extends DT to scenarios with safety constraints. Our next goal is to expand SMACOT to multi-task settings for pretraining.

**Environment-specific Encoders**    First, considering the presence of cross-domain tasks, it is challenging to use a single unified neural network for all tasks due to the inconsistency in state action dimensions. Therefore, for each environment in the set $\{\mathcal{E}_i\}_{i=1}^N$, we apply environment-specific encoders to reduce the dimensions of the state and action spaces. Specifically, for environment $\mathcal{E}_i$, we introduce two encoders, $e_{s,i}$ for states and $e_{a,i}$ for actions, along with decoders $d_{s,i}$ and $d_{a,i}$. To ensure that the action encodings retain sufficient information from the original actions, $e_{a,i}$ and $d_{a,i}$ are trained using the reconstruction loss:

$$\min_{e_{a,i}, d_{a,i}} \mathbb{E}_{a_t \sim \mathcal{D}_i}[(d_{a,i}(e_{a,i}(a_t)) - a_t)^2], \tag{7}$$

where $\mathcal{D}_i$ represents the combined offline dataset for all tasks within environment $\mathcal{E}_i$ and $a_t$ is the sampled action. As for $e_{s,i}$ and $d_{s,i}$, we introduce an additional inverse dynamics model $g_i$, and train them by simultaneously minimizing the reconstruction error and the inverse dynamics error to incorporate both state information and dynamics transition information into state encodings:

$$\min_{e_{s,i}, d_{s,i}, g_i} \mathbb{E}_{s_t, a_t, s'_t \sim \mathcal{D}_i}[(d_{s,i}(e_{s,i}(s_t)) - s_t)^2 + (g_i(e_{s,i}(s_t), e_{s,i}(s'_t)) - e_{a,i}(a_t))^2], \tag{8}$$

where $s_t, a_t, s'_t$ are the sampled state-action transitions. The use of these encoders unifies the state and action spaces across all tasks, allowing us to simplify the problem to a same-domain task scenario for further discussion.

In same-domain tasks, differences typically emerge from $(r, c, b)$. Within SMACOT's DT framework, variations in $b$ are naturally handled through different initial CTG inputs, leaving us to focus on variations in $(r, c)$. Typical methods, such as Prompt-DT (Xu et al., 2022c), work well with dense rewards, but the sparse, binary nature of $c$ presents unique challenges under safe RL scenarios. Using limited $K$-step expert trajectory segments as prompts may fail to capture steps where tasks differ in safety constraints, leading to failure in task identification.

**Constraint Prioritized Prompt Encoder**  To address the challenge posed by the sparse, binary nature of $c$, we propose the Constraint Prioritized Prompt Encoder $p_e$. Specifically, $p_e = (p_s, p_u)$ consists of two prompt encoders. Given an expert trajectory $\tau^* = (s_1, a_1, r_1, c_1, \ldots, s_T, a_T, r_T, c_T)$ for task $\mathcal{T}$, where $r_t, c_t$ are the reward and cost of time step $t$, the prompt encoding $z$ is computed as follows:

$$z = p_e(\tau^*) = \frac{1}{T} \sum_{t=1}^{T} (\mathbf{1}_{\text{condition}}(c_t = 0) p_s(s_t, a_t, r_t) + \mathbf{1}_{\text{condition}}(c_t = 1) p_u(s_t, a_t, r_t)), \quad (9)$$

where $\mathbf{1}_{\text{condition}}$ is the indicator function. Since task differences may arise from variations in state spaces (environments), reward functions, and cost functions, it is crucial for $p_e$ to capture information from all three to ensure accurate task differentiation. To accomplish this, we introduce three additional decoder networks $f_s$, $f_r$, and $f_c$, and train them by minimizing prediction errors:

$$\min_{p_e, f_s, f_r, f_c} \mathbb{E}_{\mathcal{T} \sim \{\mathcal{T}_j\}_{j=1}^M} [\mathbb{E}_{\tau^*, s_t, a_t, s_t', r_t, c_t \sim \mathcal{D}_{\mathcal{T}}} [(f_s(s_t, a_t, p_e(\tau^*)) - s_t')^2$$
$$+ (f_r(s_t, a_t, p_e(\tau^*)) - r_t)^2 + (f_c(s_t, a_t, p_e(\tau^*)) - c_t)^2]], \quad (10)$$

where $\mathcal{D}_{\mathcal{T}}$ is the dataset for task $\mathcal{T}$, containing trajectories with both reward and cost information, $\tau^*$ refers to the trajectory that includes $s_t', r_t, c_t$, and $z = p_e(\tau^*)$ as defined in Eqn. 9.

The Constraint Prioritized Prompt Encoder removes the cost information from the prompt encoder's input and instead uses it to determine which encoder network to apply. This design ensures efficient use of cost information by distinguishing tasks based on the differences in the input distributions of state, action, and reward for different encoders. The resulting prompt encoding $z$ serves as both the first token for DT and input to the CPRTG generator $q_\phi$.

### 4.3 Overall Algorithm

With the design above, we can apply SMACOT to both single-task and multi-task scenarios to learn safe policies. Below, we briefly outline SMACOT's training and deployment in multi-task scenarios. Detailed pseudo-codes are provided in App. C and the approach for task identification in unknown environments are provided in App. D.

**Training**  During training, SMACOT first learns the environment-specific encoders by Eqn. 7 and Eqn. 8. After that, the Constrained Prioritized Prompt Encoder $p_e$ is learned by Eqn. 9. Then the CPRTG generator $q_\phi$ is similarly optimized by Eqn. 4. However, in multi-task scenarios, due to the existence of cross-domain tasks, environment-specific state action input heads and action output heads are used in both DT policy and $q_\phi$. Therefore, an additional environment ID is added to the input of $q_\phi$ and DT to select the appropriate head, as well as the prompt encoding $z$. With the learned $p_e$ and $q_\phi$, we can learn the DT policy. Let the policy network be denoted as $\pi_\theta$, which has two output heads: $\pi_{\theta,a}$ for actions and $\pi_{\theta,c}$ for costs. The additional cost head is utilized to aid the policy in extracting cost-related information to identify tasks better. Given the expert trajectory $\tau^*$, the learned $p_e$, environment ID $i$, and sampled trajectory $\tau_{-K:t}, \hat{R}_t, s_t$ from task $\mathcal{T}$'s offline dataset, the input can be represented as $o_t = (\tau_{-K:t}, \hat{C}_t, \hat{R}_t, s_t, p_e(\tau^*), i)$. The cost output head $\pi_{\theta,c}$ is modeled deterministically, while the action output head is modeled as a normal distribution:

$$\pi_{\theta,a}(\cdot|o_t) = \mathcal{N}(\mu_{\theta,a}(o_t), \Sigma_{\theta,a}(o_t)), \quad (11)$$

where $\mu_{\theta,a}$ and $\Sigma_{\theta,a}$ are the mean and standard deviation networks for the action output head, respectively. We optimize the policy by minimizing the negative log-likelihood loss and negative entropy loss of the actions, as well as the difference between the predicted costs and true costs:

$$\min_{\pi_{\theta,a}, \pi_{\theta,c}} \mathbb{E}_{\mathcal{T}, i \sim \{\mathcal{T}_j\}_{j=1}^M} [\mathbb{E}_{\tau^*, \tau_{-K:t}, \hat{C}_t, \hat{R}_t, s_t, a_t, c_t \sim \mathcal{D}_{\mathcal{T}}} [- \log \pi_{\theta,a}(a_t|o_t)$$
$$- \lambda_h H[\pi_{\theta,a}(\cdot|o_t)] + \lambda_c (\pi_{\theta,c}(o_t) - c_t)^2]], \quad (12)$$

where $i$ represents the environment ID to which task $\mathcal{T}$ belongs, $H$ is the Shannon entropy regularizer commonly used in RL (Haarnoja et al., 2018), $\lambda_h$ and $\lambda_c$ are two hyperparameters that control the weighting of the entropy regularization and the cost loss, respectively.

**Deployment**  During deployment, the initial task safety threshold $\hat{C}_1$ is provided, and in each time step, the CPRTG $\tilde{R}_t \sim q_\phi(\cdot|\hat{C}_t, s_t, p_e(\tau^*), i)$ is computed to replace the original RTG $\hat{R}_t$. At this point, the policy's input is $o_t = (\tilde{\tau}_{-K:t}, \hat{C}_t, \tilde{R}_t, s_t, p_e(\tau^*), i)$, where

$$\tilde{\tau}_{-K:t} = (\hat{C}_{t-k}, \tilde{R}_{t-k}, s_{t-k}, a_{t-k}, \ldots, \hat{C}_{t-1}, \tilde{R}_{t-1}, s_{t-1}, a_{t-1}). \quad (13)$$

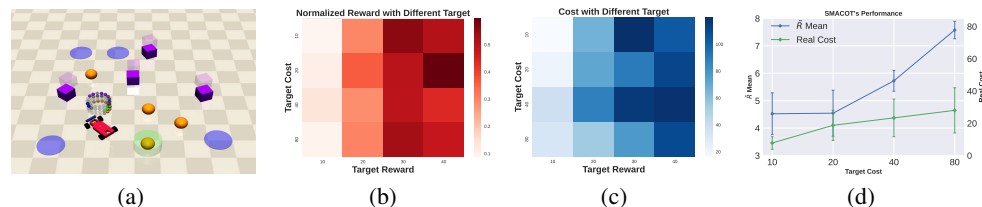

(a)  (b)  (c)  (d)

Figure 2: A case study in PointButton1 task. (a) The visualization of PointButton. (b) The normalized rewards of CDT with different initial RTGs and CTGs. (c) The real costs of CDT with different initial RTGs and CTGs. (d) The generated $\tilde{R}$ and cost performance of SMACOT with different target initial CTGs.

## 5 EXPERIMENTS

In this section, we present our experimental analysis conducted on 26 tasks from the OSRL (Liu et al., 2024b) dataset. The experiments aim to answer the following questions: (1) How does the traditional DT behave when reward maximization conflicts with safety satisfaction, and can SMACOT address this issue (Sec. 5.2)? (2) Can SMACOT outperform other baselines in safety performance across single-task and multi-task settings, and do its components contribute to this (Sec. 5.3)? (3) Will pre-training SMACOT helps improve learning efficiency in new tasks (Sec. 5.4)?

For a thorough evaluation, all results are averaged across twenty evaluation episodes, three random seeds, and four safety thresholds. For the baselines that do not use the DT framework, we train separate models for each safety threshold and report the average performance across these thresholds. For page limits, additional experimental information and results will be provided in App. G.

### 5.1 BASELINES AND TASKS

To provide a more comprehensive evaluation of SMACOT's performance, we conducted experiments with the following algorithms:

- **Ours**: (1) **SMACOT (ST)** is the single-task version of SMACOT with unified hyperparameters. (2) **SMACOT (MT)** is the multi-task version of SMACOT with unified hyperparameters. (3) **SMACOT (Oracle)** is the single-task version of SMACOT with task-specific $\beta_{\text{end}}$. It is not compared directly with the other baselines due to hyperparameter differences.

- **Single-task**: (4) **CPQ** (Xu et al., 2022a) applies CQL's (Kumar et al., 2020) conservative estimation to the cost critic and achieves state-of-the-art (SOTA) in traditional CMDP based offline safe RL algorithms. (5) **CDT** (Liu et al., 2023) adds CTG token to DT and reduces the conflict between CTG and RTG via data augmentation. It achieves SOTA in sequence modeling based offline safe RL algorithms. (6) **BC-Safe** modifies the dataset for behavior cloning only on safe trajectories, which is considered Oracle and also will not be compared to the other algorithms due to different learning data.

- **Multi-task**: (7) **MTCDT** extends CDT to multi-task settings with separate input or output heads for each environment without prompts. (8) **Prompt-CDT** (Xu et al., 2022c) builds on MTCDT with additional expert trajectory segments as prompts.

We first selected 21 tasks from the OSRL dataset, all part of the Safety-Gymnasium (Ji et al., 2023) benchmark, including 16 navigation tasks and 5 velocity tasks. The navigation tasks use 2 types of robots (Point and Car) across 4 scenarios: Button, Circle, Goal, and Push, with 2 tasks per scenario. The original 5 velocity tasks involve robots Ant, HalfCheetah, Hopper, Swimmer, and Walker2d, with one task for each robot. To enhance the dataset, we added one additional velocity task per robot with different velocity thresholds, bringing the total to 26 tasks across 13 environments. More details about baselines and tasks are provided in App. E.

## 5.2 CASE STUDY: WHEN RTG CONFLICTS WITH CTG

To explore the behavior of previous DT algorithms when RTG and CTG objectives conflict, we conducted experiments using the CDT algorithm on the PointButton1 task, just as shown in Fig. 2(a). We tested four different target rewards (initial RTG inputs) [10, 20, 30, 40] and four different target costs [10, 20, 40, 80] to compare policy behavior. The results, shown in Fig. 2(b) and Fig. 2(c), reveal that as the target reward increases, the achieved reward also rises, while the cost remains mostly unchanged regardless of the target cost. However, as the target reward increases, the cost also noticeably increases, indicating that the policy may prioritize RTG over CTG when they conflict. This reveals why previous DT methods fail to solve the conflict between rewards and costs.

Then, we aim to demonstrate that the CPRTG in SMACOT effectively resolves this issue. We conducted experiments on the same task using SMACOT with four different target costs, as shown in Fig. 2(d). The results show that the average CPRTG $\tilde{R}$ increases with the target cost but remains within a conservative range, while the real cost increases with the target cost but stays within safe limits, proving the effectiveness of SMACOT in resolving RTG-CTG conflicts.

## 5.3 COMPETITIVE RESULTS AND ABLATIONS

To validate the generality of SMACOT's outstanding safety performance, we conduct extensive experiments in the OSRL dataset. The experimental results of SMACOT and various baselines across 26 tasks are shown in Tab. 1. In the single-task setting, CPQ, which is based on CMDP, shows the poorest safety performance due to significant extrapolation errors and unstable training, failing to achieve high reward returns. In contrast, CDT, based on sequence modeling, performs better, with higher rewards and improved safety over CPQ, though still lacking satisfactory safety performance. SMACOT demonstrates a substantial improvement in safety, nearly doubling the average safety performance compared to CDT, without significantly sacrificing rewards, highlighting its effectiveness in balancing reward maximization with safety constraints.

Next, comparing the results in the multi-task setting, both MTCDT and Prompt-CDT show unsatisfactory safety performance. Part of this is due to CDT's inherent limitations in safety, while another factor is task identification failure. This issue is particularly evident in the Velocity tasks, where both MTCDT and Prompt-CDT exceed safety thresholds by tens of multiples in some tasks. This indicates that relying solely on the first $K$ time steps or short expert trajectory prompts fails to capture necessary cost-related information. In contrast, SMACOT demonstrates safety performance on par with the single-task setting, showcasing the effectiveness of the Constraint Prioritized Prompt Encoder in generating prompts for task identification.

Finally, comparing the results in the Oracle setting, BC-Safe effectively ensures policy safety by filtering offline data, meeting safety constraints in most environments. However, it cannot adapt to different safety thresholds with a single policy and requires retraining and data filtering for each new threshold. SMACOT, on the other hand, adapts to varying safety thresholds using a single policy model through different CTG token inputs and fine-tunes conservatism via the $\beta_{\text{end}}$ hyperparameter. As a result, SMACOT achieves safety in more environments and typically yields higher rewards than BC-Safe. This demonstrates that SMACOT effectively leverages both safe and unsafe training data, resulting in superior decision-making capability. The accompanying Fig. 3(a) shows the total number of tasks where each algorithm meets safety constraints, revealing that SMACOT consistently outperforms other baselines across all settings, achieving safe performance in more than twice as many tasks as previous approaches. Experiments about SMACOT under different safety thresholds and the visualization of prompt encodings are provided in App. G.

Next, we conducted ablation studies on the 26 tasks in the multi-task setting to investigate the impact of different modules on SMACOT's performance. The baselines used in the ablation studies include: (1) **W/o CP** means SMACOT without CPRTG. (2) **Det CP** uses a deterministic $q_\phi$ rather than a normal distribution. (3) **W/o CD** does not use the CTG-based $\beta$ decay. (4) **W/o PE** does not use the Constraint Prioritized Prompt Encoder for prompt generation. (5) **Simp PE** uses a simple MLP prompt encoder without separating safe and unsafe patches. (6) **Small DT** utilizes a DT backbone with fewer parameters.

The overall results are shown in Fig. 3(b). Comparing SMACOT with W/o CP and W/o CD shows that the CPRTG and CTG-based $\beta$ decay significantly enhances policy safety. Although using de-

Table 1: Final normalized reward and normalized cost return in all tasks. The rewards are normalized by the maximum and minimum reward return in the offline dataset of each task, and the costs are normalized by each safety threshold. The ↑ symbol denotes that the higher the reward, the better. The ↓ symbol denotes that the lower the cost (up to threshold 1), the better. Each value is averaged over **4 safety thresholds** $[10, 20, 40, 80]$, 20 evaluation episodes, and 3 random seeds. **Bold**: Safe agents. Gray: Unsafe agents. **Blue**: Safe agent with the highest reward in each setting.

| | Oracle | | | | Single-Task | | | | | | Multi-Task | | | | | |
| Task | BC-Safe | | SMACOT | | CPQ | | CDT | | SMACOT | | MTCDT | | Prompt-CDT | | SMACOT | |
| | r↑ | c↓ | r↑ | c↓ | r↑ | c↓ | r↑ | c↓ | r↑ | c↓ | r↑ | c↓ | r↑ | c↓ | r↑ | c↓ |
|---|---|---|---|---|---|---|---|---|---|---|---|---|---|---|---|---|
| PointButton1 | **0.04** | **0.74** | **0.09** | **0.91** | 0.67 | 5.28 | 0.54 | 5.16 | **0.05** | **0.66** | 0.49 | 4.17 | 0.55 | 4.90 | **0.04** | **0.55** |
| PointButton2 | 0.15 | 1.75 | **0.08** | **0.92** | 0.53 | 6.04 | 0.45 | 4.32 | 0.14 | 1.41 | 0.38 | 3.81 | 0.40 | 4.22 | **0.08** | **0.98** |
| PointCircle1 | **0.38** | **0.16** | **0.54** | **0.62** | **0.41** | **0.94** | **0.55** | **0.55** | 0.50 | 0.63 | **0.52** | **0.47** | **0.55** | **0.87** | 0.55 | 1.09 |
| PointCircle2 | 0.45 | 0.99 | **0.61** | **0.98** | 0.23 | 5.40 | 0.61 | 1.33 | **0.61** | **0.98** | 0.61 | 3.13 | 0.58 | 2.68 | 0.57 | 1.75 |
| PointGoal1 | **0.38** | **0.53** | **0.51** | **0.87** | **0.58** | **0.48** | 0.67 | 1.71 | **0.36** | **0.56** | 0.61 | 1.28 | 0.68 | 1.68 | **0.24** | **0.30** |
| PointGoal2 | 0.29 | 1.13 | **0.29** | **0.91** | 0.39 | 3.45 | 0.54 | 2.84 | 0.31 | 1.02 | 0.45 | 2.01 | 0.54 | 2.94 | **0.26** | **0.66** |
| PointPush1 | **0.13** | **0.67** | **0.19** | **0.88** | 0.23 | 1.60 | 0.27 | 1.42 | **0.19** | **0.88** | 0.23 | 1.11 | 0.24 | 1.25 | **0.12** | **0.69** |
| PointPush2 | 0.13 | 1.05 | **0.13** | **0.63** | 0.16 | 1.42 | 0.20 | 1.76 | 0.19 | 1.47 | 0.20 | 1.77 | 0.17 | 1.49 | **0.11** | **0.83** |
| CarButton1 | **0.07** | **0.87** | **0.07** | **0.74** | 0.48 | 15.40 | 0.20 | 3.97 | **0.07** | **0.74** | 0.23 | 4.61 | 0.29 | 6.38 | **0.04** | **0.89** |
| CarButton2 | -0.03 | 1.25 | **-0.02** | **0.89** | 0.29 | 19.32 | 0.14 | 4.70 | -0.02 | 1.33 | 0.22 | 5.19 | 0.25 | 5.46 | **-0.02** | **0.94** |
| CarCircle1 | 0.29 | 1.66 | 0.49 | 2.96 | -0.04 | 4.69 | 0.55 | 4.03 | 0.51 | 3.34 | 0.55 | 3.47 | 0.51 | 3.39 | 0.50 | 2.89 |
| CarCircle2 | 0.51 | 5.17 | **0.28** | **0.98** | 0.45 | 1.31 | 0.63 | 6.28 | **0.28** | **0.98** | 0.56 | 6.37 | 0.57 | 5.61 | 0.34 | 1.67 |
| CarGoal1 | **0.28** | **0.39** | **0.39** | **0.75** | 0.76 | 2.29 | 0.64 | 2.13 | **0.33** | **0.47** | 0.54 | 1.48 | 0.56 | 1.80 | **0.22** | **0.32** |
| CarGoal2 | **0.14** | **0.57** | **0.19** | **0.81** | 0.57 | 4.72 | 0.42 | 2.59 | **0.19** | **0.81** | 0.30 | 1.93 | 0.38 | 2.70 | **0.13** | **0.91** |
| CarPush1 | **0.15** | **0.45** | **0.28** | **0.96** | 0.03 | 1.07 | **0.29** | **0.98** | 0.20 | 0.67 | **0.25** | **0.84** | 0.25 | 0.93 | 0.18 | 0.48 |
| CarPush2 | **0.05** | **0.63** | **0.09** | **0.88** | 0.16 | 7.50 | 0.18 | 2.30 | **0.07** | **0.73** | 0.18 | 2.31 | 0.17 | 2.27 | **0.06** | **0.62** |
| SwimmerVelocityV0 | 0.52 | 0.08 | 0.62 | 0.98 | 0.09 | 0.99 | 0.71 | 1.32 | 0.63 | 1.29 | 0.72 | 7.48 | **0.72** | **0.75** | 0.69 | 0.84 |
| SwimmerVelocityV1 | 0.5 | 0.63 | 0.44 | 0.87 | 0.15 | 1.40 | 0.65 | 1.21 | **0.44** | **0.87** | 0.62 | 0.98 | **0.66** | **0.68** | 0.61 | 0.74 |
| HopperVelocityV0 | **0.50** | **0.25** | **0.18** | **0.52** | 0.04 | 2.01 | **0.84** | **0.92** | 0.84 | 1.50 | 0.68 | 13.37 | 0.89 | 5.01 | 0.57 | 4.28 |
| HopperVelocityV1 | **0.42** | **0.65** | **0.18** | **0.86** | 0.15 | 1.49 | 0.72 | 1.60 | 0.35 | 1.17 | 0.68 | 1.13 | 0.68 | 5.50 | 0.27 | 1.09 |
| HalfCheetahVelocityV0 | 0.92 | 1.11 | **0.67** | **0.38** | 0.40 | 2.24 | 0.94 | 1.05 | **0.51** | **0.36** | 0.89 | 14.71 | 1.08 | 35.24 | **0.70** | **0.36** |
| HalfCheetahVelocityV1 | **0.89** | **0.75** | 0.84 | 1.00 | 0.38 | 2.20 | **0.98** | **0.93** | 0.84 | 1.00 | 0.94 | 1.16 | **0.95** | **0.41** | 0.75 | 1.22 |
| Walker2dVelocityV0 | 0.24 | 1.45 | 0.32 | 2.90 | **0.04** | **0.46** | 0.29 | 1.91 | 0.32 | 2.90 | 1.25 | 24.51 | 0.81 | 14.30 | 0.35 | 4.44 |
| Walker2dVelocityV1 | **0.79** | **0.01** | 0.78 | 0.12 | **0.03** | **0.36** | **0.79** | **0.09** | 0.73 | 0.42 | **0.79** | **0.26** | 0.76 | 0.13 | 0.66 | 0.73 |
| AntVelocityV0 | 0.86 | 0.61 | **0.90** | **0.84** | -0.94 | 0.00 | 0.90 | 0.95 | **0.90** | **0.84** | 0.93 | 1.40 | 0.96 | 2.56 | 0.95 | 4.89 |
| AntVelocityV1 | **0.96** | **0.38** | 0.97 | 1.58 | -1.01 | 0.00 | **0.97** | **0.81** | 0.98 | 1.75 | 0.99 | 0.88 | 0.99 | 0.71 | 0.92 | 3.42 |
| Average | 0.39 | 0.92 | 0.39 | 0.99 | 0.19 | 3.54 | 0.56 | 2.19 | 0.4 | 1.11 | 0.57 | 4.2 | 0.58 | 4.38 | 0.38 | 1.45 |
| Average Ranking | 3.46 | | 2.15 | | 6.46 | | 4.96 | | 3.42 | | 5.58 | | 5.38 | | 4.00 | |

terministic $q_\phi$ improves safety performance, it limits $\beta_{\text{end}}$ adjustments, leading to reduced rewards and less flexibility in tuning policy conservatism for Det CP. Furthermore, the Constraint Prioritized Prompt Encoder enhances task identification, improving safety, as seen when comparing SMACOT with W/o PE and Simp PE. Lastly, the Small DT results demonstrate the importance of DT parameter size, underscoring the necessity of using Transformers for scalability. Therefore, we can conclude that all parts of SMACOT's design contribute positively to its strong safety performance. More detailed ablation results can be seen in App. G.

Additionally, we conducted ablation studies on how different values of $\beta_{\text{end}}$ will affect the safety performance, with the results shown in Fig. 3(c). It is evident that as $\beta_{\text{end}}$ increases, both the reward and cost of the policy also gradually rise. This confirms SMACOT's robust capability to quickly adjust conservatism by tuning $\beta_{\text{end}}$.

## 5.4 POLICY TRANSFER

In this section, we aim to explore the impact of SMACOT pretraining on task transfer. We designed 5 additional tasks within the Velocity task set, where each task uses 10 expert trajectories for transfer under a single safety threshold. We tested three methods: (1) **from_scratch**: training SMACOT from scratch on the new task. (2) **FFT**: full fine-tuning of the pretrained model on the new task; (3) **LoRA** (Hu et al., 2022): using LoRA to fine-tune the pretrained model. For a clearer evaluation of only the transfer ability, the CPRTG token was not used during testing.

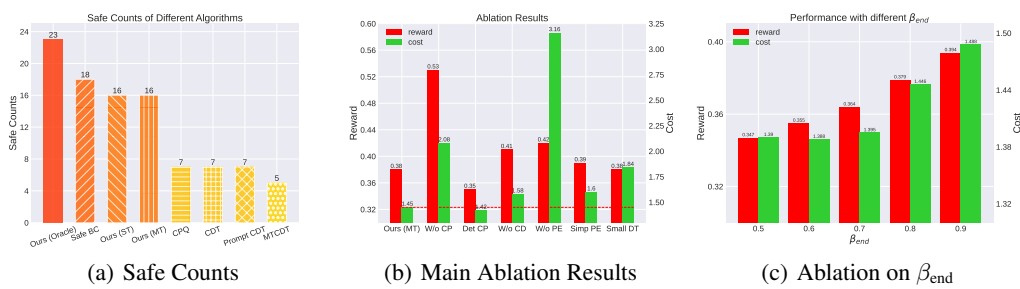

(a) Safe Counts      (b) Main Ablation Results      (c) Ablation on $\beta_{\text{end}}$

Figure 3: (a) The number of tasks each algorithm can make a safe decision. (b) The mean ablation results of different algorithms in all tasks. (c) The mean ablation results of different $\beta_{\text{end}}$.

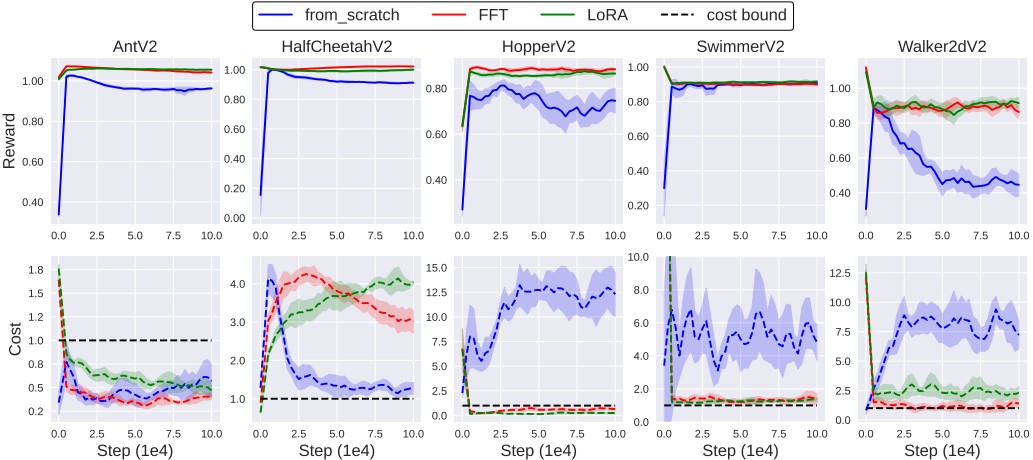

Figure 4: The transfer results of SMACOT in 5 new tasks.

The results are shown in Fig. 4. In four out of the five tasks, although the SMACOT pretrained model can't guarantee zero-shot safety, fine-tuning with FFT or LoRA shows clear advantages in both safety performance and reward compared to training from scratch. In one environment, the pretrained model ensures zero-shot safety, but fine-tuning results in a performance drop, likely due to insufficient trajectory coverage. Comparing FFT and LoRA, both perform similarly, possibly due to the relatively small model size, with LoRA offering lower fine-tuning overhead. Based on the above results, it can be found that the pretraining of SMACOT can effectively improve the policy's adaptation ability in new tasks, enhancing the algorithm's applicability. More transfer results and ablation studies on different LoRA ranks on policy's adaptation ability are provided in App. G.

## 6 FINAL REMARKS

In this work, we present a novel algorithm, SMACOT, designed to tackle the challenge of learning safe policies using the Transformer in both single-task and multi-task scenarios. SMACOT assigns higher priority to safety constraints through the use of CPRTG token, effectively addressing conflicts between RTG and CTG. Additionally, the design of Constraint Prioritized Prompt Encoder leverages the sparse and binary nature of cost for efficient information extraction and task identification. Extensive experiments on the OSRL dataset demonstrate SMACOT's strong performance in both safety and task identification, as well as its effective task transfer capabilities. Further studies on optimizing the Constraint Prioritized Prompt Encoder's learning, such as introducing contrastive loss, could be a promising way to improve task identification performance. In addition, achieving zero-cost exploration in safe RL using large language models (Wang et al., 2024) and deploying safe RL in embodied robots (Liu et al., 2024a) are also highly promising future research directions.

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

## A ADDITIONAL INTERPRETATIONS OF CPRTG FROM THE PERSPECTIVE OF OFFLINE RL

A core challenge in offline RL is mitigating the extrapolation errors caused by visiting OOD regions (Prudencio et al., 2023), which is partially solved by DT by restricting the policy to the offline dataset through supervised learning. However, when RTG is treated as part of the state, OOD RTG values can still introduce extrapolation errors. The generation of CPRTG can be seen as producing only RTG values seen in the offline dataset, reducing extrapolation errors and thus improving safety performance. From this perspective, we can make some theoretical analysis about the policy's performance bound.

**Lemma 1.** *(Janner et al., 2019) Suppose we have two distributions $p_1(x, y) = p_1(x)p_1(y|x)$ and $p_2(x, y) = p_2(x)p_2(y|x)$. We can bound the total variation distance (TVD) of the joint as*

$$D_{TV}(p_1(x,y)||p_2(x,y)) \leq D_{TV}(p_1(x)||p_2(x)) + \mathbb{E}_{x \sim p_1}[D_{TV}(p_1(y|x)||p_2(y|x))]. \quad (14)$$

**Lemma 2.** *(Janner et al., 2019) Suppose the expected TVD between two dynamics distributions is bounded as $\max_t \mathbb{E}_{s \sim p_1^t(s)}[D_{TV}(p_1(s'|s,a)||p_2(s'|s,a))] \leq \epsilon_m$, and $\max_S D_{TV}(\pi_1(a|s)||\pi_2(a|s)) \leq \epsilon_\pi$, where $p_1^t(s)$ is the state distribution of $\pi_1$ under dynamics $p_1(s'|s,a)$. Then the returns are bounded as:*

$$|\eta_1 - \eta_2| \leq \frac{2R_{max}\gamma(\epsilon_\pi + \epsilon_m)}{(1-\gamma)^2} + \frac{2R_{max}\epsilon_\pi}{1-\gamma}, \quad (15)$$

*where $\eta_i$ is the expected reward return under $\pi_i$ and $p_i$, $\gamma$ is the shared discount factor and $R_{max}$ is the maximum possible reward.*

**Theorem 1.** *Suppose the transition distribution of CTG given the next state during deployment is $p_1(\hat{C}_{t+1}|s', s, \hat{R}_t, \hat{C}_t, a)$, and that induced from the offline dataset is $p_2(\hat{C}_{t+1}|s', s, \hat{R}_t, \hat{C}_t, a)$. The transition distribution of RTG given the next state and next*

*CTG during deployment is $p_1(\hat{R}_{t+1}|\hat{C}_{t+1}, s', s, \hat{R}_t, \hat{C}_t, a)$, and that induced from the offline dataset is $p_2(\hat{R}_{t+1}|\hat{C}_{t+1}, s', s, \hat{R}_t, \hat{C}_t, a)$. Let the TVD between the CTG transition distribution $D_{TV}(p_1(\hat{C}_{t+1}|s', s, \hat{R}_t, \hat{C}_t, a)||p_2(\hat{C}_{t+1}|s', s, \hat{R}_t, \hat{C}_t, a))$ be TV(C, t), and the TVD between the RTG transition distribution $D_{TV}(p_1(\hat{R}_{t+1}|\hat{C}_{t+1}, s', s, \hat{R}_t, \hat{C}_t, a)||p_2(\hat{R}_{t+1}|\hat{C}_{t+1}, s', s, \hat{R}_t, \hat{C}_t, a))$ be TV(R, t). If*

$$\max_t \mathbb{E}_{s \sim p_1^t(s), s' \sim p_1(\cdot|s, \hat{R}_t, \hat{C}_t, a)}[TV(C, t)] \leq \epsilon_C, \tag{16}$$

$$\max_t \mathbb{E}_{s \sim p_1^t(s), s' \sim p_1(\cdot|s, \hat{R}_t, \hat{C}_t, a), \hat{C}_{t+1} \sim p_1(\cdot|s', s, \hat{R}_t, \hat{C}_t, a)}[TV(R, t)] \leq \epsilon_R, \tag{17}$$

*then we have*

$$\eta_1^R \geq \eta_2^R - \frac{2R_{max}\gamma(\epsilon_\pi + \epsilon_C + \epsilon_R)}{(1-\gamma)^2} - \frac{2R_{max}\epsilon_\pi}{1-\gamma}, \tag{18}$$

$$\eta_1^C \leq \eta_2^C + \frac{2\gamma(\epsilon_\pi + \epsilon_C + \epsilon_R)}{(1-\gamma)^2} + \frac{2\epsilon_\pi}{1-\gamma}, \tag{19}$$

*where $p_1^t(s)$ is the state distribution of the learned DT policy in timestep t, $p_1(\cdot|s, \hat{R}_t, \hat{C}_t, a)$ is the dynamics transition distribution of the target task, $\eta_1^R, \eta_1^C$ are the expected reward return and cost return for the learned DT policy during deployment, and $\eta_2^R, \eta_2^C$ is the expected reward return and cost return for the behavior policy under the state, CTG, RTG transition induced from the dataset.*

*Proof.* We view RTG $\hat{R}_t$ and CTG $\hat{C}_t$ from a different perspective, rather than the condition, but part of the state. Then, we take the state, RTG and CTG transition distribution induced from the offline dataset $p_2(s', \hat{R}_{t+1}, \hat{C}_{t+1}|s, \hat{R}_t, \hat{C}_t, a)$ as the ground truth transition distribution, but the state, RTG and CTG transition distribution during deployment as the environment model transition.

First, applying Bayes rule we have

$$p_i(s', \hat{R}_{t+1}, \hat{C}_{t+1}|s, \hat{R}_t, \hat{C}_t, a)$$

$$= p_i(s'|s, \hat{R}_t, \hat{C}_t, a)p_i(\hat{C}_{t+1}|s', s, \hat{R}_t, \hat{C}_t, a)p_i(\hat{R}_{t+1}|\hat{C}_{t+1}, s', s, \hat{R}_t, \hat{C}_t, a), \tag{20}$$

$i = 1, 2$, and $p_1(s'|s, \hat{R}_t, \hat{C}_t, a) = p_2(s'|s, \hat{R}_t, \hat{C}_t, a)$ due to the same state transition distribution. Therefore, apply Lemma 1 we can obtain

$$D_{TV}(p_1(s', \hat{R}_{t+1}, \hat{C}_{t+1}|s, \hat{R}_t, \hat{C}_t, a)||p_2(s', \hat{R}_{t+1}, \hat{C}_{t+1}|s, \hat{R}_t, \hat{C}_t, a))$$

$$\leq D_{TV}(p_1(s'|s, \hat{R}_t, \hat{C}_t, a_t)||p_2(s'|s, \hat{R}_t, \hat{C}_t, a))$$

$$+ \mathbb{E}_{s' \sim p_1(\cdot|s, \hat{R}_t, \hat{C}_t, a)}[D_{TV}(p_1(\hat{R}_{t+1}, \hat{C}_{t+1}|s', s, \hat{R}_t, \hat{C}_t, a)||p_2(\hat{R}_{t+1}, \hat{C}_{t+1}|s', s, \hat{R}_t, \hat{C}_t, a))]$$

$$\leq \mathbb{E}_{s' \sim p_1(\cdot|s, \hat{R}_t, \hat{C}_t, a)}[D_{TV}(p_1(\hat{C}_{t+1}|s', s, \hat{R}_t, \hat{C}_t, a)||p_2(\hat{C}_{t+1}|s', s, \hat{R}_t, \hat{C}_t, a))$$

$$+ \mathbb{E}_{\hat{C}_{t+1} \sim p_1(\cdot|s', s, \hat{R}_t, \hat{C}_t, a)}[D_{TV}(p_1(\hat{R}_{t+1}|\hat{C}_{t+1}, s', s, \hat{R}_t, \hat{C}_t, a)||p_2(\hat{R}_{t+1}|\hat{C}_{t+1}, s', s, \hat{R}_t, \hat{C}_t, a))]]. \tag{21}$$

Since

$$\max_t \mathbb{E}_{s \sim p_1^t(s), s' \sim p_1(\cdot|s, \hat{R}_t, \hat{C}_t, a)}[TV(C, t)] \leq \epsilon_C, \tag{22}$$

$$\max_t \mathbb{E}_{s \sim p_1^t(s), s' \sim p_1(\cdot|s, \hat{R}_t, \hat{C}_t, a), \hat{C}_{t+1} \sim p_1(\cdot|s', s, \hat{R}_t, \hat{C}_t, a)}[TV(R, t)] \leq \epsilon_R, \tag{23}$$

and thus

$$\max_t \mathbb{E}_{s \sim p_1^t(s)}[D_{TV}(p_1(s', \hat{R}_{t+1}, \hat{C}_{t+1}|s, \hat{R}_t, \hat{C}_t, a)||p_2(s', \hat{R}_{t+1}, \hat{C}_{t+1}|s, \hat{R}_t, \hat{C}_t, a))]$$

$$\leq \max_t \mathbb{E}_{s \sim p_1^t(s), s' \sim p_1(\cdot|s, \hat{R}_t, \hat{C}_t, a)}[TV(C, t)]$$

$$+ \max_t \mathbb{E}_{s \sim p_1^t(s), s' \sim p_1(\cdot|s, \hat{R}_t, \hat{C}_t, a), \hat{C}_{t+1} \sim p_1(\cdot|s', s, \hat{R}_t, \hat{C}_t, a)}[TV(R, t)] \tag{24}$$

$$\leq \epsilon_C + \epsilon_R.$$

Therefore, treat $p_i(s', \hat{R}_{t+1}, \hat{C}_{t+1}|s, \hat{R}_t, \hat{C}_t, a)$ as the state transition $p_i(s'|s, a)$ in Lemma 2, we further obtain

$$|\eta_1^R - \eta_2^R| \leq \frac{2R_{max}\gamma(\epsilon_\pi + \epsilon_C + \epsilon_R)}{(1-\gamma)^2} + \frac{2R_{max}\epsilon_\pi}{1-\gamma}, \tag{25}$$

$$|\eta_1^C - \eta_2^C| \leq \frac{2\gamma(\epsilon_\pi + \epsilon_C + \epsilon_R)}{(1-\gamma)^2} + \frac{2\epsilon_\pi}{1-\gamma}. \tag{26}$$

Finally, we have

$$\eta_1^R \geq \eta_2^R - \frac{2R_{\max}\gamma(\epsilon_\pi + \epsilon_C + \epsilon_R)}{(1-\gamma)^2} - \frac{2R_{\max}\epsilon_\pi}{1-\gamma}, \tag{27}$$

$$\eta_1^C \leq \eta_2^C + \frac{2\gamma(\epsilon_\pi + \epsilon_C + \epsilon_R)}{(1-\gamma)^2} + \frac{2\epsilon_\pi}{1-\gamma}. \tag{28}$$

$\square$

Theorem 1 provides an upper bound on the performance gap between the DT policy during deployment and the offline data-driven behavior policy. This gap is primarily influenced by three factors: $\epsilon_\pi$, $\epsilon_C$, and $\epsilon_R$. $\epsilon_\pi$ is mainly determined by the degree of optimization of the policy loss function, which is difficult to alter. As for $\epsilon_C$, we rely on the generalization ability of the Transformer for the CTG to adapt to different safety thresholds, and thus, we do not wish to modify the initial settings or update method of the CTG. Therefore, a natural approach to improving the lower bound of policy performance is to reduce the value of $\epsilon_R$. In this context, the CPRTG generator in SMACOT can be viewed as a neural network approximation of $p_2(\hat{R}_{t+1}|\hat{C}_{t+1}, s', s, \hat{R}_t, \hat{C}_t, a)$, which helps to lower $\epsilon_R$ during deployment.

Future research could also explore addressing OOD CTG values, for instance, by mapping large initial CTGs (those exceeding the maximum in the offline dataset) to the dataset's maximum, thus ensuring safety while further mitigating OOD-related extrapolation errors.

## B    MORE DETAILS ABOUT RELATED WORK

**Safe RL**    Safe RL is a kind of machine learning approach aimed at learning policies that maximize cumulative rewards while adhering to additional predefined safety constraints (Gu et al., 2022). Safe RL algorithms are broadly divided into two categories: safe exploration and safe optimization (García & Fernández, 2015). Safe exploration algorithms do not focus on directly optimizing the policy. Instead, they aim to modify the policy's behavior through additional mechanisms to prevent violations of safety constraints. A typical example of these algorithms is shielding-based methods (Alshiekh et al., 2018; Cheng et al., 2019; Xiao et al., 2023), which construct or learn logical structures known as "shields" or "barriers" that ensure the actions taken in a given state comply with the safety constraints. However, the decoupling from policy learning of safe exploration methods results in lower learning efficiency, leading to a growing focus on safe optimization algorithms. Safe optimization algorithms typically model the problem as a CMDP, with Lagrangian multiplier-based algorithms being the mainstream solution, as discussed in the main paper. Other than Lagrangian multiplier-based algorithms, trust region methods are among the most prevalent approaches in safe optimization. They attempt to keep policies within a safe trust region during updates via low-order Taylor expansions (Achiam et al., 2017; Yang et al., 2020) or variational inference (Liu et al., 2022). Due to their robust learning process, trust region methods are also further applied to multi-agent scenarios (Gu et al., 2023). However, their on-policy nature results in lower data efficiency. In response, recent works have increasingly focused on off-policy safe optimization. CAL (Wu et al., 2024) improves the optimization of Lagrange multipliers using the augmented Lagrangian method and enhances the conservatism of the cost function learned off-policy via the use of upper confidence bound. Meanwhile, SafeDreamer (Huang et al., 2024) increases data efficiency by learning an environment model and using model rollouts for data augmentation. Recently, more attention has been directed toward safety-conditioned RL. CCPO (Yao et al., 2023) effectively adapts to different safety thresholds in an online algorithm by incorporating the safety threshold into the input of CVPO. On the other hand, SDT (Guo et al., 2024) attempts to integrate safety prior knowledge expressed through temporal logic into the input of DT, further enhancing the policy's safety performance while adapting to various temporal logic safety constraints. This provides a fresh perspective for the practical application of safety RL. With the growing body of research on safe RL, these algorithms have found increasing applications in various fields. Notable examples include ensuring the safety of vehicles in autonomous driving (Kiran et al., 2021) and safeguarding robots in industrial settings (Brunke et al., 2022). In cutting-edge research, safe RL has also been applied to safe value alignment in large language models (Dai et al., 2024).

**Offline RL**    Offline RL trains policies using pre-collected datasets, avoiding real-world trial and error, which is critical for deploying RL in practical settings. Its primary challenge is addressing

the extrapolation errors (Prudencio et al., 2023). Methods like BCQ (Fujimoto et al., 2019) and CQL (Kumar et al., 2020) tackle this by constraining actions to those seen in the offline data or by penalizing unseen actions. Others, such as MOReL (Kidambi et al., 2020) and MOPO (Yu et al., 2020), learn the environment models from the offline data and utilize these models with uncertainty estimates to avoid OOD regions with low model accuracy. However, these methods only focus on reducing extrapolation errors, without addressing the challenge of generalizing in OOD areas. To tackle this limitation, MOREC (Luo et al., 2024) employs adversarial learning in the model learning process, improving model generalization abilities.

**Meta RL** Meta RL is similar to multi-task RL, with both involving multi-task training. However, Meta RL does not receive additional expert trajectories as prompts during testing. Instead, it must collect data in the unknown environment to generate prompts. Additionally, it focuses on training across large-scale similar tasks for generalization to new ones (Zhu et al., 2023). PEARL (Rakelly et al., 2019) uses a probabilistic encoder to facilitate task identification and employs Thompson sampling for data collection in new environments. Other works, like FOCAL (Li et al., 2021) and CORRO (Yuan & Lu, 2022), focus on designing contrastive loss functions for the encoder, improving task encoding robustness. COSTA (Guan et al., 2024) first considers safety in meta RL, designing a cost-based contrastive loss and a safety-aware data collection framework, improving policy safety in both task identification and deployment.

## C  ALGORITHMS

In this part, we will offer the detailed algorithms of SMACOT in multi-task scenarios. As described in the main paper, the workflow of SMACOT primarily includes four parts: $q_\phi$ training, $p_e$ training, policy training, and policy deployment. For $q_\phi$ and $p_e = (p_s, p_u)$, we both use simple multi-layer perceptron (MLP) networks, and additional prompt embeddings and environment IDs will be used as inputs for $q_\phi$ in multi-task scenarios:

$$\min_{q_\phi} \mathbb{E}_{\mathcal{T},i\sim\{\mathcal{T}_j\}_{j=1}^M}[\mathbb{E}_{\tau^*,s_t,\hat{R}_t,\hat{C}_t\sim\mathcal{D}_\mathcal{T}}[-\log q_\phi(\hat{R}_t|\hat{C}_t,s_t,p_e(\tau^*),i)]]. \tag{29}$$

The use of $p_s$ and $p_u$ in $p_e$ is similar to traditional context-based meta RL (Rakelly et al., 2019; Li et al., 2021; Yuan & Lu, 2022). Assume that the safe patch, classified using cost information, is represented as $\{(s_t, a_t, r_t)\}_{t=1}^{T_s}$. For each sample $(s_t, a_t, r_t)$ within this patch, we first concatenate it into a single vector $x_t$. Then, $x_t$ is passed through the MLP neural network $p_s$ to obtain an output vector $z_t$. Consequently, we obtain $T_s$ output vectors for the safe patch. Similarly, we can obtain $T_u$ output vectors for the unsafe patch. By averaging these $T_s + T_u$ output vectors, we obtain the final prompt embedding $z$. During the training of $p_e$, the prompt embedding $z$ is further input into MLP networks $f_s$, $f_r$, and $f_c$ to attempt to predict the corresponding $s'$, $r$, and $c$ values based on the given $(s, a)$ information ($f_s$, $f_r$, and $f_c$ are decoupled from the DT policy). The prediction is then used to compute a regression loss, which allows gradients to be backpropagated into $p_s$ and $p_u$. The relationship between $p_e$ and $f_s$, $f_r$, $f_c$ is essentially that of the encoder and decoder in a traditional autoencoder (Zhai et al., 2018). They are trained jointly before DT training, but only the frozen encoder (not updated with DT) is required for the DT policy training and deployment phase.

Detailed pseudo-codes for $q_\phi$ training, $p_e$ training, and policy training are provided in Alg. 1, while the pseudo-codes for policy deployment are provided in Alg. 2. Actually, from Alg. 1 we can learn that the training process of $q_\phi$ and $\pi_\theta$ are quite similar, which inspires us the potential of combining $q_\phi$ and $\pi_\theta$ together by using another head of DT as the CPRTG generator in future works.

## D  DISTINGUISH TASKS IN UNKNOWN ENVIRONMENTS

In this section, we will introduce the task identification method when the environment ID is unknown. Based on the definition in Sec. 3.1, an environment is determined by its state space, action space, and dynamics transition. Therefore, we need to infer the true environment ID based on this information. First, we filter out potential candidate environments from the previously seen environments based on the state space and action space dimensions of the unknown environment. Then, we sequentially use the environment-specific encoders, environment-specific decoders, and the inverse dynamics model from the candidate environments to test the given trajectory in the unknown environment. Specifically, given the trajectory $(s_t, a_t, s'_t)_{t=1}^T$, and the set $\{e_{s,i}, e_{a,i}, d_{s,i}, d_{a,i}, g_i\}_{i=1}^{N'}$ of

---

**Algorithm 1 SMACOT Training**

---

**Input:** task set $\{\mathcal{T}_j\}_{j=1}^M$, environment set $\{\mathcal{E}_i\}_{i=1}^N$, offline dataset for each task $\{\mathcal{D}_{\mathcal{T}_j}\}_{j=1}^M$, DT trajectory length $K$, hyperparameters $\lambda_h, \lambda_c$.

**Initialize:** Constraint Prioritized Prompt Encoder $p_e$, decoders $f_s, f_r, f_c$, state action encoders and decoders for each environment $\{e_{s,i}, e_{a,i}, d_{s,i}, d_{a,i}, g_i\}_{i=1}^N$, CPRTG generator $q_\phi$, DT policy $\pi_\theta$.

 1: **for** step in environment-specific training steps **do**
 2:     **for** each environment $\mathcal{E}_i$ **do**
 3:         Merge each task dataset in this environment to get the environment dataset $\mathcal{D}_i$.
 4:         Sample a batch $\{(s_t, a_t, s'_t)\}$.
 5:         Update $e_{a,i}$ and $d_{a,i}$ with Eqn. 7.
 6:         Update $e_{s,i}, d_{s,i}$ and $g_i$ with Eqn. 8.
 7:     **end for**
 8: **end for**
 9: **for** step in prompt encoder training steps **do**
10:     **for** each task $\mathcal{T}$ with its environment ID $i$ **do**
11:         Sample a batch $\{(\tau^*, s_t, a_t, s'_t, r_t, c_t)\}$ from $\mathcal{D}_{\mathcal{T}}$.
12:         Encode states sampled with $e_{s,i}$ and actions sampled with $e_{a,i}$.
13:         Update $p_e, f_s, f_r, f_c$ with Eqn. 10.
14:     **end for**
15: **end for**
16: **for** step in policy training steps **do**
17:     **for** each task $\mathcal{T}$ with its environment ID $i$ **do**
18:         Sample a batch $\{(\tau^*, \tau_{-K:t}, \hat{C}_t, \hat{R}_t, s_t, a_t, c_t)\}$ from $\mathcal{D}_T$.
19:         Update $\pi_\theta$ with Eqn. 12.
20:         Update $q_\phi$ with Eqn. 29.
21:     **end for**
22: **end for**
23: Return $\{e_{s,i}, e_{a,i}\}_{i=1}^N, p_e, q_\phi, \pi_\theta$.

---

**Algorithm 2 SMACOT Deployment**

---

**Input:** initial CTG $\hat{C}_1$, environment ID $i$, Constraint Prioritized Prompt Encoder $p_e$, state action encoders $e_{s,i}, e_{a,i}$, CPRTG generator $q_\phi$, DT policy $\pi_\theta$, expert trajectory $\tau^*$, DT trajectory length $K$, hyperparameters $X, \beta_{\text{start}}, \beta_{\text{end}}$.

**Initialize:** input sequence $\tau = []$.

 1: Encode states and actions in $\tau^*$ with $e_{s,i}$ and $e_{a,i}$.
 2: Compute the prompt encoding $z$ according to Eqn. 9.
 3: **for** t=1,...,T **do**
 4:     Observe current state $s_t$.
 5:     Compute $\beta_t$ according to Eqn. 6.
 6:     Sample $X$ values in distribution $q_\phi(\cdot|\hat{C}_t, s_t, z, i)$ and select the $\beta_t$-quantile of it as $\tilde{R}_t$.
 7:     Sample action $a_t$ from $\pi_{\theta,a}(\cdot|\tau[-K:], \hat{C}_t, \tilde{R}_t, s_t, z, i)$.
 8:     Step action $a_t$ in the task environment to get $r_t, c_t$.
 9:     Compute $\hat{C}_{t+1} = \hat{C}_t - c_t$.
10:     Append $\{\hat{C}_t, \tilde{R}_t, s_t, a_t\}$ to $\tau$.
11: **end for**

---

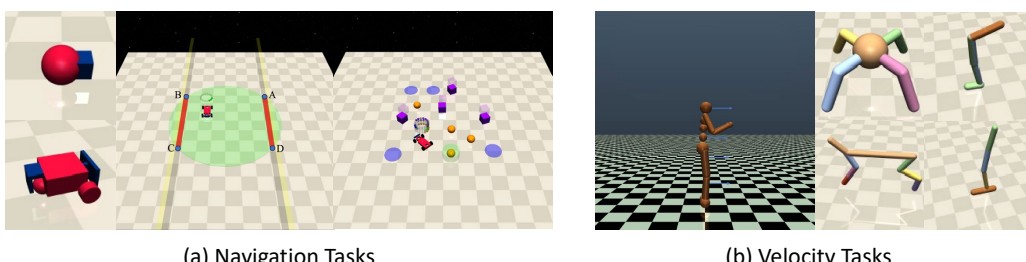

(a) Navigation Tasks                (b) Velocity Tasks

Figure 5: Tasks used in this paper. (a) Navigation Tasks based on Point and Car robots. (b) Velocity Tasks based on Ant, HalfCheetah, Hopper, Walker2d, and Swimmer robots.

all $N'$ candidate environments, the objective is as follows:

$$\min_i \frac{1}{T} \sum_{t=1}^{T} [(d_{a,i}(e_{a,i}(a_t)) - a_t)^2 + (d_{s,i}(e_{s,i}(s_t)) - s_t)^2 + (g_i(e_{s,i}(s_t), e_{s,i}(s'_t)) - e_{a,i}(a_t))^2], \quad (30)$$

where the first term is the action reconstruction loss, which aims to ensure consistency in the action space; the second term is the state reconstruction loss, which aims to ensure consistency in the state space; and the third term is the inverse dynamics error, which ensures consistency in the dynamics transition. Once the environment ID is determined, we revert to the previous setup, where the trajectory is passed through the environment-specific state encoder, environment-specific action encoder, and the Constraint Prioritized Prompt Encoder to obtain the prompt encoding, which serves as the basis for task identification.

## E  DETAILED DESCRIPTION OF THE TASKS AND BASELINES

### E.1  TASKS AND DATASETS

All pretraining tasks used in this paper are derived from the Safety-Gymnasium's Navigation Tasks and Velocity Tasks. In the Navigation Tasks, there are two different types of robots: Point and Car, which we need to control to navigate through the environment and earn rewards by reaching target points, pressing the correct buttons, or moving in designated directions. Different tasks also have varying costs, such as avoiding collisions with specific targets, preventing incorrect button presses, and staying within designated boundaries.

The Velocity Tasks are built on traditional MuJoCo (Todorov et al., 2012) simulations, requiring robots such as Ant, HalfCheetah, Swimmer, and Walker2d to move, where higher speeds result in higher rewards. However, each robot has specific safety velocity thresholds for different tasks, and exceeding these thresholds leads to unsafe states. For detailed descriptions of each task, refer to the original Safety-Gymnasium paper (Ji et al., 2023). Besides the mentioned tasks, we designed five new Velocity tasks for task transfer, which differ from previous ones only in their velocity thresholds, as detailed in Tab. 2.

The offline datasets used for each task are sourced from OSRL (Liu et al., 2024b). Specifically, the datasets for the VelocityV0 and VelocityV2 tasks were additionally collected using OSRL's original data collection methods.

### E.2  BASELINES

We provide a more detailed introduction to the baselines of the experiment in this section.

- **BC-Safe** is a widely-used Oracle baseline. When given a target safety threshold, it first filters the dataset to include only trajectories that satisfy this threshold, and then applies behavior cloning on these safe trajectories. Before, it was the only algorithm that achieved safe performance in OSRL Safety-Gymnasium tasks even if only evaluated by 3 target safety thresholds [20, 40, 80].

Table 2: Detailed velocity thresholds for new designed tasks.

| Tasks | Velocity Threshold |
|---|---|
| AntVelocityV2 | 2.52 |
| HalfCheetahVelocityV2 | 3.05 |
| HopperVelocityV2 | 0.56 |
| SwimmerVelocityV2 | 0.18 |
| Walker2dVelocityV2 | 2.00 |

- **CPQ** is a CMDP-based offline safe RL algorithm built upon the classic offline RL method CQL (Kumar et al., 2020). It incorporates the conservative regularization operator from CQL into the cost critic, treating out-of-distribution samples as unsafe. Unlike traditional methods that use Lagrangian multipliers for policy updates, CPQ directly truncates the reward critic to 0 for unsafe state-action pairs, preventing unsafe policy execution. This algorithm has become one of the most common baselines in offline safe algorithms and is considered state-of-the-art in CMDP-based offline safe algorithms.

- **CDT** is the SOTA algorithm under the traditional safe RL setting discussed in this paper. After incorporating CTG into DT, CDT also seeks to address the conflict between safety constraints and reward maximization. To tackle this issue, CDT proposes a data augmentation approach, where reward returns for certain safe but low-reward trajectories in the offline dataset are re-labeled with higher values. However, the effectiveness of this data augmentation method is still limited by the amount of augmented data and does not fundamentally resolve the underlying issue.

- **MTCDT** is a straightforward multi-task extension of CDT. It addresses the varying state and action dimensions in cross-domain tasks by utilizing distinct input and output heads for each environment. Additionally, MTCDT attempts to identify different tasks based solely on the sequential inputs of DT for multi-task decision-making.

- **Prompt-CDT** is also a multi-task extension of CDT. Building on MTCDT, Prompt-CDT utilizes additional expert trajectory segments as prompts to assist in task identification.

## F    HYPERPARAMETERS

The training and deployment of SMACOT both involve the selection of hyperparameters. To ensure reproducibility, this section outlines the specific hyperparameters used in our experiments. SMACOT is implemented based on CDT within the OSRL framework, and the default parameters from the framework are retained for any hyperparameters not explicitly mentioned in Tab. F.

For SMACOT (Oracle), the choices of $\beta_{\text{end}}$ in all tasks are as following:

$$
\begin{aligned}
[0.9, 0.6, 0.99, 0.8, 0.99, 0.7, 0.8, 0.5, 0.8, 0.6, 0.5, 0.8, 0.99, \\
0.8, 0.99, 0.9, 0.7, 0.8, 0.5, 0.5, 0.99, 0.8, 0.8, 0.99, 0.8, 0.99],
\end{aligned}
\tag{31}
$$

the order corresponds to the order of tasks in the table in the main paper. CDT also utilizes the values listed in the table for the shared parameters. For Small DT, the *DT embedding dim* is 128, the *DT num layers* is 2, and the *DT num heads* is 4.

## G    MORE EXPERIMENTAL RESULTS

### G.1    TIME COMPLEXITY ANALYSIS

Time complexity during policy training and deployment is indeed a critical issue in real-world applications. Therefore, in this section, we provide the analysis of SMACOT's time complexity. Due to the use of neural networks, it is not feasible to provide a quantitative analysis. Instead, we first compare SMACOT qualitatively with other baseline algorithms and present the specific results through actual data.

Table 3: Hyperparameter choices of SMACOT.

| | Hyperparameter | Value |
|---|---|---|
| environment-specific encoders | state encode dim | 32 |
| | action encode dim | 2 |
| | all network hidden layers | [128, 128, 128] |
| | update steps | 100000 |
| | batch size | 2048 |
| | learning rate | 0.0001 |
| prompt encoder $p_e$ | prompt encode dim | 16 |
| | all encoder hidden layers | [256, 256, 256] |
| | all decoder hidden layers | [128, 128] |
| | update steps | 100000 |
| | batch size | 2048 |
| | learning rate | 0.0001 |
| RTG generator $q_\phi$ | environment-specific state input head output dim | 64 |
| | environment-specific state input head hidden layers | [128, 128, 128] |
| | generator hidden layers | [256, 128, 128] |
| | update steps | 100000 |
| | batch size | 2048 |
| | learning rate | 0.0001 |
| policy learning and deployment | DT embedding dim | 512 |
| | DT num layers | 3 |
| | DT num heads | 8 |
| | DT sequence len | 20 |
| | environment-specific state input head output dim | 64 |
| | environment-specific state input head hidden layers | [128, 128, 128] |
| | environment-specific action input head output dim | 32 |
| | environment-specific action input head hidden layers | [128, 128, 128] |
| | environment-specific action output head input dim | 32 |
| | environment-specific action output head hidden layers | [128, 128, 128] |
| | update steps | 200000 |
| | batch size | 1024 |
| | learning rate | 0.0001 |
| | $\lambda_h$ | 0.1 |
| | $\lambda_c$ | 0.02 |
| | $\beta_{\text{start}}$ | 0.99 |
| | $\beta_{\text{end}}$ | 0.8 |
| | $X$ | 1000 |

Table 4: Time complexity comparison.

| | SMACOT (ST) | CDT | SMACOT (MT) | MTCDT | Prompt-CDT |
|---|---|---|---|---|---|
| Prompt Encoder Training | \ | \ | 1.330 h | \ | \ |
| DT Policy Training | 15.734 h | 15.737 h | 19.584 h | 19.288 h | 33.008 h |
| CPRTG Generator Training | 0.250 h | \ | 1.404 h | \ | \ |
| Deployment | 0.012 s/step | 0.008 s/step | 0.017 s/step | 0.008 s/step | 0.014 s/step |

First, during the training process in the single-task scenario, the policy training for SMACOT and CDT is identical, with the only difference being in the CPRTG generator's training. Since we use a traditional MLP neural network in the CPRTG, its computational complexity is much lower than that of the large Transformer networks used in policy training, resulting in minimal additional overhead for SMACOT during training. In the multi-task scenario, SMACOT, compared to MTCDT and Prompt-CDT, involves not only the CPRTG generator but also the training of the prompt encoder. This training includes both environment-specific encoder training and Constraint Prioritized Prompt Encoder training. Similar to CPRTG training, the prompt encoder training only involves MLP networks, so it does not introduce significant additional overhead. The specific data is shown in Tab. 4. In the single-task scenario, we use the training of a fixed task as the result, and in the

multi-task scenario, we report the average training time per task. During deployment, both single-task and multi-task scenarios use the same task for testing. All experiments were conducted on a single NVIDIA GeForce RTX 4090, and fairness was ensured even in the presence of CPU resource contention. The results in the table align with our earlier analysis, showing that the primary overhead during training comes from the policy training, with the additional MLP network training overhead being minimal. In contrast, Prompt-CDT incurs higher training overhead due to the use of sequence-based prompts.

In the policy deployment process, the additional time complexity for SMACOT mainly arises from the use of the CPRTG generator. By observing the last row of Tab. 4, we can see that in the single-task scenario, the use of the CPRTG generator does introduce some extra overhead. In the multi-task scenario, the additional overhead increases, as the use of prompts adds computational complexity both during policy inference and in the CPRTG generator, but the additional overhead is still within an acceptable range (smaller than 0.01 second) for real-world deployment. Further analysis of the additional time complexity introduced by the CPRTG generator is provided in Sec. G.7.

### G.2 ZERO-SHOT GENERALIZATION AND THE TRADE-OFF BETWEEN COSTS AND REWARDS

In this section, we further examine the zero-shot generation ability of SMACOT to new tasks with different constraints and different safety thresholds.

First, we conducted experiments to evaluate the performance of three different multi-task algorithms, including SMACOT, when directly deployed on new tasks with different constraints without fine-tuning. The results are shown in Tab. 5. It can be observed that none of the methods achieve satisfactory safety performance when facing new constraints without fine-tuning. We believe this phenomenon is expected, as in our pre-training tasks, only two tasks are similar to the target generalization task, making it difficult to acquire the necessary knowledge for generalization from just these two tasks. In contrast, traditional meta RL, which emphasizes generalization, may require pre-training on dozens of similar tasks to achieve even limited generalization capability (Rakelly et al., 2019; Li et al., 2021; Yuan & Lu, 2022). Additionally, we believe that, in safety-critical settings, fine-tuning is a more appropriate approach when encountering new constraints, as it better ensures safety performance.

Next, we tested the generalization capability of SMACOT and CDT under ten safety thresholds in four tasks. The results are shown in Fig. 6. It is shown that SMACOT is able to effectively adapt under any safety threshold, ensuring the safety performance of the policy. At the same time, both its cost and reward exhibit a clear increasing trend as the safety threshold rises. Although CDT shows some advantages in terms of reward, it clearly lags behind in terms of safety. In three of the environments, it fails to demonstrate adaptability to different safety thresholds, resulting in poor safety performance when the safety threshold is low. These experimental results validate the strong generalization capability of SMACOT across different safety thresholds.

Based on the results from Fig. 2, Fig. 3(c), and Fig. 6, we can perform a more comprehensive analysis of the trade-off between safety and performance. First, from the changes in cost and reward under different objectives for the CDT algorithm, as shown in Fig. 2, we observe that the trends in cost and reward are generally consistent. As the reward increases, the cost also rises, leading to a decrease in safety. We can attribute these identical changes in reward and cost to the same factor, namely, the conservatism of the policy. Therefore, the core objective in safe RL is to find an optimal level of conservatism for the policy. In traditional safe RL algorithms, once a policy is learned, its conservatism is fixed. However, Fig. 3(c) and Fig. 6 demonstrate two ways in which SMACOT can effectively adjust the conservatism of the policy without altering the parameters of the policy's neural network. Fig. 3(c) shows that by adjusting the $\beta_{\text{end}}$ hyperparameter, it is possible to change the policy's conservatism effectively while keeping the safety threshold fixed. A higher value of $\beta_{\text{end}}$ makes the policy more aggressive, leaning towards higher rewards while slightly compromising safety. Fig. 6 shows that as the safety threshold (i.e., the initial CTG input) increases, SMACOT also becomes more aggressive and achieves higher rewards. Thus, when the algorithm is applied in practice, SMACOT can optimize the level of conservatism through the following process. First, estimate the desired level of conservatism based on the expected safety threshold, then set the initial CTG and use default parameters for rollout. If the policy turns out to be too conservative and performance is below expectations, the first step is to decrease conservatism by increasing the $\beta_{\text{end}}$

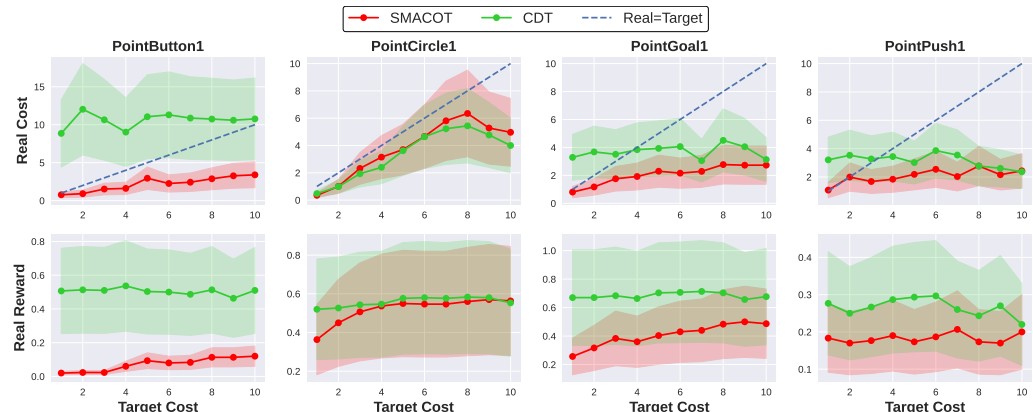

Figure 6: Performance of SMACOT and CDT under various safety thresholds. Target costs and real costs are normalized by 10.

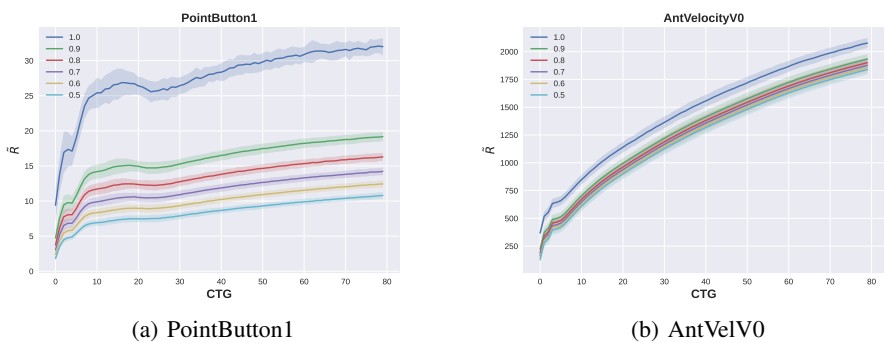

(a) PointButton1         (b) AntVelV0

Figure 7: Generated $\tilde{R}_t$ under different $\hat{C}_t$ and $\beta_t$.

hyperparameter. If the policy is still too conservative even after $\beta_{\text{end}}$ is increased to its maximum value, the initial CTG input can be adjusted, not strictly relying on the true target safety threshold. In this way, a better balance between safety and performance can be achieved in the trade-off.

Table 5: Zero-shot generation to tasks with different constraints.

| Task | SMACOT (MT) | | MTCDT | | Prompt-CDT | |
|------|-------------|------|-------|------|------------|------|
| | reward | cost | reward | cost | reward | cost |
| AntV2 | $0.99 \pm 0.01$ | $2.18 \pm 0.02$ | $1.11 \pm 0.00$ | $3.56 \pm 0.07$ | $1.10 \pm 0.01$ | $5.16 \pm 0.16$ |
| HalfCheetahV2 | $\mathbf{1.02 \pm 0.01}$ | $\mathbf{0.01 \pm 0.00}$ | $0.99 \pm 0.02$ | $4.02 \pm 0.32$ | $1.15 \pm 0.01$ | $7.74 \pm 0.63$ |
| HopperV2 | $0.52 \pm 0.03$ | $9.96 \pm 3.06$ | $0.70 \pm 0.00$ | $1.56 \pm 0.49$ | $0.70 \pm 0.10$ | $16.14 \pm 10.25$ |
| SwimmerV2 | $1.05 \pm 0.02$ | $54.98 \pm 0.67$ | $1.01 \pm 0.01$ | $31.09 \pm 0.43$ | $1.11 \pm 0.03$ | $54.21 \pm 10.85$ |
| Walker2dV2 | $1.21 \pm 0.01$ | $17.60 \pm 0.81$ | $0.82 \pm 0.05$ | $20.03 \pm 0.36$ | $0.63 \pm 0.43$ | $16.50 \pm 9.77$ |

## G.3 VISUALIZATION OF CPRTGS

In this section, we visualize how $\tilde{R}_t$ changes in response to variations in $\hat{C}_t$ and $\beta_t$, as shown in Fig. 7. The results indicate that $\tilde{R}_t$ increases significantly with higher values of $\hat{C}_t$ and also rises notably as $\beta_t$ increases. This confirms the rationale behind modeling RTG under CTG conditions and applying decay to $\beta_t$, allowing the policy to gradually adjust its conservatism based on potential future safety violations.

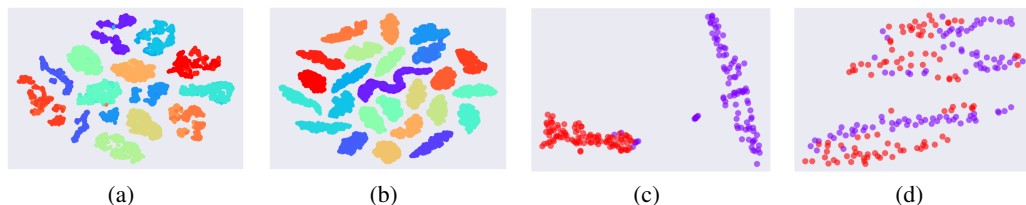

|        (a)        |        (b)        |        (c)        |        (d)        |

Figure 8: Different visualization results. (a) Visualization of state encodings after using the environment-specific state encoders. (b) Visualization of prompt encodings of the Constraint Prioritized Prompt Encoder using expert trajectories as prompts. (c) Visualization of prompt encodings of the Constraint Prioritized Prompt Encoder using trajectories collected by a same behavior policy for PointCircle1 and PointCircle2. (d) Visualization of prompt encodings of the simple MLP encoder using trajectories collected by a same behavior policy for PointCircle1 and PointCircle2.

## G.4 VISUALIZATION OF PROMPT ENCODINGS

In this section, we additionally explored the encoding process and properties of the Constraint Prioritized Prompt Encoder. First, we visualized the state distributions after encoding each environment's state separately using the environment-specific state encoder (Fig. 8(a)). The results reveal clear separations between different environments, though tasks within the same environment remain indistinguishable. Next, we visualized the results after applying the prompt encoder (Fig. 8(b)). At this stage, tasks within the same environment are also successfully differentiated, confirming the effectiveness of our design and loss function selection.

Next, we aim to further explore the properties of the Constraint Prioritized Prompt Encoder, specifically whether it focuses more on differences in state-action distribution driven by varying cost information, rather than on the state-action distribution itself. To test this, we specifically selected tasks where the cost function significantly affects the state-action distribution (e.g., PointCircle1 and PointCircle2). Using the policy of PointCircle1, we collected data in both tasks to ensure consistency in state-action distribution for the prompts. The results, as shown in Fig. 8(c) and Fig. 8(d), reveal that the Constraint Prioritized Prompt Encoder effectively distinguishes tasks based on cost information, while the traditional MLP encoder suffers from task confusion. This highlights the robust cost information extraction capability of the Constraint Prioritized Prompt Encoder.

## G.5 COMPARISON WITH TRAJECTORY TRANSFORMER

Table 6: Comparison with TT.

| Task | TT | | SMACOT (ST) | |
|------|------|------|------|------|
|      | reward | cost | reward | cost |
| PointButton1 | **0.05** | **0.86** | **0.05** | **0.66** |
| PointButton2 | 0.15 | 1.90 | 0.14 | 1.41 |
| PointGoal1 | **0.24** | **0.61** | **0.36** | **0.56** |
| PointGoal2 | 0.27 | 1.13 | 0.31 | 1.02 |

Trajectory Transformer (TT) (Janner et al., 2021) is a similar Transformer-based baseline to DT. Therefore, we further compare our method with TT in the Single-task setting, and the results are shown in Tab. 6. In the safe RL problem, we treat cost and reward in the same way, adding an additional step cost token as a prediction target of TT. From the results, we can observe that SMACOT consistently outperforms TT across all experimental tasks, further demonstrating its effectiveness in solving offline safe RL problems. However, on the other hand, it is evident that TT performs better than CDT in terms of safety (see results before). We believe that this occurs primarily because TT uses a training approach similar to BC, which does not incorporate RTG and CTG as inputs, but as selection criteria for beam search, thus avoiding the conflict between RTG and CTG.

Table 7: Comparison with FISOR in both Single-Task and Oracle settings.

| Task | Oracle | | | | Single-Task | | | |
| --- | --- | --- | --- | --- | --- | --- | --- | --- |
| | FISOR | | SMACOT | | FISOR | | SMACOT | |
| | r↑ | c↓ | r↑ | c↓ | r↑ | c↓ | r↑ | c↓ |
| PointButton1 | **-0.01** | **0.28** | **0.09** | **0.91** | 0.08 | 1.30 | **0.06** | **0.66** |
| PointButton2 | **0.05** | **0.43** | **0.08** | **0.92** | 0.11 | 1.41 | 0.14 | 1.41 |
| PointCircle1 | **0.05** | **0.06** | **0.54** | **0.62** | 0.44 | 5.54 | **0.5** | **0.63** |
| PointCircle2 | **0.20** | **0.00** | **0.61** | **0.98** | 0.71 | 6.21 | 0.61 | 0.98 |
| PointGoal1 | **0.03** | **0.01** | **0.51** | **0.87** | 0.66 | 2.14 | **0.36** | **0.56** |
| PointGoal2 | **0.05** | **0.08** | **0.29** | **0.91** | 0.29 | 1.28 | 0.31 | 1.02 |
| PointPush1 | **0.31** | **0.89** | 0.19 | 0.88 | **0.31** | **0.89** | 0.19 | 0.88 |
| PointPush2 | **0.09** | **0.29** | **0.13** | **0.63** | 0.24 | 1.40 | 0.19 | 1.47 |
| Average | 0.10 | 0.26 | 0.31 | 0.84 | 0.36 | 2.52 | 0.30 | 0.95 |

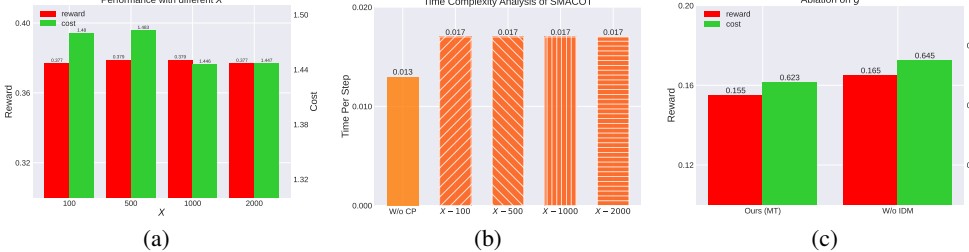

(a)  (b)  (c)

Figure 9: (a) Performance ablation on $X$. (b) Time complexity analysis of $X$. (c) Performance ablation on whether using the inverse dynamics model $g$ in 4 tasks.

## G.6  COMPARISON WITH FISOR

To further demonstrate the benefits of using CPRTG in SMACOT for improving policy safety performance, we additionally compared it with the latest SOTA offline safe RL method, FISOR (Zheng et al., 2024). FISOR uses a diffusion model (Yang et al., 2023) to identify the feasible region and solve the hard constraint problem. When the policy is in an unsafe region, FISOR guides the policy towards a safe region, and when the policy is in the safe region, it seeks the behavior that maximizes reward while keeping the policy within the safe region. We conducted the comparison in two different settings: Oracle and Single-Task. In the Single-Task setting, FISOR was trained using the hyperparameters proposed in the original paper for all tasks. In the Oracle setting, we adjusted FISOR's reverse expectile parameter $\tau$ for each task, $[0.8, 0.8, 0.7, 0.7, 0.8, 0.8, 0.9, 0.8]$ for the 8 test tasks specifically. According to the ablation results in FISOR, $\tau$ is positively correlated with the conservativeness of the policy. The results are shown in Table 7. First, in the Single-Task setting, SMACOT demonstrates significantly better safety performance than FISOR, highlighting the effectiveness of CPRTG in resolving the conflict between reward and safety. In the Oracle setting, while both methods are able to meet safety constraints for all tasks, SMACOT achieves significantly better reward performance, which underscores the higher flexibility of CPRTG in adjusting the conservativeness of the policy. Another clear advantage of SMACOT in the Oracle setting is that the hyperparameter $\beta_{\text{end}}$ is a test-phase-only parameter. This means that adjusting this parameter does not require retraining the policy, making it extremely convenient to fine-tune. In contrast, FISOR's hyperparameter $\tau$ is a training-phase parameter, so adjusting it requires retraining the policy. Overall, these results clearly demonstrate the effectiveness of CPRTG in handling the reward-safety trade-off, which is a core challenge in safe RL.

## G.7  ABLATION ON CPRTG SAMPLE NUMBER $X$

In this section, we first conducted an ablation study on different choices of the CPRTG sample number $X$ to investigate the impact of this hyperparameter on policy performance. The results are

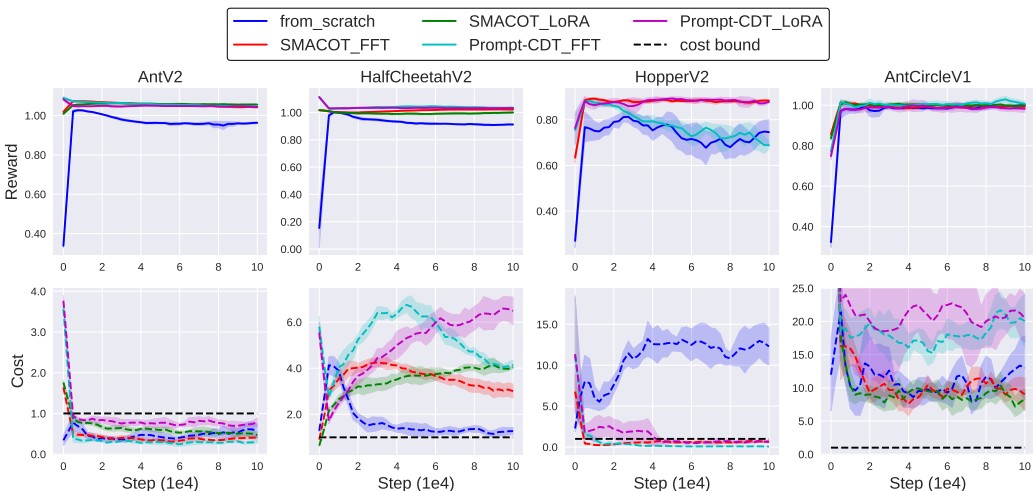

Figure 10: Transfer comparison with Prompt-CDT and transfer results in a dissimilar task.

shown in Fig. 9(a). It can be observed that as $X$ increases from 100 to 2000, there is almost no significant change in the policy's reward, but a noticeable reduction in the policy's cost. This result confirms that as the CPRTG sample number increases, the sampled values are closer to the desired quantile points, leading to better performance and a certain improvement in safety.

Additionally, we performed a further analysis of the time complexity of the CPRTG generation process in SMACOT, with the results presented in Fig. 9(b). From the figure, we can draw two conclusions. First, when CPRTG is used, the time consumption does indeed increase compared to not using CPRTG, indicating that the CPRTG generation process introduces additional computational overhead. Second, when $X$ increases from 100 to 2000, the time overhead remains almost unchanged, suggesting that the additional cost brought by CPRTG mainly comes from the inference of the CPRTG generator's neural network, rather than the sampling of quantile points. Therefore, in practice, we can increase $X$ as much as possible to achieve better policy performance without introducing significant additional computational cost.

## G.8 DISCUSSION AND ABLATION ON INVERSE DYNAMICS MODEL $g$

In Sec. 4.2, we introduced an additional inverse dynamics model $g$ to compute the inverse dynamics error for training the environment-specific state encoders. The primary motivation for using the inverse dynamics model is to address tasks with identical state and action spaces but different dynamics transitions. While such tasks have not appeared in our main experiments, they are still quite common (Nagabandi et al., 2019; Eysenbach et al., 2021; Zhang et al., 2024). In these cases, the inverse dynamics error based on the inverse dynamics model can effectively produce different state representations during the environment-specific state encoder learning phase, thereby reducing the learning difficulty for the Constraint Prioritized Prompt Encoder. Moreover, as described in Sec. D, when the environment ID is unknown, the inverse dynamics error based on the inverse dynamics model becomes the core method for distinguishing these tasks, making it an essential component. We also conducted an additional ablation study to ensure that the use of the inverse dynamics model does not negatively impact the policy performance, with results shown in Fig. 9(c), which aligns with our expectations.

## G.9 MORE TASK TRANSFER RESULTS

In this section, to demonstrate that our prompt encoder design can include more effective information than directly inputting sequence prompts during task transfer, we compare the results with Prompt-CDT under two fine-tuning methods: FFT and LoRA. Additionally, we introduce a new environment, AntCircle, which has a lower similarity to the pretraining task. In AntCircle, the state space, action space, and dynamics transition are consistent with AntVelocity, but both the reward

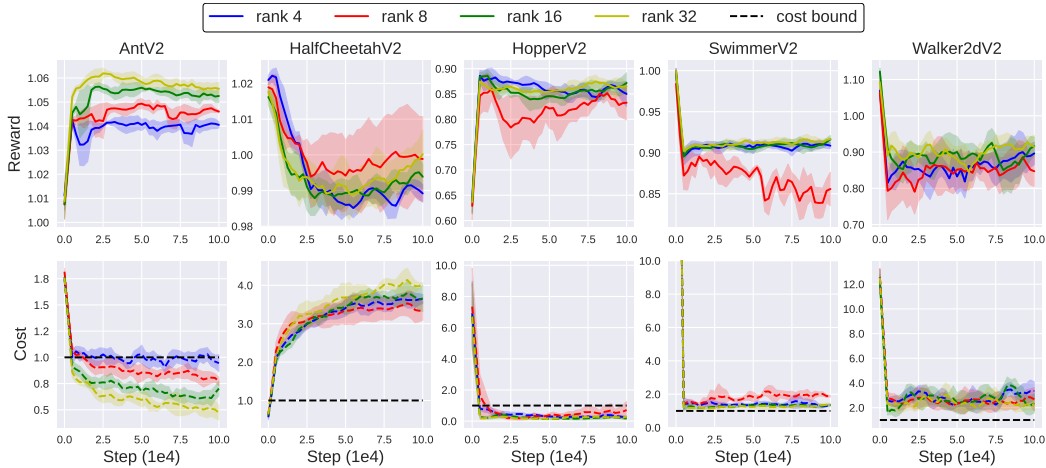

Figure 11: Ablation on different LoRA ranks.

function and cost function undergo significant changes. The reward function is modified to represent the speed at which the Ant robot moves along a circle, while the cost function now penalizes the robot's x-coordinate rather than its speed. The results are shown in Fig. 10.

First, by observing the results in similar tasks, we see that Prompt-CDT's multitask pretraining also provides a certain level of improvement in task transfer in environments other than HalfCheetah. However, compared to SMACOT, Prompt-CDT still exhibits inferior task transfer performance. In HalfCheetah, it leads to a significantly higher violation of safety constraints. In Hopper, FFT shows a noticeable drop in performance, while LoRA causes instability in safety. This clearly indicates that SMACOT's use of the prompt encoder provides more effective information for knowledge transfer than directly using sequence prompts.

Next, by observing the results in dissimilar tasks, we find that even in scenarios with low task similarity, SMACOT's pretraining still provides some performance improvement compared to learning from scratch. However, due to the limited amount of transferable knowledge, this improvement is less significant than in similar tasks. In contrast, Prompt-CDT's pretraining results in a noticeable decline in safety performance. This indicates that the sequential prompts used in Prompt-CDT do not always bring additional information gain and may sometimes interfere with the extraction of effective information. Furthermore, these results suggest that in low-similarity scenarios, few-shot adaptation may not always yield stable results, and using larger datasets for training might be a better alternative. Additionally, increasing the diversity of tasks during pretraining is an effective way to enhance the policy's transferability.

### G.10   ABLATION ON LORA RANK

In LoRA, performance is mainly influenced by the LoRA rank $r$ and the LoRA $\alpha$ (Hu et al., 2022). Following standard practice, we set $\alpha$ to twice the value of $r$ and conducted experiments on task transfer with various $r$ values. The results, shown in Fig. 11, indicate that performance generally improves as $r$ increases, except in the HalfCheetah task, where an anomaly occurred, consistent with previous findings. These results suggest that when the model has relatively few parameters, increasing the number of fine-tuned parameters positively impacts performance.

### G.11   DETAILED MAIN ABLATION RESULTS

In this section, we also provide the detailed results of the ablation studies, as shown in Tab. 8.

Table 8: Detailed results of the ablation studies done in the multi-task setting.

| Task | Ours (MT) | | W/o CP | | Det CP | | W/o CD | | W/o PE | | Simp PE | | Small DT | |
|---|---|---|---|---|---|---|---|---|---|---|---|---|---|---|
| | r↑ | c↓ | r↑ | c↓ | r↑ | c↓ | r↑ | c↓ | r↑ | c↓ | r↑ | c↓ | r↑ | c↓ |
| PointButton1 | **0.04** | **0.55** | 0.49 | 3.94 | **0.03** | **0.62** | 0.09 | 1.14 | **0.08** | **0.94** | 0.02 | 0.60 | 0.07 | 0.73 |
| PointButton2 | **0.08** | **0.98** | 0.32 | 3.31 | **0.03** | **0.95** | 0.11 | 1.41 | 0.07 | 1.01 | 0.10 | 1.21 | 0.11 | 1.19 |
| PointCircle1 | 0.55 | 1.09 | **0.57** | **0.93** | 0.52 | 1.05 | 0.55 | 1.05 | **0.48** | **0.70** | 0.53 | 0.86 | 0.53 | 0.71 |
| PointCircle2 | 0.57 | 1.75 | 0.60 | 1.92 | 0.55 | 1.85 | 0.57 | 1.85 | 0.57 | 2.59 | 0.55 | 1.51 | 0.53 | 1.46 |
| PointGoal1 | **0.24** | **0.30** | 0.62 | 1.44 | **0.20** | **0.35** | 0.33 | 0.53 | 0.22 | 0.32 | 0.26 | 0.36 | 0.28 | 0.48 |
| PointGoal2 | **0.26** | **0.66** | 0.49 | 2.33 | **0.23** | **0.63** | 0.30 | 0.89 | 0.22 | 0.79 | 0.22 | 0.77 | 0.27 | 0.99 |
| PointPush1 | **0.12** | **0.69** | 0.25 | 1.40 | **0.10** | **0.57** | 0.17 | 0.7 | 0.13 | 0.52 | 0.12 | 0.56 | 0.17 | 0.69 |
| PointPush2 | **0.11** | **0.83** | 0.15 | 1.25 | **0.10** | **0.78** | 0.10 | 0.94 | 0.11 | 0.77 | 0.11 | 0.80 | 0.14 | 1.20 |
| CarButton1 | **0.04** | **0.89** | 0.23 | 4.70 | **0.02** | **0.66** | 0.07 | 0.77 | 0.02 | 0.65 | 0.04 | 0.67 | 0.02 | 0.60 |
| CarButton2 | **-0.02** | **0.94** | 0.17 | 4.77 | **-0.02** | **0.77** | -0.05 | 1.56 | 0.01 | 1.07 | -0.01 | 1.05 | -0.04 | 1.14 |
| CarCircle1 | 0.50 | 2.89 | 0.52 | 4.10 | 0.49 | 2.87 | 0.50 | 3.17 | 0.42 | 2.44 | 0.51 | 3.27 | 0.56 | 4.53 |
| CarCircle2 | 0.34 | 1.67 | 0.55 | 4.61 | 0.31 | 1.40 | 0.35 | 2.05 | 0.46 | 4.06 | 0.32 | 1.37 | 0.38 | 2.41 |
| CarGoal1 | **0.22** | **0.32** | 0.50 | 1.32 | **0.19** | **0.29** | 0.24 | 0.37 | 0.19 | 0.28 | 0.21 | 0.34 | 0.26 | 0.45 |
| CarGoal2 | **0.13** | **0.91** | 0.32 | 1.83 | **0.15** | **0.94** | 0.20 | 1.01 | **0.15** | **0.76** | 0.14 | 0.94 | 0.19 | 1.06 |
| CarPush1 | **0.18** | **0.48** | 0.24 | 0.73 | **0.16** | **0.43** | 0.20 | 0.45 | 0.17 | 0.33 | 0.18 | 0.35 | 0.19 | 0.38 |
| CarPush2 | **0.06** | **0.62** | 0.16 | 2.25 | **0.04** | **0.46** | 0.07 | 1.15 | **0.04** | **0.54** | 0.05 | 0.61 | 0.06 | 0.81 |
| SwimmerVelocityV0 | **0.69** | **0.84** | 0.72 | 0.72 | 0.53 | 0.64 | 0.72 | 0.82 | 0.67 | 5.79 | 0.70 | 1.93 | 0.67 | 3.12 |
| SwimmerVelocityV1 | **0.61** | **0.74** | 0.66 | 0.65 | 0.43 | 1.38 | **0.66** | **0.69** | 0.56 | 0.51 | 0.59 | 0.74 | 0.56 | 0.76 |
| HopperVelocityV0 | 0.57 | 4.28 | 0.83 | 2.31 | 0.41 | 3.77 | 0.60 | 4.13 | 0.55 | 2.69 | 0.55 | 3.25 | 0.37 | 1.85 |
| HopperVelocityV1 | 0.27 | 1.09 | 0.64 | 3.53 | **0.23** | **0.96** | 0.37 | 1.60 | **0.44** | **0.52** | 0.28 | 1.43 | 0.16 | 2.08 |
| HalfCheetahVelocityV0 | **0.70** | **0.36** | 0.95 | 0.78 | 0.67 | 0.31 | 0.76 | 0.40 | 0.75 | 13.77 | **0.7** | **0.38** | 0.78 | 0.17 |
| HalfCheetahVelocityV1 | 0.75 | 1.22 | **0.95** | **0.27** | 0.73 | 1.06 | 0.80 | 1.13 | **0.73** | **0.65** | 0.88 | 0.49 | 0.67 | 0.69 |
| Walker2dVelocityV0 | 0.35 | 4.44 | 0.28 | 2.47 | 0.36 | 4.51 | 0.36 | 4.49 | 1.44 | 27.45 | 0.40 | 5.57 | 0.36 | 4.64 |
| Walker2dVelocityV1 | **0.66** | **0.73** | 0.74 | 0.09 | **0.66** | **0.72** | 0.67 | 0.72 | 0.69 | 0.71 | 0.72 | 0.59 | 0.58 | 2.28 |
| AntVelocityV0 | 0.95 | 4.89 | 0.94 | 1.65 | 0.96 | 5.01 | 0.96 | 4.99 | 0.85 | 7.02 | 0.98 | 8.45 | 0.99 | 10.13 |
| AntVelocityV1 | 0.92 | 3.42 | **0.98** | **0.71** | 0.91 | 3.80 | 0.95 | 3.13 | 0.92 | 5.24 | 0.96 | 3.44 | 0.96 | 3.29 |
| Average | 0.38 | 1.45 | 0.53 | 2.08 | 0.35 | 1.42 | 0.41 | 1.58 | 0.42 | 3.16 | 0.39 | 1.60 | 0.38 | 1.84 |

