# OpenReview forum: "Safe Multi-task Pretraining with Constraint Prioritized Decision Transformer"
_ICLR.cc/2025/Conference — Submitted to ICLR 2025_

### Official Review · Reviewer_GfAj · 2024-10-28

**Soundness:** 2
**Presentation:** 3
**Contribution:** 3
**Rating:** 6
**Confidence:** 3

**Summary:**

This paper proposes SMACOT, a framework for safe reinforcement learning from offline data, addressing both safety and task identification. By introducing a Constraint Prioritized Return-To-Go token, SMACOT balances reward and safety. Experiments on the OSRL dataset show SMACOT’s superior safety performance, meeting safety constraints in over twice as many environments compared to baselines.

**Strengths:**

1) Well-motivated approach
2) Experimental results are promising
3) Clear and well-structured writing

**Weaknesses:**

1) The cost constraint functions as a soft constraint, which does not fully guarantee meeting safety requirements.
2) In many cases, the Return-To-Go (RTG) set in Decision Transformer (DT) methods does not align with the actual return achieved, and the same issue applies to cost-to-go. The safty environment may be more sensitive.
3) Time complexity is unkown.

**Questions:**

1) How to distinguish tasks in unknown environments?
2) Given that the cost constraint functions as a soft constraint, how can it be ensured that safety requirements are fully met?
3) In DT-based framework, how to set proper cost to go?
4) What is the time complexity and the comparison of exact training time with other methods?

---

> ### Author Response · Authors · 2024-11-20
> **Response to Reviewer GfAj (1/2)**
>
> Thank you for your kind review. We have added a comparison of time complexity and a description of the task distinguishment method under unknown environments. Here are our responses to your proposed concerns.
>
> ### Q1 Soft constraint and hard constraint
>
> Safety under soft constraints and safety under hard constraints are both important research areas in safe reinforcement learning (RL) [1,2,3]. **Our work, similar to many prior studies [4,5,6,7], focuses on addressing the problem of safety under soft constraints**. Under the soft constraint setting, the policy is given a threshold $b$ for allowable safety constraint violations, and it is required that the expected cumulative safety constraint violation does not exceed this threshold [2]. Therefore, **the policy is not required to fully satisfy the safety constraints, allowing for some level of constraint violations**. In fact, many real-world applications involve soft constraints. For example, in autonomous driving, the vehicle's fuel level can be modeled as a soft constraint, requiring the vehicle to reach its destination before running out of fuel. Similarly, in scenarios like average speed monitoring, exceeding the speed limit at any given moment does not constitute a violation, but rather the requirement is that the overall average speed remains below a given threshold. Of course, safety under hard constraints is also a critical research issue in safe RL [8], and extending our approach to hard constraint scenarios is an important direction for future work.
>
> ### Q2 The setting of proper RTG and CTG in DT
>
> During deployment, the Return-To-Go (RTG) and Cost-To-Go (CTG) might not perfectly align with the true reward return and cost return of the policy. Therefore, **SMACOT (our method) handles and validates their settings with special consideration**. **One of our core contributions is the setting of RTG**. In **Appendix A**, we analyze RTG not from the perspective of the target condition but from the offline RL perspective, **treating RTG as part of the state and viewing Constraint Prioritized Return-To-Go** (**CPRTG) as a fitting of the RTG distribution from the offline data**. From this perspective, RTG no longer needs to align perfectly with the true reward return, and the policy's performance can still be guaranteed to a certain extent.
>
> For the setting of **CTG**, if we were to treat it similarly to RTG, it could also improve the performance bound of the policy. However, we aim to leverage the generalization capability of the Transformer for CTG to adapt to different safety thresholds and levels of conservatism. The most straightforward method of setting CTG is to **use the safety threshold $b$ provided by the soft constraint as the initial CTG and update it at each step using** $\hat{C}_{t+1} = \hat{C}_t - c_t$, where the true cost $c_t$ is subtracted. As seen in the experiment in **Appendix G.2**, the true cost return in SMACOT align with the initial CTG set by safety threshold in trend.
>
> Therefore, we can **adjust the initial CTG setting based on the rollout results during deployment in the real world**. **Initially, we set the CTG to the safety threshold and perform a rollout. If the rollout results are too conservative, we can increase the initial CTG. Conversely, if the rollout results are too aggressive, we can lower the initial CTG setting.**

---

> > ### Author Response · Authors · 2024-11-20
> > **Response to Reviewer GfAj (2/2)**
> >
> > ### Q3 Time complexity comparison
> >
> > We have added a comparison of the time complexity across different algorithms.
> >
> > |  | SMACOT (ST) | CDT | SMACOT (MT) | MTCDT | Prompt-CDT |
> > | :---: | :---: | :---: | :---: | :---: | :---: |
> > | Prompt Encoder Training | \ | \ | 1.330 h | \ | \ |
> > | DT Policy Training | 15.734 h | 15.737 h | 19.584 h | 19.288 h | 33.008 h |
> > | CPRTG Generator Training | 0.250 h | \ | 1.404 h | \ | \ |
> > | Deployment | 0.012 s/step | 0.008 s/step | 0.017 s/step | 0.008 s/step | 0.014 s/step |
> >
> > The results from the table show that **while the use of CPRTG and the prompt encoder in SMACOT introduces a slight time overhead during training and deployment, it is acceptable compared with the improvement in performance**. During training, whether in a single-task or multi-task setting, the additional time required for training the prompt encoder and the CPRTG generator is minimal due to their simple MLP networks. The training of each component **does not exceed one-tenth of the time spent on policy training** itself, making the extra time overhead small. In the multi-task setting, since SMACOT uses a single vector as the prompt instead of trajectory segments, the training time overhead is **less than two-thirds of that required by Prompt-CDT**. Therefore, SMACOT's training time complexity is relatively low. During deployment, since each step involves additional inference through the CPRTG generator's neural network, there is indeed some increase in time overhead. However, **the extra time per step does not exceed 0.01 seconds**, so the impact on policy deployment is negligible. For more results and analysis, please refer to **Appendix G.1** and **Appendix G.6**.
> >
> > ### Q4 Distinguish tasks in unknown environments
> >
> > Our method can effectively identify tasks even when the environment ID is unknown. Specifically, we begin by selecting potential candidate environments from those seen during training based on the **state and action space dimensions** of the unknown environment.
> >
> > Next, for a given trajectory in the unknown environment, we compute the following:
> >
> > 1. **Action Reconstruction Loss**: Using each candidate environment's **environment-specific action encoder and action decoder**, we calculate the average action reconstruction loss over the trajectory.
> > 2. **State Reconstruction Loss**: Using the **environment-specific state encoder and state decoder** of each candidate, we calculate the average state reconstruction loss over the trajectory.
> > 3. **Inverse Dynamics Error**: Using the **inverse dynamics model**, we compute the average inverse dynamics error for the trajectory.
> >
> > Finally, we select the candidate environment that **minimizes the sum of the action reconstruction loss, state reconstruction loss, and inverse dynamics error** as the inferred environment. Once the environment ID is determined, we use the corresponding environment-specific encoders and the Constraint Prioritized Prompt Encoder to encode the trajectory and obtain the prompt encoding, which is then used as the basis for task identification. This approach allows us to effectively identify the correct environment and subsequently perform accurate task identification even when the environment ID is initially unknown. For more details, please refer to **Appendix D**.
> >
> > We hope that our analysis and explanations can address your concerns about our paper. Please feel free to add a comment if you have further questions.
> >
> > > [1] Garcıa, Javier, and Fernando Fernández. "A comprehensive survey on safe reinforcement learning." JMLR 2015.\
> > [2] Gu, Shangding, et al. "A Review of Safe Reinforcement Learning: Methods, Theories and Applications." TPAMI 2024.\
> > [3] Wachi, Akifumi, Xun Shen, and Yanan Sui. "A Survey of Constraint Formulations in Safe Reinforcement Learning." CoRR 2024.\
> > [4] Achiam, Joshua, et al. "Constrained policy optimization." ICML 2017.\
> > [5] Liu, Zuxin, et al. "Constrained variational policy optimization for safe reinforcement learning." ICML 2022.\
> > [6] Liu, Zuxin, et al. "Constrained decision transformer for offline safe reinforcement learning." ICML 2023.\
> > [7] Guo, Zijian, Weichao Zhou, and Wenchao Li. "Temporal Logic Specification-Conditioned Decision Transformer for Offline Safe Reinforcement Learning." ICML 2024.\
> > [8] Zhao, Weiye, et al. "State-wise safe reinforcement learning: A survey." IJCAI 2023.
> > >

---

> > > ### Comment · Reviewer_GfAj · 2024-11-21
> > > **Thanks for the reply**
> > >
> > > Thanks for the reply. I have no further questions.

---

### Official Review · Reviewer_eMX9 · 2024-10-29

**Soundness:** 3
**Presentation:** 3
**Contribution:** 2
**Rating:** 5
**Confidence:** 4

**Summary:**

n this work, the authors propose a modified version of decision transformer framework which introduces a constraint prioritized return-to-go token that models the return-to-go token conditioned on the cost-to-go. The proposed framework also utilizes a specialized prompt encoder that helps identify tasks during inference by separately encoding safe and unsafe transitions. This approach introduces an effective method to learn safe RL policies using decision transformers while addressing the conflict between reward maximization and safety constraints. The authors conduct comprehensive evaluations on the OSRL dataset, demonstrating significant improvements over several baseline methods, with their approach achieving safe performance in more than twice as many environments.

**Strengths:**

- Overall, this paper is well-written. The authors did a great job in describing the multi-task safe RL problem. Also, the authors use figures to illustrate their proposed method
- The authors provided pretty comprehensive empirical evaluation on 26 tasks with thorough ablation studies and clear visualizations demonstrating each component's contribution
- Additional adaptation method such as low-rank adaptation show potential of such pretraining strategy.

**Weaknesses:**

- It would be great if the authors can provided theoretical analysis for the proposed method.
- Right now the evaluation seems only focusing on success/failure. The author should consider analysis of the trade-off between safety margin and performance.
- The transfer learning tasks are still conducted on relatively similar tasks.
- It seems that there is limited investigation of zero-shot generation performance.

**Questions:**

Does this pretraining strategy enable decision transformer to demonstrate in-context learning abilities?
- Have you explored whether SMACOT can adapt to slightly different safety thresholds or constraints without fine-tuning, similar to how large language models can adapt to new tasks through in-context examples?

---

> ### Author Response · Authors · 2024-11-20
> **Response to Reviewer eMX9 (1/2)**
>
> # Response to Reviewer eMX9
>
> Thank you for your careful reviews and constructive suggestions. We have added the corresponding analysis and prepared additional experimental results for your proposed weakness. We hope they can relieve your concern.
>
> ### Q1 Theoretical analysis
>
> We have added the theoretical analysis of our method in **Appendix A**. Intuitively, this theorem reveals that **the RTG prediction in Constraint Prioritized Return-To-Go (CPRTG) can be interpreted as a mechanism to improve the performance bound of the policy by fitting the RTG transitions in the offline data**.
>
>  To derive this theorem, we approach it from the perspective of offline reinforcement learning, no longer treating Return-To-Go (RTG) and Cost-To-Go (CTG) as conditions; instead, we consider them as part of the state. In such case, the RTG and CTG distributions in the offline dataset can be viewed as the true distributions, while the RTG and CTG during deployment are the results obtained from rollouts within a model. Consequently, by analyzing the accuracy of the model's rollouts, we can derive a performance bound for the trained Decision Transformer (DT) policy during deployment. This performance bound is influenced by three factors: the optimization level of the policy, the accuracy of CTG transitions, and the accuracy of RTG transitions. Since the optimization level of the policy is difficult to control, and we aim to rely on the Transformer's generalization capability for CTG to achieve safe decision-making under various safety thresholds, the best option for improving the performance bound is to modify the setting of RTG to fit the distribution observed in the offline dataset.
>
> ### Q2 Analysis of the trade-off between safety margin and performance
>
> The trade-off between safety margin and reward performance is indeed one of the primary additional challenges introduced by safe RL compared to traditional RL [1]. **SMACOT demonstrates a clear advantage in addressing this challenge by flexibly and adaptively adjusting the policy's level of conservatism within the given safety constraints based on external safety requirements**.
>
> First, from the perspective of **motivation**, traditional value-function-based RL methods can prioritize the safety margin in the trade-off between safety margin and reward performance by employing optimization techniques such as the Lagrangian multiplier method [2,3,4]. These methods aim to maximize reward as much as possible within the constraints of a given safety margin. However, their limitation lies in **handling only a single safety margin at a time**. On the other hand, safety-conditioned methods based on DT attempt to address multiple safety margins by conditioning the input through the Transformer [5]. However, these methods do not explicitly model the higher priority of the safety margin over reward performance, making them prone to **ignoring safety margin requirements and focusing solely on reward performance**. One of our primary motivations is to combine the strengths of these two approaches. By introducing CPRTG, we model the relationship between the safety margin and reward performance, ensuring that **the satisfaction of safety constraints comes first** for a given safety margin. Simultaneously, **as the safety margin changes, the reward performance goal is adjusted accordingly**, avoiding a complete sacrifice of reward performance for the sake of the safety margin.
>
> From the experimental perspective, the results further validate this point. The experiments in **Appendix G.2**, which evaluate the policy's performance as the safety margin (threshold) changes, clearly demonstrate that **SMACOT can adapt its conservatism according to the given safety margin**. When the safety margin is low, SMACOT prioritizes safety. As the safety margin gradually increases, SMACOT progressively reduces its conservatism while remaining within safety constraints, thereby improving reward performance. In contrast, CDT, despite achieving higher reward performance, exhibits significant safety compromises in three out of four environments. It is unable to adjust its conservatism based on external safety margins, leading to severe violations of safety constraints when strict safety margins are required. Additionally, the ablation study on the hyperparameter $\beta_{\text{end}}$ in **Section 5.3** highlights SMACOT's **flexibility in handling the trade-off between safety margin and reward performance**. When reward performance is prioritized, SMACOT can adopt more aggressive decisions by increasing $\beta_{\text{end}}$. Conversely, when safety is prioritized, SMACOT can enhance its conservatism by reducing $\beta_{\text{end}}$, ensuring adaptability to varying safety requirements.
>
> If we have not explained something clearly, please let us know, and we would be happy to provide further clarification. For more results and analysis, please refer to **Appendix G.2**.

---

> > ### Author Response · Authors · 2024-11-20
> > **Response to Reviewer eMX9 (2/2)**
> >
> > ### Q3 Transfer to a dissimilar task
> >
> > We have added a transfer experiment in a task dissimilar from the pretraining task. The results show that **SMACOT's pretraining still brings performance improvement during transfer.** These results demonstrate that even in scenarios with low task similarity, SMACOT's multi-task pretraining can still achieve a certain degree of knowledge transfer between tasks, facilitating efficient policy learning in low-data regimes. However, due to the limited amount of transferable knowledge in low-similarity tasks, the benefits of multi-task pretraining are less pronounced compared to those observed in similar-task scenarios. This finding highlights the potential value of incorporating a more diverse set of tasks during pretraining to enhance the capabilities of the pretrained policy in future work. For detailed results and more information, please refer to **Appendix G.8**.
> >
> > ### Q4 About the zero-shot generalization performance
> >
> > Our work primarily focuses on learning safe policies and identifying tasks in multi-task scenarios, rather than exploring zero-shot generalization capabilities. Therefore, the experiments were conducted on the OSRL dataset, which contains a limited number of similar tasks. The results of the additional conducted zero-shot generalization experiments confirm that, **with pretraining on only a small number of similar tasks, neither SMACOT nor other baseline algorithms demonstrated satisfactory generalization performance**. And we leave it in the future work for pretraining on dozens or even hundreds of similar tasks for scaling law of large safe decision model for better generalization.
> >
> > We have also added experiments to investigate the performance of the policy as the safety threshold increases from 10 to 100 in steps of 10. The results clearly demonstrate that **SMACOT (our method) exhibits strong adaptability to various safety thresholds**. When the safety threshold is low, SMACOT increases its conservatism to ensure safety constraints are met. On the other hand, when the safety threshold is high, SMACOT reduces its conservatism accordingly to improve reward performance. In contrast, the baseline algorithm **CDT fails to show good adjustment capabilities for the safety threshold** in three out of four tasks. While CDT performs well when the safety threshold is large, it continues to exhibit aggressive behavior and severely violates safety constraints when the safety threshold is low.
> >
> > From the perspective of in-context learning, enabling the policy to generalize across diverse tasks and achieve varying levels of conservatism by leveraging given trajectory contexts and safety threshold requirements, is indeed our ultimate vision. We hope that through further research, we can make significant progress toward generalizable safe decision-making by training on larger-scale tasks and datasets. For detailed results and more analysis, please refer to **Appendix G.2**.
> >
> > We hope our clarifications and answers can help address your concerns. Please let us know if there are further questions.
> >
> > > [1] Gu, Shangding, et al. "A Review of Safe Reinforcement Learning: Methods, Theories and Applications." TPAMI 2024.\
> > [2] Achiam, Joshua, et al. "Constrained policy optimization." ICML 2017.\
> > [3] Stooke, Adam, Joshua Achiam, and Pieter Abbeel. "Responsive safety in reinforcement learning by pid lagrangian methods." ICML 2020.\
> > [4] Liu, Zuxin, et al. "Constrained variational policy optimization for safe reinforcement learning." ICML 2022.\
> > [5] Liu, Zuxin, et al. "Constrained decision transformer for offline safe reinforcement learning." ICML 2023.
> > >

---

> > > ### Author Response · Authors · 2024-11-23
> > > **Dear Reviewer eMX9, are our responses address your questions?**
> > >
> > > Dear Reviewer eMX9:
> > >
> > > We thank you again for your comments and hope our responses could address your questions. As the response system will end in five days, please let us know if we missed anything. More questions on our paper are always welcomed. If there are no more questions, we will appreciate it if you can kindly raise the score.
> > >
> > > Sincerely yours,
> > >
> > > Authors of Paper6916

---

> > > > ### Comment · Reviewer_eMX9 · 2024-11-27
> > > > **Reponse**
> > > >
> > > > I would like to thank the authors for their diligent response, as well as for the following contributions: (1) adding new theoretical analyses, (2) providing discussions regarding the trade-off between safety margins and performance, (3) exploring transfer to dissimilar tasks, and (4) addressing zero-shot generalization.
> > > >
> > > > While the current theoretical claims provide an encouraging argument, it appears that the proof does not necessarily demonstrate the improvement achieved by using RTG and CTG tokens. The theoretical results build upon existing lemmas in [Janner et al. 2019]; however, in Theorem A.1 of [Janner et al. 2019], they introduce and subtract a reference return term. This reference return is obtained by executing one policy under two bounded dynamics (via their Lemma B.3). However, the proof (specifically in Eq. (24) and Eq. (25)) skips directly to the final results. Consequently, the theoretical results only show that the expected return and cost are close to those obtained without using RTG and CTG tokens.
> > > >
> > > > I believe the underlying issue is that the theoretical results in [Janner et al. 2019] analyze the monotonic model-based improvement by bounding the estimated return and the ground truth return. Here, however, the two returns and costs being compared are both estimated terms. Additionally, in Lemma 2, you assume the two dynamics are bounded by $\epsilon$. However, in line 856 (Bayes rule), you assume the two dynamics are the same. Does this not contradict the assumption in Lemma 2?
> > > >
> > > > I appreciate the addition of results for transfer to dissimilar tasks. However, I do not believe multi-task learning is fundamentally different from zero-shot generalization. In fact, I think one of the primary motivations for using decision transformers instead of traditional MLP-based deep reinforcement learning is to exploit the generalization capability brought by the auto-regressive transformer model. This should also be the goal of tokenizing certain types of information.
> > > >
> > > > In conclusion, I commend the authors’ efforts in addressing some of my concerns. However, I believe more substantial improvements are needed to convincingly demonstrate the significance of the proposed CPRTG token and the pre-training framework. I remain on the fence and will keep my scores unchanged.

---

> > > > > ### Author Response · Authors · 2024-11-27
> > > > > **Further response to Reviewer eMX9**
> > > > >
> > > > > Thank you very much for your further suggestions. We will provide answers to your questions regarding the theoretical analysis and zero-shot performance.
> > > > >
> > > > > ## Q1 About theoretical analysis
> > > > >
> > > > > First, our theoretical proof **does not contradict Lemma 2**. In Lemma 2, the dynamics transitions $p_1(s'|s,a)$ and $p_2(s'|s,a)$ are assumed to have their total variation distance (TVD) limited by $\epsilon_m$. However, in our theoretical proof, we **treat the RTG and CTG as part of the state**, and thus modify the aforementioned dynamics transitions to $p_1(s’,\hat{R}\_{t+1},\hat{C}\_{t+1}|s,\hat{R}_t,\hat{C}_t,a)$ and $p_2(s’,\hat{R}\_{t+1},\hat{C}\_{t+1}|s,\hat{R}_t,\hat{C}_t,a)$. We then **decompose these using Bayes' rule**. As a result, the actual assumption in our Theorem is that the **TVD between** $p_1(s’,\hat{R}\_{t+1},\hat{C}\_{t+1}|s,\hat{R}_t,\hat{C}_t,a)$ **and** $p_2(s’,\hat{R}\_{t+1},\hat{C}\_{t+1}|s,\hat{R}_t,\hat{C}_t,a)$ **is bounded by** $\epsilon_R+\epsilon_C$. The reason we claim that $p_1(s'|s,a)$ and $p_2(s'|s,a)$ are the same is because of our interpretation of $p_1$ and $p_2$. Specifically, $p_1$ represents the state transitions encountered during policy deployment, while $p_2$ represents the state transitions in the offline dataset. Since the **offline dataset and the deployment environment are consistent, the state transitions themselves are identical**. The jump between Equation (23) and Equation (25) is essentially the result of the TVD being bounded by $\epsilon_R+\epsilon_C$ between $p_1(s’,\hat{R}\_{t+1},\hat{C}\_{t+1}|s,\hat{R}_t,\hat{C}_t,a)$ and $p_2(s’,\hat{R}\_{t+1},\hat{C}\_{t+1}|s,\hat{R}_t,\hat{C}_t,a)$. We apologize for any confusion this may have caused, and we have **added more detailed descriptions** to clarify this point.
> > > > >
> > > > > Secondly, the purpose of our theory is **not to demonstrate the additional benefits brought by using CTG and RTG**, but rather to show the **improvement achieved by using CPRTG compared to the traditional RTG**. Specifically, our theory explains that by modeling RTG on top of CTG, CPRTG can **achieve a higher performance bound**, especially in terms of the upper limit for violations of safety constraints. It is precisely this enhanced performance bound that allows SMACOT to achieve significantly better safety performance compared to CDT, and our experiments have validated this result.
> > > > >
> > > > > Finally, the two types of returns we compare here are **not both estimated returns**. Specifically, $\eta_1$ represents the **estimated return during policy deployment**, while $\eta_2$ represents the return of the behavior policy in the offline dataset under the transition distribution of the offline data, which is **essentially the return of the trajectories in the offline dataset**. From a model-based perspective, MBPO aims to bound the ground truth return using the estimated return, whereas in our case, we are attempting to **bound the estimated return using the ground truth return**.
> > > > >
> > > > > ## Q2 About zero-shot performance
> > > > >
> > > > > Our main contribution in multi-task pretraining is solving the task misidentification issue caused by cost sparsity, and multi-task pretraining allows us to obtain a policy that can **simultaneously address multiple training tasks and quickly adapt to new, similar tasks**. Achieving zero-shot capability is indeed one of our visions, but not our main motivation. Through the use of DT, we have observed that the policy can indeed achieve **zero-shot** capability **across different safety thresholds**. However, when it comes to different cost functions, due to the limited number of similar tasks in our training set, the policy's zero-shot ability is not fully demonstrated at this stage. To further investigate this, we are conducting an additional experiment by pretraining on a larger number of similar tasks to more clearly showcase the zero-shot potential. Due to the time constraint, we will try our best to provide the corresponding test results for zero-shot capability upon finishing the experiments.

---

> > > > > ### Author Response · Authors · 2024-12-02
> > > > > **Dear Reviewer eMX9, are our responses address your further questions?**
> > > > >
> > > > > Dear Reviewer eMX9:
> > > > >
> > > > > We would like to express our sincere gratitude for taking the time to evaluate our paper and for your continued support of our community.
> > > > >
> > > > > In response to your concerns, we have provided additional explanations and conducted further experiments to demonstrate the zero-shot generalization ability of our method. As the response system will close in **two days** (**one day** for reviewers to respond), please let us know if we have overlooked anything. We welcome any further questions or feedback on our paper.
> > > > >
> > > > > Sincerely yours,
> > > > >
> > > > > Authors of Paper6916

---

> ### Author Response · Authors · 2024-12-02
> **More results on zero-shot generalization**
>
> Dear Reviewer eMX9,
>
>  Thank you very much for your interest in the zero-shot generalization capability. We have conducted an additional experiment in AntVel to test how zero-shot generalization changes after pretraining on a larger number of similar tasks. The results show that **as the number of similar pretraining tasks increases, the zero-shot generalization ability indeed improves**. In this experiment, in addition to the policy pretrained on 2 similar tasks in the main experiment, we also trained a policy on 6 similar tasks and tested it on two unseen tasks: AntV2 (with a velocity limit of 2.52) and AntV7 (with a velocity limit of 2.39). Among the pretraining tasks, besides the main experiment tasks (AntV0 with a velocity limit of 2.57 and AntV1 with a velocity limit of 2.62), we also added additional tasks with velocity limits of 2.67, 2.55, 2.47, and 2.42. The experimental results show that as the number of pretraining tasks increases, the policy can more effectively leverage the knowledge from additional pretraining tasks, achieving stronger zero-shot generalization. This further supports our vision that if SMACOT is pretrained on a larger number of tasks (over a hundred), it could demonstrate even more powerful zero-shot generalization abilities.
>
> |  | SMACOT (2 similar tasks) |  | SMACOT (6 similar tasks) |  |
> | :---: | :---: | :---: | :---: | :---: |
> |  | reward | cost | reward | cost |
> | AntV2 | 0.99 | 2.18 | 0.98 | 1.56 |
> | AntV7 | 0.99 | 3.28 | 0.96 | 2.34 |

---

### Official Review · Reviewer_xwJ3 · 2024-11-04

**Soundness:** 3
**Presentation:** 2
**Contribution:** 3
**Rating:** 8
**Confidence:** 3

**Summary:**

The authors propose a new approach called Safe Multi-task Pretraining with Constraint Prioritized Decision Transformer (SMACOT) to address the challenge of learning safe policies from offline data in reinforcement learning (RL). SMACOT uses a transformer-based architecture that can accommodate varying safety threshold objectives and ensure scalability.

**Strengths:**

The key innovations include:
1. Constraint Prioritized Return-To-Go (CPRTG): a token that emphasizes cost priorities in the inference process, balancing reward maximization with safety constraints.
2. Constraint Prioritized Prompt Encoder: designed to leverage the sparsity of cost information for task identification.
3. As a result, experiments on the public OSRL dataset show that SMACOT achieves exceptional safety performance in both single-task and multi-task scenarios, satisfying different safety constraints in over 2x as many environments compared with strong baselines.

**Weaknesses:**

1. No mentioning of the work about "Trajectory Transformer" (TT) [1] which is quite similar to Decision Transformer but focuses on the beam-search-based planning as opposed to reward conditioning for DT. The work would be even more solidified if having TT as a baseline.
2. No any ablation / reasoning behind the number $X$ of the samples used to select the $\beta_t$-quantile for $\tilde{R}_t$
3. No any ablation behind the need for the dynamics model $g_i$ in the Section 4.2 for environment-specific encoders.

Additionally, some misprints like "chosose" on Line 237.

[1] Janner, Michael, Qiyang Li, and Sergey Levine. "Offline reinforcement learning as one big sequence modeling problem." Advances in neural information processing systems 34 (2021): 1273-1286.

**Questions:**

N/A

---

> ### Author Response · Authors · 2024-11-20
> **Response to Reviewer xwJ3**
>
> Thank you for your inspiring and thoughtful reviews. We have prepared the following experimental results and comments for your proposed weakness, and we hope they can relieve your concern.
>
> ### Q1 Comparison with Trajectory Transformer (TT)
>
> We have added additional comparison experiments with TT, and the results are consistent with previous comparisons.
>
> | Task | TT |  | SMACOT |  |
> | :---: | :---: | :---: | :---: | :---: |
> |  | reward   | cost | reward | cost |
> | PointButton1 | **0.05** | **0.86** | **0.05** | **0.66** |
> | PointButton2 | 0.15 | 1.90 | 0.14 | 1.41 |
> | PointGoal1 | **0.24** | **0.61** | **0.36** | **0.56** |
> | PointGoal2 | 0.27 | 1.13 | 0.31 | 1.02 |
> | Average | 0.18 | 1.13 | 0.22 | 0.91 |
>
>  **SMACOT (our method) demonstrates superior performance than TT in both safety and reward performance**. However, an additional observation is that **TT performs significantly better than the baseline CDT in terms of safety**. We believe this occurs because TT uses a BC-based training method and does not incorporate the additional Return-To-Go (RTG) and Cost-To-Go (CTG) inputs, thus **avoiding the conflict between RTG and CTG**.
>
> However, it could introduce other problems. First, when the quality of the offline dataset is poor, TT's policy performance deteriorates because it cannot select relatively optimal trajectories for imitation. Second, once the policy is trained, TT's conservativeness is fixed, and it cannot adjust its conservativeness according to different safety thresholds or other parameters. Therefore, when safety requirements are relaxed, TT may show a significant disadvantage in terms of reward performance. For detailed results and more information, please refer to **Appendix G.5**.
>
> ### Q2 Ablation on Constraint Prioritized Return-To-Go (CPRTG) sample number $X$
>
> We have added additional ablation results on the CPRTG sample number $X$. From the results, we can draw two conclusions:
>
> 1. As $X$ increases, the sampled results become closer to the quantile targets we set, making CPRTG more stable, leading to a **significant improvement in the policy's safety performance**.
> 2. As$X$ increases, **the time overhead during policy deployment does not show a significant increase**, allowing us to increase $X$ as much as possible within the allowable range to achieve better performance.
>
> For detailed results and more information, please refer to **Appendix G.6**.
>
> ### Q3 Ablation on inverse dynamics model $g$
>
> The primary purpose of using the inverse dynamics model is to **handle the case where state space and action space are identical but the dynamics transitions differ across tasks**. Although such scenarios do not appear in the OSRL dataset we used, this situation is quite common in reinforcement learning [1,2,3]. In these cases, the use of the inverse dynamics model $g$ allows us to **obtain different state representations for tasks with identical state and action spaces but different dynamics transitions, simplifying the task classification challenge for the Constraint Prioritized** **Prompt Encoder**. Moreover, when the **environment ID is unknown**, the inverse dynamics error, based on the inverse dynamics model, becomes **a core method for identifying environments and tasks with identical state-action spaces but different dynamics transitions**. We also conducted additional ablation experiments, confirming that even in the absence of tasks with differing dynamics transitions, the use of the inverse dynamics model does not negatively impact performance.
>
> | Task | W/o IDM |  | SMACOT |  |
> | :---: | :---: | :---: | :---: | :---: |
> |  | reward | cost | reward | cost |
> | PointButton1 | **0.07** | **0.62** | **0.04** | **0.55** |
> | PointButton2 | 0.10 | 1.05 | **0.08** | **0.98** |
> | PointGoal1 | **0.26** | **0.28** | **0.24** | **0.30** |
> | PointGoal2 | **0.23** | **0.63** | **0.26** | **0.66** |
> | Average | 0.17 | 0.65 | 0.16 | 0.62 |
>
> For more analysis and results, please refer to **Appendix D** and **Appendix G.7**.
>
> We hope that our additional experiments can address your concerns about our paper. Please feel free to add a comment if you have further questions.
>
> > [1] Nagabandi, Anusha, et al. "Learning to Adapt in Dynamic, Real-World Environments through Meta-Reinforcement Learning." ICLR 2019.\
> [2] Eysenbach, Benjamin, et al. "Off-Dynamics Reinforcement Learning: Training for Transfer with Domain Classifiers." ICLR 2021.\
> [3] Zhang, Xinyu, et al. "Debiased Offline Representation Learning for Fast Online Adaptation in Non-stationary Dynamics." ICML 2024.
> >

---

> > ### Comment · Reviewer_xwJ3 · 2024-11-20
> >
> > Thanks for the detailed answers! they fully address my concerns, especially in terms of inverse dynamic model. Increasing the score toward the acceptance.

---

### Official Review · Reviewer_BUtY · 2024-11-04

**Soundness:** 2
**Presentation:** 1
**Contribution:** 2
**Rating:** 3
**Confidence:** 4

**Summary:**

This paper proposes the Safe Multi-task Pretraining with Constraint Prioritized Decision Transformer (SMACOT) for safe offline multi-task reinforcement learning (RL). SMACOT aims to address challenges in safe multi-task RL by using a Constraint Prioritized Return-To-Go (CPRTG) token, which prioritizes safety constraints over reward maximization during training. Additionally, the model employs a Constraint Prioritized Prompt Encoder to aid in task identification using sparse cost information. Experimental results on OSRL tasks reportedly show superior safety performance compared to baselines.

**Strengths:**

- **Innovative use of transformers in Safe RL**: The paper extends the Constrained Decision Transformer to accommodate safety prioritization in multi-task RL through a novel cost-priorized returns-to-go token and a prompt encoder, similar to the prompt DT structures in unconstrained offline RL settings.
- **Somewhat clear motivation**: The motivation to prioritize safety constraints and manage cost sparsity issues is well-articulated, which addresses a current challenge in safe RL.

**Weaknesses:**

Several critical weaknesses significantly limit the contribution, novelty, methodology clarity of this work:

- **Necessity of safety prioritization**: for the Constrained Decision Transformer structure the authors adopted, the action token is essentially conditioned on all the costs, rewards, and states tokens. However, it remains unclear to me what is the specific advantage of this token order: CTG-State-RTG-Action. For example, if the authors want to have a more accurate cost token inputs, why aren't then using other orders like state-CTG-RTG-Action?
- **Marginal technical contribution over existing work**: The CPRTG and Constraint Prioritized Prompt Encoder for multi-task safe RL seem like minor extensions over established safety-conditioned reinforcement learning methods (e.g., [1, 2]). A comparison with these existing works might be helpful.
- **Ambiguity for task identification**: the authors mentioned that they remove cost information in the constraint prioritized prompt encoder. However, the authors fail to present any technical details about they can use the cost and distinguish the task merely by the 'input distribution' of state, action, and reward. It seems like the prompt encoder splits the safe and unsafe tokens and still conduct a (reweighted version of) the next token prediction. See questions for more details
- **Unrelated experiment setting**: in research question (2), the authors mentioned they evaluate their approaches in single-agent and multi-agent safe RL tasks. However, the methodology aims to resolve multi-task safe RL instead of multi-agent safe RL.
- **Unclear experiment contribution**: in the title, the authors mention about safe multi-task pertaining. However, they evaluate the policy transfer results in the experiment as well, and evaluate the difference between training from scratch, FFT and LoRA approaches. However, if the contribution is the CPRTG and prompt encoder in the pretraining phase and still want to show the few-shot adaptivity, they should evaluate the performance of such fine-tuning techniques over other baselines for pertaining as well.


> [1] Yao, Yihang, et al. "Constraint-conditioned policy optimization for versatile safe reinforcement learning." NeurIPS 2023.
>
> [2] Guo, Zijian, et al., "Temporal Logic Specification-Conditioned Decision Transformer for Offline Safe Reinforcement Learning" ICML 2024.

**Questions:**

- **About CPRTG**: eventually the output of the model is the action. Is there any difference in the Bayes factorization by ordering CTG, state and RTG one way or another? For example, consider this one-step conditioned generation:

$$
\begin{aligned}
p(\hat{a}_t, s_t, \hat{C}_t, \hat{R}_t | \{\tau\}\_\{t-1\}) & \propto p(\hat{a}_t |\hat{R}_t,\{\tau\}\_\{t-1\}) p(\hat{R}_t | s_t, \hat{C}_t,\{\tau\}\_\{t-1\}) p(s_t|\hat{C}_t,\{\tau\}\_\{t-1\}) p(\hat{C}_t| \{\tau\}\_\{t-1\})  \\\\
& \propto  p(\hat{a}_t |s_t,\{\tau\}\_\{t-1\}) p(s_t | \hat{R}_t, \hat{C}_t, \{\tau\}\_\{t-1\}) p(\hat{R}_t|\hat{C}_t, \{\tau\}\_\{t-1\}) p(\hat{C}_t|\{\tau\}\_\{t-1\})
\end{aligned}
$$

- **About task identification**:
  - What is the definition of a task? Is it a cost threshold? Or is it a different morphology of the agent and navigation tasks in OSRL?
  - How do you identify a task? Is it by a classifier head of the output in the transformer?
  - How do you parameterize the joint distribution of the reward, state and action and use that to identify the task?

In general, despite a lot of empirical results, the the paper is poorly written and hard to follow. The main points seem to be very ambiguous and keep diverging throughout the paper. I would encourage the authors to rethink their major contribution and significanly revise this paper before it is ready for the top ML conferences.

---

> ### Author Response · Authors · 2024-11-20
> **Response to Reviewer BUtY (1/5)**
>
> Thank you for your constructive suggestions, we sincerely apologize for the confusion caused by some unclear phrasing and typographical errors. We have now corrected typographical errors and provided some clarifications for your comments, and we hope they can help address your concerns about our paper.
>
> ### Q1 **Necessity of safety prioritization**
>
> In Decision Transformer (DT), **the order of state, Cost-To-Go (CTG) and Return-To-Go (RTG) will not affect the policy training or action selection result**. The reasons are as follows:
>
> - First, it is important to emphasize that although DT uses the Transformer framework, it is different from traditional Transformer which **autoregressively generates each token.** DT’s prediction target is solely the action, while **the state, CTG, and RTG tokens are externally provided** and do not require to be predicted. Specifically, they come from the environment, being set with initial values and updated according to corresponding update formula, or being generated like our proposed CPRTG, respectively [1]. We have added the clarification in **Section 3.2**.
> - Next, the attention mechanism in the Transformer block is **order-independent**. To distinguish different tokens, we created separate embedding layers for CTG, state, and RTG. Therefore, the differences between the three tokens are **primarily reflected through the embedding layers**, rather than the **order** in which they are processed.
> - Meanwhile, in DT, the positional embedding in the Transformer block is replaced by sequence embedding. The sequence embeddings will **remain the same for CTG, state, and RTG at the same time step $t$** .
>
> Therefore, **the order of state, CTG and RTG will not affect the policy training or action selection result. There is no necessity of safety prioritization in terms of order.**
>
> In our work, constraint prioritization refers to the fact that in the traditional DT framework, both CTG and RTG are initialized with externally given values and updated at each time step based on their respective formulas (given in **Section 3.2**). This could lead to potential conflicts between the CTG and RTG. In contrast,  our Constraint Prioritized Return-To-Go (CPRTG) **uses an additional neural network to model the RTG objective, predicting it using the current time step's state and CTG information**. This ensures that, **given an external CTG target, the RTG is automatically generated in a way that satisfies the CTG goal while maximizing the reward as much as possible**. Therefore, **the prioritization is reflected in the** **neural network modeling aspect**. Moreover, this neural network is **not part of the DT** itself, so the order of inputs to the DT does not affect the results.

---

> > ### Author Response · Authors · 2024-11-20
> > **Response to Reviewer BUtY (2/5)**
> >
> > ### Q2 **Marginal technical contribution over existing work**
> >
> > Our work is the first to effectively leverage the Transformer architecture to address the problem of multi-task offline safe reinforcement learning (RL). It innovatively uses neural networks to model the higher priority of CTG over RTG, effectively resolving the core challenge of conflicts between reward and cost conditions in previous safety-conditioned methods. Additionally, it makes innovative use of the sparse binary nature of cost to design the prompt encoder structure, enabling efficient task identification. Below are the comparisons with the existing work:
> >
> > - **Constraint-conditioned policy optimization for versatile safe reinforcement learning (CCPO)**:
> >
> >     Compared to CCPO, our method shares similarities in conditioning the constraint during input to enable the policy's adaptation to different constraint thresholds. However, there are **several key differences** between two approaches.
> >
> >     First, **CCPO is an online reinforcement learning (RL) approach, while our method focuses on offline setting**. In the offline setting, the policy training does not involve any additional unsafe interactions with the environment, making it an ideal framework for learning safe policies [2].
> >
> >     Second, instead of the adapting to different constraint thresholds through constraint-conditioned inputs, which just benefits from DT framework such as CDT [3], our **main contribution** lies in **addresses the core challenge in constraint-conditioned work**—conflicts between CTG and RTG objectives.  To deal with this challenge explicitly highlighted in CDT [3], which can lead to suboptimal safety performance, we propose to **model the differing priority relationships between the constraint-condition CTG and reward-condition RTG through the neural network**.
> >
> > - **Temporal Logic Specification-Conditioned Decision Transformer for Offline Safe Reinforcement Learning (SDT)**:
> >
> >     Compared to SDT, both methods attempt to use the DT framework to address the offline safe RL problem, but the **focuses of two works are entirely different**.
> >
> >     The goal of SDT is to modify the constraint-conditioned approach by **incorporating more task-related safety priors into the DT framework using temporal logic**, thereby improving the safety of the resulting policy.
> >
> >     In contrast, our method **focuses on addressing the core challenge of conflicts between CTG and RTG under the traditional constraint-conditioned approach**. To the best of our knowledge, previous work on constraint-conditioning such as CDT and SDT primarily focused on how to design the constraint condition. Our method is **the first to explicitly identify the core challenge and shift the design from constraint condition to reward condition**. This perspective shift is undoubtedly innovative. We believe that, although SDT and our method address offline safe RL from different angles, both are effective solutions. In the future, combining these two approaches could lead to further advancements in the field.
> >
> >
> > We have added a more detailed introduction to these two methods in **Appendix B.1**, hoping to provide readers with a clearer understanding. The comparison with the two methods above highlights the innovative contributions of our approach in the constraint-conditioned aspect. Additionally, our method also investigates the task representation problem in **multi-task safe RL** and demonstrates that, under safe settings, multi-task pretraining can indeed have a beneficial impact on few-shot adaptation for similar tasks. Therefore, we believe that our method has not only marginal technical contributions over existing work.

---

> > > ### Author Response · Authors · 2024-11-20
> > > **# Response to Reviewer BUtY (3/5)**
> > >
> > > ### Q3 **Ambiguity for task identification**
> > >
> > > - **About task definition**:
> > >
> > >     **Each task is an independent Constrained Markov Decision Process (CMDP)** (details could be found in **Section 3.1**), and the **differences between tasks may arise from six aspects of the CMDP: state space, action space, dynamics transition, reward function, cost function, and safety threshold**. This way of task definition is widely used in both multi-task RL and meta RL [4].
> > >
> > >     In the experiments of our work, different navigation tasks in OSRL lead to differences in cost function and reward fu nction, and option of different robots (Ant, Hopper, etc.) lead to differences in state space, action space, and dynamics transition. Thus, it can be simply understood as a different morphology of the robot and navigation tasks in OSRL.
> > >
> > >     For example, in OSRL, PointGoal1 and PointGoal2 share the same state space, action space, dynamics transition, and reward function, but differ in cost function. On the other hand, PointButton1 and PointCircle1 differ in state space, action space, dynamics transition, reward function, and cost function. We have revised the wording in **Section 3.1** to provide a clearer definition of the task.
> > >
> > > - **About task identification**:
> > >
> > >     During task identification, we obtain a trajectory corresponding to the task and extract task-related information for task recognition, which is consistent with the setup in previous Transformer-based multi-task RL approaches [5,6]. However, our method differs in the way of processing the given trajectory. Previous methods directly incorporate the trajectory as part of the Transformer model sequence input, while our method **introduces an additional neural network prompt encoder**. This encoder first encodes the trajectory into a single prompt vector, which is then used as the first token input in DT. This approach of encoding the trajectory via a neural network prompt encoder is **adopted in context-based meta RL** [7,8,9].
> > >
> > >     In specific, given a trajectory $(s_1,a_1,r_1,c_1,\dots,s_T,a_T,r_T,c_T)$ of length $T$, we first  transform it into a batch form, $\{(s_i,a_i,r_i,c_i)\}\_{i=1}^T$. Next, we divide the batch into two parts—safe batch and unsafe batch—based on the cost-related information $c_i$ for each sample in the batch. Let the safe batch be  $\{(s_j,a_j,r_j)\}\_{j=1}^{T_s}$, where $T_s=\sum_{i=1}^T \mathbb I(c_i)$, we extract $T_s$ vectors by feeding them into the MLP network $p_s(s, a, r)$. Meanwhile, the unsafe batch with $T-T_s$ transitions will be fed into a different MLP network $p_u$ to obtain corresponding vectors. Finally, we derive the prompt vector by averaging such $T_s + (T - T_s)$ output vectors.
> > >
> > >     For training the MLPs $(p_s,p_u)$ used for the safe and unsafe batches, we employ a **training method similar to that of an AutoEncoder** [10]. Specifically, we introduce decoder networks $f_c,f_c,f_r$ to calculate the reconstruction error. In our method, the reconstruction error does not involve simply reconstructing the entire input batch but instead focuses on **reconstructing the next state $s_{i+1}$, reward $r_i$, and cost $c_i$ given the input sample $(s_i,a_i)$**. The reason for reconstructing these three targets is that, in our task definition, we assume that different tasks may vary in state space, reward function, and cost function. Therefore, we want the encoder to ultimately generate a prompt vector that retains information about these three components. The reconstruction error **allows the gradients to be backpropagated to the MLP networks** $(p_s,p_u)$, thereby enabling training.  After training (which is **decoupled from the policy training**), **only the encoder network will participate in the subsequent policy training and deployment, and its parameters will no longer be updated**. **The decoder network will not be used during this phase**. We have visualized the encoding obtained through this prompt encoder in **Appendix G.4**. The results clearly demonstrate that this approach effectively distinguishes between all the training tasks.
> > >
> > >     In the figure of our framework, for simplicity and clarity, we **omit the decoder networks $f_s,f_r,f_c$ and the training process of encoder**, but only present **the use of the trained encoder in the DT policy training process**. Therefore, the decoder network is actually **unrelated to the DT policy and is not a part of the output head**.  We have also added a more detailed description of the task identification process in **Appendix C** to help readers better understand our method.
> > >
> > >
> > > ### Q4 **Unrelated experiment setting**
> > >
> > > We apologize for the confusion caused by **typographical error**. We have corrected the term "agent" to "task" in **Section 5** to maintain consistency.

---

> > > > ### Author Response · Authors · 2024-11-20
> > > > **# Response to Reviewer BUtY (4/5)**
> > > >
> > > > ### Q5 **Unclear experiment contribution**
> > > >
> > > > We have added a comparison between SMACOT and another multi-task baseline, Prompt-CDT, under two different fine-tuning methods: FFT and LoRA. The experimental results show that, whether in similar or dissimilar tasks, **SMACOT outperforms Prompt-CDT in task transfer performance**. This demonstrates that SMACOT's use of the prompt encoder provides more effective information for knowledge transfer than directly using sequence prompts.
> > > >
> > > > In fact, the logic behind our experiments is as follows:
> > > >
> > > > - The motivation for using **CPRTG** is to address the conflict between RTG and CTG when using the DT architecture for safe policy learning. Through **experimental results under both the Oracle and Single-Task settings, we have thoroughly demonstrated that our method effectively resolves this issue**, enabling the Transformer to be applied successfully in this context.
> > > > - The motivation for using the **Constraint Prioritized Prompt Encoder** is to effectively leverage the sparse binary nature of the cost to facilitate task identification, thereby extending the method to the multi-task setting. **In the multi-task setting, experimental results comparing our approach with other baselines have successfully demonstrated that our method achieves more accurate task identification**, highlighting the effectiveness of the Constraint Prioritized Prompt Encoder.
> > > > - It is precisely due to **the use of the previous two components that our method is able to pre-train a policy that performs well across multiple training tasks**. In our task adaptation experiments, the primary goal is to **demonstrate that multi-task pretraining, in itself, provides benefits in few-shot transfer for similar tasks**, compared to learning from scratch.
> > > >
> > > > In conclusion, the overall logic is as follows: The use of CPRTG and Constraint Prioritized Prompt Encoder leads to a strong pretrained policy → the multi-task pretraining within this good pretrained policy is beneficial for task adaptation. Therefore, the experiment in Section 5.4 is primarily designed to **validate whether multi-task pretraining can enable the policy to transfer more efficiently to similar tasks, rather than focusing on the impact of our components on task transfer**. Since other baselines even did not learn a good pretraining policy during the pretraining phase, we have thus omitted comparisons with them regarding transfer performance. For more detailed results and analysis, please refer to **Appendix G.8**.

---

> > > > > ### Author Response · Authors · 2024-11-20
> > > > > **# Response to Reviewer BUtY (5/5)**
> > > > >
> > > > > ### Q6 The order in CPRTG
> > > > >
> > > > > In fact, **the order of state, RTG, and CTG in this formula does have an impact**. First, for the state $s_t$, since its transition is fully determined by the dynamics transition of the CMDP, it **depends solely on $s_{t-1}$ and $a_{t-1}$, and is independent of $\hat{R}_t$ and $\hat{C}_t$**. Therefore, we have  $p(s_t|\hat{R}_t,\hat{C}_t,\tau\_{t-1})=p(s_t|\hat{C}_t,\tau\_{t-1})=p(s_t|\tau\_{t-1})$. Additionally, in SMACOT, $\hat{C}_t$ is still updated via $\hat{C}_t=\hat{C}\_{t-1}-c\_{t-1}$, so $\hat{C}_t$ is also **only dependent on** $\tau\_{t-1}$.
> > > > >
> > > > > Under the previous CDT setup, $\hat{R}_t$ is updated as $\hat{R}_t=\hat{R}\_{t-1}-r\_{t-1}$, and it also only depends on $\tau\_{t-1}$. However, when both $\hat{R}_t$ and$\hat{C}_t$ are only dependent on $\tau\_{t-1}$, we argue that **this independence is the root cause of the conflict between reward and safety**.
> > > > >
> > > > > To resolve this conflict, SMACOT relaxes this independence assumption in RTG. It no longer assumes that $\hat{R}_t$ depends solely on$\tau\_{t-1}$, but also on $\hat{C}_t,s_t$. Therefore, **the only form of this formula** in SMACOT is as follows:
> > > > >
> > > > > $$
> > > > > p(\hat{a}_t,s_t,\hat{R}_t,\hat{C}_t|\tau\_{t-1})=p(\hat{a}_t|s_t,\hat{R}_t,\hat{C}_t,\tau\_{t-1})p(\hat{R}_t|s_t,\hat{C}_t,\tau\_{t-1})p(s_t|\tau\_{t-1})p(\hat{C}_t|\tau\_{t-1})
> > > > > $$
> > > > >
> > > > > Although the order has an impact in the presented formula, the **DT policy does not follow this formula**, because its goal is solely to output $\hat{a}_t$ given $s_t,\hat{C}_t,\hat{R}_t$, while the rest of the process is handled externally.
> > > > >
> > > > > We hope that our explanations of the above issues could help you better understand the contributions and methodology of our paper. We are happy to answer any further questions and sincerely thank you for your careful reviews.
> > > > >
> > > > > > [1] Chen, Lili, et al. "Decision transformer: reinforcement learning via sequence modeling." NeurIPS 2021.\
> > > > > [2] Liu, Zuxin, et al. "Datasets and benchmarks for offline safe reinforcement learning." DMLR 2024.\
> > > > > [3] Liu, Zuxin, et al. "Constrained decision transformer for offline safe reinforcement learning." ICML 2023.\
> > > > > [4] Beck, Jacob, et al. "A survey of meta-reinforcement learning." CoRR 2023.\
> > > > > [5] Reed, Scott, et al. "A Generalist Agent." TMLR 2022.\
> > > > > [6] Xu, Mengdi, et al. "Prompting decision transformer for few-shot policy generalization." ICML 2022.\
> > > > > [7] Rakelly, Kate, et al. "Efficient off-policy meta-reinforcement learning via probabilistic context variables." ICML 2019.\
> > > > > [8] Li, Lanqing, Rui Yang, and Dijun Luo. "FOCAL: Efficient Fully-Offline Meta-Reinforcement Learning via Distance Metric Learning and Behavior Regularization." ICLR 2021.\
> > > > > [9] Yuan, Haoqi, and Zongqing Lu. "Robust task representations for offline meta-reinforcement learning via contrastive learning." ICML 2022.\
> > > > > [10] Zhai, Junhai, et al. "Autoencoder and its various variants." SMC 2018.
> > > > > >

---

> > > > > > ### Author Response · Authors · 2024-11-23
> > > > > > **Dear Reviewer BUtY, are our responses address your questions?**
> > > > > >
> > > > > > Dear Reviewer BUtY:
> > > > > >
> > > > > > We thank you again for your comments and hope our responses could address your questions. As the response system will end in five days, please let us know if we missed anything. More questions on our paper are always welcomed. If there are no more questions, we will appreciate it if you can kindly raise the score.
> > > > > >
> > > > > > Sincerely yours,
> > > > > >
> > > > > > Authors of Paper6916

---

> > > > > > > ### Comment · Reviewer_BUtY · 2024-11-26
> > > > > > >
> > > > > > > We thank the authors for their dedicated efforts in responding to all the reviewers' initial reviews. I have checked the response to my first-round review and also looked into the additional experiment results the authors provided for other reviews, including the comparison with TT and some ablation variants.
> > > > > > >
> > > > > > > I list some of my feedback on the authors' responses below. My further concerns lie in the evaluation protocol and experiment results:
> > > > > > > 1. For Q1 and Q6, I acknowledge the authors for a better explanation of their core methodology, CPRTG. I have understood the motivation and implementation of CPRTG.
> > > > > > > 2. For Q2, I acknowledge the contribution of using the transformer structure for multi-task offline safe RL, yet existing works have been working on multi-task offline RL that balance different tasks with different objectives [1].
> > > > > > > 3. For Q4, we thank the authors for addressing the typos between "multi-task" and "multi-agent," which initially confused me.
> > > > > > > 4. (**Experiment questions: misaligned evaluation protocols**) In Table 1 of SMACOT, the authors mention they used four thresholds: [10, 20, 40, 80]. In the original OSRL benchmark instead (as well as the follow-up publications based on this benchmark), all the methods are evaluated under three environment-specific thresholds (see Table 5 of [2]). The misalignment between benchmark evaluation protocols could hinder the contribution of experiments.
> > > > > > > 5. (**Experiment questions: missing SOTA baselines in the Oracle setting**) In the Oracle setting, why does the author only compare the CPRTG-based SMACOT with BC-Safe, which essentially filters out the safe training dataset and does not have any return condition in the policy? To my understanding, this setting is to demonstrate the key benefits of CPRTG, which could be better verified by comparing with CDT and other stronger baselines like the one in single-task settings beyond the OSRL original baselines, such as FISOR [3].
> > > > > > > 6. (**Experiment questions: missing SOTA baselines in the Multi-task setting**): It is not a strong statement to simply the variant of CDT in Oracle and Multi-task setting, especially given the existence of [1] in using DT-based structure in multi-task offline RL.
> > > > > > > 7. (**Clarification questions on new TT results**): We thank the authors for providing additional experiment results on TT. However, I would appreciate it if the authors could the authors elaborate on the following clarification questions. (i) The original TT has a step reward token; does the adapted TT in their setting have an additional step cost token? (ii) In which of the three settings (oracle, single-task, multi-task) do the authors compare SMACOT with TT?
> > > > > > >
> > > > > > > I did not raise some of my questions regarding the experiments because I was too confused by the seemingly irrelevant "multi-agent" and "single-agent" terms in the heading paragraph of the experiments.
> > > > > > >
> > > > > > > I deeply appreciate the efforts of clarification and additional experiments during the rebuttal phase from the authors.
> > > > > > > I would like to further hear from the authors regarding my additional concerns and reconsider my final evaluation of current manuscripts.
> > > > > > >
> > > > > > >
> > > > > > > > [1] Hu, Shengchao, et al. "HarmoDT: Harmony Multi-Task Decision Transformer for Offline Reinforcement Learning." ICML 2024
> > > > > > > >
> > > > > > > > [2] Liu, Zuxin, et al. "Datasets and benchmarks for offline safe reinforcement learning." DMLR 2024
> > > > > > > >
> > > > > > > > [3] Zheng, Yinan, et al. "Safe offline reinforcement learning with feasibility-guided diffusion model." ICLR 2024
> > > > > > > >

---

> ### Comment · Reviewer_BUtY · 2024-11-26
> **Additional Questions Regarding the Theoretical Results**
>
> The current results in Appendix A basically try to make an analogy between MBPO and their current DT-based approach. However, I significantly doubt the correctness of the paper, as there are some significant differences between the two:
> - **Difference in setting**: MBPO is an online MBRL problem, SMACOT/DT-based approach is offline RL with sequence modeling formulation. *If there should be any closer work, MOPO [1] and MoReL [2] may be better choices to set up the theoretical pipeline.*
> - **Difference in assumptions**: the original theorem in MBPO holds an assumption over the policy and dynamics, and the Bellman backup gives the final performance bound, yet the learning and inference procedure of SMACOT does not fit in such a setting. Therefore, making an assumption over the reward and cost conditional distribution does not necessarily give the same results in Lemma 1 and Lemma 2.
>
> Given the current manuscripts, a more micro-scope theoretical analysis of CPRTG would be more helpful than this seemingly incorrect performance bound in Theorem 1.
>
> > [1] Yu, Tianhe, et al. "Mopo: Model-based offline policy optimization." NeurIPS 2020
> >
> > [2] Kidambi, Rahul, et al. "Morel: Model-based offline reinforcement learning." NeurIPS 2020

---

> > ### Author Response · Authors · 2024-11-29
> > **Dear Reviewer BUtY, are our responses address your questions?**
> >
> > Dear Reviewer BUtY:
> >
> > We would like to thank you once again for taking the time to evaluate our paper and for your continued support of our community. We have provided clear explanations for the additional questions and the comparison with the state-of-the-art baseline FISOR.
> >
> > We understand that you are very busy, but we would greatly appreciate it if you could take some time to check whether our responses have addressed your concerns. If there is anything we may have missed, any additional questions or comments are always welcome. If there are no further concerns, we would be grateful if you could consider reevaluating our paper.

---

> ### Author Response · Authors · 2024-11-26
> **Response to additional questions of Reviewer BUtY (1/2)**
>
> Thank you for your further detailed questions and suggestions regarding our work. We will do our best to address your concerns:
>
> ## Q1 Misaligned evaluation protocols
>
> We introduce a stricter safety threshold in order to **provide a more stringent evaluation of the safety performance** of the policy, and this does not hinder the contribution of our experiments. In OSRL, all tasks involved in our experiments are evaluated using the same three safety thresholds of **[20, 40, 80]**. This setting is somewhat lenient and may not adequately reflect the performance of policies in scenarios that require strong safety guarantees. As a result, we add an **additional threshold of 10** to establish a more rigorous safety criterion. The experiments in Appendix G.2 clearly demonstrate that CDT fails to make effective and safe decisions under this stricter safety threshold, whereas SMACOT succeeds. Since **all baselines are tested with this safety threshold configuration**, the comparison remains fair. Therefore, we believe that our modification of the safety threshold more effectively highlights the contributions of our work in the experimental section.
>
> ## Q2 Oracle setting and SOTA baselines
>
> We use the Oracle setting primarily to demonstrate **the flexibility of the CPRTG** used in SMACOT, which can **adjust its conservativeness based on different tasks during testing**.
>
> Unlike BC-Safe, SMACOT (Oracle) does not filter out the safe training dataset but instead adjusts $\beta_{\text{end}}$ according to different tasks. As shown in Table 1,  BC-Safe fails to adapt to different safety thresholds using a single policy. However, **SMACOT can flexibly adjust its conservativeness during testing by simply modifying**  $\beta_{\text{end}}$. This comparison demonstrates that **SMACOT can effectively utilize a more diverse dataset, achieving superior performance both in terms of reward and safety, while using a single policy that can adapt to different safety thresholds**.
>
> The reason we do not experiment with CDT in the Oracle setting is that **CDT already uses different initial RTGs for different tasks in the Single-task setting**, which is quite similar to our approach of adjusting $\beta_{\text{end}}$ for different tasks in the Oracle setting. CDT does not have **additional factors that can be adjusted based on the specific task**. Meanwhile, we keep the hyperparameters used in the training of the DT policy consistent between SMACOT and CDT, and therefore ensure the fairness.
>
> Thank you for your pointing out the need of comparison with a stronger baseline. FISOR is a powerful offline safe RL baseline, and we are currently conducting corresponding experiments under the Oracle setting. **We will add related discussions and update the results as soon as we obtain more data.**
>
> ## Q3 Experiments in the Multi-task setting
>
> The main goal of our experiments in the multi-task setting is to demonstrate that **SMACOT's Constraint Prioritized Prompt Encoder can effectively address the issue of cost sparsity**, which was not adequately handled by previous **multi-task methods**. Therefore, in our comparison, we primarily choose to evaluate SMACOT against the **classic prompt-based multi-task method, Prompt-DT**.
>
> HarmoDT is an effective approach for solving multi-task problems, and its masking strategy does not rely on the safe or reward settings. **However, it does not take cost sparsity into consideration, which would limit its direct application in multi-task safe RL**. (HarmoDT is essentially a multi-task method **built on Prompt-DT**, utilizing Prompt-DT for **task identification**. During training, when the true task IDs are known, it adds task masks to the model parameters for different tasks to obtain the harmony space for each task. Therefore, the issue of task misidentification caused by cost sparsity still persists in HarmoDT. Our approach actually **addresses a different aspect of the multi-task problem** compared to HarmoDT.) Meanwhile, the relatively long training time required in the multi-task setting have prevented us from providing experimental results in a timely manner.
>
> In future work, we aim to fully unleash the potential of HarmoDT by combining it with SMACOT. We will include related discussions in the future version.
>
> ## Q4 Clarification on TT
>
> In the TT experiments, we incorporate a step cost token for prediction, which treats cost similarly as reward. The comparison with TT is conducted in the Single-task setting. We will add detailed descriptions in the updated version.

---

> > ### Author Response · Authors · 2024-11-26
> > **Response to additional questions of Reviewer BUtY (2/2)**
> >
> > ## Q5 About theoretical results
> >
> > In Appendix A, we provide an alternative interpretation of CPRTG from a model-based RL perspective, rather than simply drawing an analogy between MBPO and SMACOT. While the proofs are partially inspired by MBPO, the differences between the two settings do not affect the correctness of our conclusions.
> >
> > First and foremost, the lemmas introduced from MBPO are independent of the setup (offline or online), and do not require unique assumptions over policy or dynamics. Specifically, Lemma1 is purely related   probability theory, and its proof does not involve any RL-specific contents.  On the other hand, **Lemma 2** measures the performance difference between two policies, $\pi_1$ and $\pi_2$, when rolled out in two different dynamics transitions, $p_1$ and $p_2$. This measurement **does not rely on any additional assumptions and is independent of the policy training process property, such as the contraction of the Bellman operator**. ****Despite our training method differing from traditional Bellman updates, **the performance evaluation during policy deployment remains the same**. Therefore, **using these two Lemmas here is not erroneous**.
> >
> > Secondly, our motivation for using Lemma 2 is as follows: we treat the behavior policy in the offline dataset as $\pi_2$ and the transition distribution of the offline dataset as $p_2$. The performance evaluation $\eta_2$ in Lemma 2 can thus be viewed as the expected return of the trajectories from the offline dataset. We treat our learned DT policy as $\pi_1$, and the state transition, RTG transition, and CTG transition involved during the deployment of the DT policy as $p_1$. In this way, we can evaluate the expected return of the DT policy during deployment. The transitions in terms of the state are identical between $p_1$ and $p_2$; the only differences lie in the RTG and CTG transitions.
> >
> > Finally, from the perspective of model rollout, MBPO essentially treats $p_1$ in Lemma 2 as the real environment and $p_2$ as the model, **without introducing any additional model-based priors or assumptions**. In this paper, we are simply **offering an alternative interpretation** of $p_1$ and $p_2$ in Lemma 2. Specifically, $p_1$ represents the real state transitions during deployment, combined with the RTG and CTG transitions that we define, while $p_2$ represents the state transitions, RTG transitions, and CTG transitions in the offline dataset. Therefore, although we use Lemmas from MBPO, these are **independent of whether the setup is model-based or whether it involves online training**.
> >
> > In conclusion, we provide this additional interpretation of CPRTG to further support our main contribution, addressing the core challenge in multi-task offline safe RL—conflicts between CTG and RTG objectives.

---

> ### Author Response · Authors · 2024-11-27
> **New baseline FISOR**
>
> ## Comparison with SOTA baseline FISOR
>
> Thanks a lot for your constructive suggestion, we have added additional comparison results between SMACOT and a SOTA baseline FISOR in both the **Oracle and Single-Task** settings, as shown in the table. (When the cost is greater than 1, smaller cost values are preferable, while for cases where cost ≤ 1, larger reward values are prioritized.)
>
> | Task | Oracle |  | Single-Task |  |
> | :---: | :---: | :---: | :---: | :---: |
> |  | SMACOT | FISOR | SMACOT | FISOR |
> |  | r↑  c↓ | r↑  c↓ | r↑  c↓ | r↑  c↓ |
> | PointButton1 | **0.09  0.91** | **-0.01  0.28** | **0.06  0.66** | 0.08  1.30 |
> | PointButton2 | **0.08  0.92** | **0.05  0.43** | 0.14  1.41 | 0.11  1.41 |
> | PointCircle1 | **0.54  0.62** | **0.05  0.06** | **0.50  0.63** | 0.44  5.54 |
> | PointCircle2 | **0.61  0.98** | **0.20  0.00** | **0.61  0.98** | 0.71  6.21 |
> | PointGoal1 | **0.51  0.87** | **0.03  0.01** | **0.36  0.56** | 0.66  2.14 |
> | PointGoal2 | **0.29  0.91** | **0.05 0.08** | 0.31  1.02 | 0.29  1.28 |
> | PointPush1 | **0.19  0.88** | **0.31  0.89** | **0.19  0.88** | **0.31  0.89** |
> | PointPush2 | **0.13  0.63** | **0.09  0.29** | 0.19  1.47 | 0.24  1.40 |
> | Average | **0.31  0.84** | **0.10  0.26** | **0.30  0.95** | 0.36  2.52 |
>
> As is shown in the table, it is evident that **SMACOT outperforms FISOR in both the Oracle and Single-Task settings**. First, in the Single-Task setting, SMACOT still shows significantly **better safety performance**, meeting the safety constraints in 4 additional tasks compared to FISOR, which is trained using the default hyper-parameters provided by the authors. This clearly demonstrates the effectiveness of SMACOT’s CPRTG in addressing the reward-safety conflict. In the Oracle setting, we adjust FISOR’s **reverse expectile parameter** $\tau$ for each task, setting $\tau = [0.8, 0.8, 0.7, 0.7, 0.8, 0.8, 0.9, 0.8]$ for the 8 test tasks specifically. According to the ablation results of FISOR, $\tau$ is positively correlated with the conservativeness of the policy. From the results, we see that **both SMACOT and FISOR are able to satisfy safety constraints in the Oracle setting, but SMACOT achieves better reward performance**. This highlights the flexibility of SMACOT’s CPRTG in adjusting the conservativeness of the policy. Another clear advantage of SMACOT in the Oracle setting is that the hyperparameter $\beta_{\text{end}}$ is a **test-phase-only** parameter. This means that adjusting this parameter does not require retraining the policy, making it extremely convenient for fine-tuning. In contrast, FISOR’s hyperparameter $\tau$ is a **training-phase** parameter, and adjusting it requires retraining the policy, which is time-consuming. Overall, these results clearly demonstrate the effectiveness of CPRTG in handling the reward-safety trade-off, a core challenge in safe reinforcement learning. The relevant results and discussions have been added in **Appendix G.6**.
>
> Once again, thank you for your insightful questions, which have greatly helped improve our paper. We truly appreciate your valuable feedback! We hope that the inclusion of the new baseline helps you gain a deeper understanding of the effectiveness of SMACOT's CPRTG. We are happy to answer any further questions.

---

> ### Comment · Reviewer_BUtY · 2024-12-01
>
> Dear authors,
>
> Thanks for the detailed  reply, I list my feedback regarding some remaining concerns:
>
> 1. **Missing comparison with CDT in Oracle setting**: the authors stated in their response that:
> > The reason we do not experiment with CDT in the Oracle setting is that CDT already uses different initial RTGs for different tasks in the Single-task setting, which is quite similar to our approach of adjusting $\beta_{end}$ for different tasks in the Oracle setting. CDT does not have additional factors that can be adjusted based on the specific task. Meanwhile, we keep the hyperparameters used in the training of the DT policy consistent between SMACOT and CDT, and therefore ensure the fairness.
>
> However, to my understanding, **this similarity between CDT and SMACOT** in the single-task setting **should NOT preclude a fair comparison between them in the oracle setting**. For example, in CDT with the oracle setting, we can adjust an optimal initial RTGs for CDT based on the privileged information from offline dataset. Please correct me if I missed anything in this part.
>
> 2. **Comparison with TT and FISOR**: I appreciate the authors providing detailed clarifications and responses to new experiments, which could potentially improve the contribution of the paper. Specifically, a detailed description of FISOR implementation in their setting is helpful. However, there are two questions regarding the **comparison with FISOR**:
>
> - 2.1 **Clarification of the difference between single-task and oracle**: this part is still confusing to me a little bit, especially seeing such a big performance drop of FISOR between the two settings. There is **no clear definition of what an oracle and single-task setting are in the main text, and how different their training and evaluation settings are. Specifically, on page 7, line 365, the authors categorize **BC-Safe** as the single-task baseline instead of an oracle baseline.
>
> - 2.2 **Limited coverage of environments**: the FISOR works cover different environments among Safety-gymnasium, Bullet-safety-gym and MetaDrive in their experiments, yet the SMACOT only compares with it in the `Point-XX` environments of Safety-gymnasium. This is not a very convincing result since the SMACOT also compares with the baselines in `Car-XX` and the `mujoco-based` environments (swimmer, hopper, etc.). It is worth mentioning that **`Car-XX` and `mujoco-based` environments are reported in the original FISOR paper, while the `Point-XX` experiments are not**. A more comprehensive experiment comparison that includes `Car-xx` and `mujoco-based` environments could help in addressing the experiment concern.
>
> 3. **Non-convincing theoretical results**: we thank the authors for providing detailed clarification and response in this part. Unfortunately, the current theoretical results are problematic and would offset the understanding of the authors' empirical contributions.
> - 3.1 **Difference in the learning paradigm**: MBPO needs the difference in transition dynamics (which the authors believe are analogical to the difference in RTG and CTG in their setting) mainly because they use the imagination rollout samples to further train the model-free RL policies. However, no synthetic data is used in SMACOT.
> - 3.2 **Difference in neural network parameterization**: given the current definition of state and reward in MBPO and the authors' manuscripts, MBPO takes in a single-step state and action in the transition dynamics (as well as the actor-critic model-free RL parts), while for SMACOT, it takes in the history trajectories and the action sequence in online inference is not GT at all. However, the MBPO bound fails to capture this compounding error in multi-step modeling in value (RTG, CTG) prediction or policy learning.
> - 3.3 **Blurred key contributions in theoretical results**: as reviewer eMX9 stated in their additional response, the assumption of the identical training and evaluation environments is very confusing. The authors stated that: `offline dataset and the deployment environment are consistent, the state transitions themselves are identical`. and this is not the case in multi-task settings and few-shot transfer experiments where dynamics could differ.
>
> In general, instead of mimicking the unsuitable theoretical guarantees from MBPO, I would encourage the authors to further consider how to enhance the theoretical understanding of their **unique empirical contribution** in SMACOT, e.g. (i) a more generalizable/adaptive pre-training and (ii) the mechanism in constraint prioritization for offline safe RL through a lens of DT-based sequence modeling.
>
> Although I appreciate the authors' efforts during the rebuttal and discussion phase, given the considerable number of remaining concerns, I decided to keep my current score.

---

> > ### Author Response · Authors · 2024-12-02
> > **Response to additional questions of Reviewer BUtY (1/2)**
> >
> > We thank Reviewer BUtY's further discussion and hope the following responses can clarify potential misunderstandings.
> >
> > ## Q1 Comparison with CDT in the Oracle setting
> >
> > We apologize for not clarifying this in the earlier response. The technique you propose, which adjusts the optimal initial RTGs based on privileged information from the offline dataset, is indeed insightful. **In fact, CDT itself implements this approach [1], though it was not explicitly mentioned in the original paper**. Therefore, this is **also implemented in the Single-Task versio of CDT in our paper**. Despite employing this "oracle-like" technique, CDT still fails to achieve safer and higher-performing actions compared to SMACOT in the Single-Task setting. We will provide a more detailed description of CDT and its implementation in the revised version.
> >
> > ## Q2 Difference between Single-Task and Oracle in comparison with TT and FISOR
> >
> > Thank you for pointing out the lack of corresponding clarifications regarding the experimental settings.
> >
> > First, the Oracle setting is a special case of the Single-Task setting. In the Single-Task setting, the **algorithm uses a unified set of hyperparameters across all tasks**, whereas in the Oracle setting, the algorithm **can adjust components based on the specific task** at hand. The FISOR agent in the Single-Task setting is trained using the hyperparameters reported in [2]. In the Oracle setting, we adjust hyperparameter $\tau$ for each task to achieve better performance.
> >
> > Secondly, Oracle, Single-Task, and Multi-Task in Table 1 refer to three different experimental settings for training and evaluation. Specifically, agents are trained using a dataset from a specific task and evaluated under the corresponding task in both Oracle and Single-Task settings. However, in the Oracle setting, we adjust important components of the algorithm for each task. In the Multi-Task setting, agents are trained and evaluated across multiple tasks.
> >
> > We classified BC-Safe as a Single-Task baseline on page 7, line 365, because “Single-Task” here refers to algorithms that are trained and evaluated under a specific task. SMACOT (Oracle) is also classified as a “single-task version” on the same page, line 359.
> >
> > We apologize for any misunderstandings caused by the expressions in the main text, and we will clarify this in future versions.
> >
> > ## Q3 Limited coverage of environments in experiments
> >
> > We agree that a more comprehensive experimental comparison will better highlight our empirical contributions. As such, **we are conducting additional experiments with FISOR in the Safety-Gymnasium environments**. We will provide the updated results as soon as they become available.

---

> > ### Author Response · Authors · 2024-12-02
> > **Response to additional questions of Reviewer BUtY (2/2)**
> >
> > ## Q4 Difference in the learning paradigm
> >
> > The difference in learning paradigms does not affect the validity of our theoretical results.
> >
> > On one hand, the theoretical results of MBPO try to “address” the issue  brought by difference in transition dynamics, which is not a need but is inherent to model-based RL. In our analysis, the **distribution of RTG (CTG) will differ in offline dataset and during deployment**.
> >
> > On the other hand, we treat the **RTG and CTG from the offline dataset as real data**, while the **manually set RTG and CTG during deployment are considered synthetic data**, since they do not align with the distribution of the offline dataset.
> >
> > ## Q5 Difference in neural network parameterization
> >
> > The neural network parameterizations are different in MBPO and SMACOT, but our theoretical results remain valid regardless of whether an additional preceding sequence $\tau$ is included.
> >
> > The aim of our analysis is to emphasize the benefit of CPRTG, as the CPRTG's optimization of $\epsilon_R$ can lead to a better performance bound. When trajectory $\tau$ is provided as an input, it can **similarly be treated as part of the state**. In this case, $\epsilon_R$ is determined by $p_1(\hat{R}\_{t+1}, \hat{C}\_{t+1} | s', s, \hat{R}_t, \hat{C}_t, a, \tau_t)$ and $p_2(\hat{R}\_{t+1}, \hat{C}\_{t+1} | s', s, \hat{R}_t, \hat{C}_t, a, \tau_t)$. This does not introduce additional compounding errors from $\tau$, because $\tau$'s input is the same for both $p_1$ and $p_2$.
> >
> > For the overall performance bound, the introduction of $\tau$ does indeed add **an extra factor determined by the TVD of $\tau$'s transition between $p_1$ and $p_2$**. However, this factor exists in both traditional DT frameworks and in SMACOT's CPRTG-based DT framework. Additionally, since the transition of $\tau$ is directly determined by $s, a, \hat{R}_t, \hat{C}_t$, and the transition of $s, a, \hat{C}_t$ remains consistent with the traditional DT framework, SMACOT actually **reduces the TVD in the transition of $\tau$ by reducing the TVD in $\hat{R}_t$'s transition**.
> >
> > Therefore, the conclusion that CPRTG optimizes $\epsilon_R$ to achieve a better performance bound remains unchanged, regardless of whether $\tau$ is considered as an input.
> >
> > ## Q6 Blurred key contributions in theoretical results
> >
> > We apologize for not providing a clearer explanation of the setting in the theoretical results. In our analysis, CPRTG is discussed within the context of a specific task, meaning the state transitions are assumed to be the same. This conclusion **remains valid in multi-task scenarios, provided the agent can accurately identify the task at hand**. Our Constrained Prioritized Prompt Encoder is specifically designed to enhance task identification accuracy. In few-shot transfer scenarios, the **additional few-shot data for new tasks is treated as training data, ensuring that the training and testing environments remain consistent**.
> >
> > We will include a more detailed explanation in the Appendix to clarify these points.
> >
> > > [1] Zuxin Liu. (2024). Elegant implementations of offline safe RL algorithms in PyTorch [[https://github.com/liuzuxin/osrl](https://github.com/liuzuxin/osrl)]\
> > [2] Zheng, Yinan, et al. "Safe offline reinforcement learning with feasibility-guided diffusion model." ICLR 2024
> > >

---

> > ### Author Response · Authors · 2024-12-02
> > **Dear Reviewer BUtY, are our responses address your further questions?**
> >
> > Dear Reviewer BUtY:
> >
> > We would like to express our sincere gratitude for taking the time to evaluate our paper and for your continued support of our community.
> >
> > In response to your concerns, we have provided additional explanations and conducted further experiments  of baseline FISOR. As the response system will close in **two days** (**one day** for reviewers to respond), please let us know if we have overlooked anything. We welcome any further questions or feedback on our paper.
> >
> > Sincerely yours,
> >
> > Authors of Paper6916

---

> ### Author Response · Authors · 2024-12-02
> **More results of FISOR**
>
> Dear Reviewer BUtY,
>
>  Thank you for your valuable suggestions. We have now added FISOR experimental results for the Car tasks. Indeed, FISOR performs noticeably better in the Car tasks compared to its performance in the Point tasks, but the **overall conclusions remain unchanged**. In the Single-Task setting, in the Car tasks, both SMACOT and FISOR satisfy safety constraints in the same number of tasks, with similar overall reward performance. However, FISOR more severely violates the safety constraints in the Circle tasks. In the Oracle setting, FISOR does manage to meet safety requirements in the CarCircle1 task, which is its advantage over SMACOT. Nevertheless, safety performance remains similar in other tasks, and SMACOT outperforms FISOR in terms of reward performance. Therefore, overall, SMACOT performs similarly to FISOR in the Car tasks, with a distinct advantage in the Point tasks, highlighting the effectiveness of SMACOT's CPRTG.
>
> | Task | Oracle |  | Single-Task |  |
> | :---: | :---: | :---: | :---: | :---: |
> |  | FISOR | SMACOT | FISOR | SMACOT |
> |  | r↑  c↓ | r↑  c↓ | r↑  c↓ | r↑  c↓ |
> | PointButton1 | **-0.01  0.28** | **0.09  0.91** | 0.08  1.30 | **0.06  0.66** |
> | PointButton2 | **0.05  0.43** | **0.08  0.92** | 0.11  1.41 | 0.14  1.41 |
> | PointCircle1 | **0.05  0.06** | **0.54  0.62** | 0.44  5.54 | **0.50  0.63** |
> | PointCircle2 | **0.20  0.00** | **0.61  0.98** | 0.71  6.21 | **0.61  0.98** |
> | PointGoal1 | **0.03  0.01** | **0.51  0.87** | 0.66  2.14 | **0.36  0.56** |
> | PointGoal2 | **0.05  0.08** | **0.29  0.91** | 0.29  1.28 | 0.31  1.02 |
> | PointPush1 | **0.31  0.89** | **0.19  0.88** | **0.31  0.89** | **0.19  0.88** |
> | PointPush2 | **0.09  0.29** | **0.13  0.63** | 0.24  1.40 | 0.19  1.47 |
> | CarButton1 | **-0.02  0.78** | **0.07  0.74** | **-0.06  0.16** | **0.07  0.74** |
> | CarButton2 | **-0.02  0.40** | **-0.02  0.89** | **-0.02  0.40** | -0.02  1.33 |
> | CarCircle1 | **0.21  0.24** | 0.49  2.96 | 0.69  5.35 | 0.51  3.34 |
> | CarCircle2 | **0.40  0.42** | **0.28  0.98** | 0.51  4.13 | **0.28  0.98** |
> | CarGoal1 | **0.43  0.72** | **0.39  0.75** | **0.43  0.72** | **0.33  0.47** |
> | CarGoal2 | **0.07  0.27** | **0.19  0.81** | **0.07  0.27** | **0.19  0.81** |
> | CarPush1 | **0.25  0.43** | **0.28  0.96** | **0.25  0.43** | **0.20  0.67** |
> | CarPush2 | **0.13  0.59** | **0.09  0.88** | **0.13  0.59** | **0.07  0,73** |
> | Average | 0.14  0.36 | 0.26  0.98 | 0.30  2.01 | 0.25  1.04 |

---

> > ### Author Response · Authors · 2024-12-02
> > **Overall comparison with FISOR**
> >
> > We have completed the comparison between SMACOT and FISOR across all environments. Overall, both SMACOT and FISOR have their respective strengths in addressing the offline safe RL problem.
> >
> > **SMACOT's Strengths:**
> >
> > - In the Single-Task setting, SMACOT has a safety advantage in Point tasks.
> > - In the Oracle setting, it achieves higher overall reward performance in safety-satisfied tasks.
> > - In the Oracle setting, hyperparameter adjustments of SMACOT do not require any additional neural network training.
> > - The same policy of SMACOT can adapt to various safety thresholds.
> >
> > **FISOR's Strengths:**
> >
> > - In the Single-Task setting, FISOR demonstrates a safety advantage in Mujoco tasks.
> > - In the Oracle setting, FISOR satisfies safety constraints in three additional environments.
> >
> > We believe that the different strengths of SMACOT and FISOR arise primarily from the way they model the offline safe RL problem. **SMACOT models the problem as a soft constraint problem**, emphasizing the trade-off between reward and safety. On the other hand, **FISOR treats the problem as a hard constraint problem**, focusing more on the absolute satisfaction of safety requirements. Therefore, depending on the safety requirements of the application, one can choose between these two algorithms.
> >
> > Overall, the design of SMACOT’s CPRTG enables the policy to **handle the reward-safety trade-off better than traditional RTGs**, achieving performance similar to FISOR, a SOTA method in hard constraint modeling. Additionally, it offers **efficient adjustment of conservatism in the testing phase and adaptability to multiple safety thresholds**. Hence, we believe SMACOT makes a valuable contribution to offline safe reinforcement learning in the single-task setting as well.
> >
> > | Task | Oracle |  | Single-Task |  |
> > | :---: | :---: | :---: | :---: | :---: |
> > |  | FISOR | SMACOT | FISOR | SMACOT |
> > |  | r↑  c↓ | r↑  c↓ | r↑  c↓ | r↑  c↓ |
> > | PointButton1 | **-0.01  0.28** | **0.09  0.91** | 0.08  1.30 | **0.06  0.66** |
> > | PointButton2 | **0.05  0.43** | **0.08  0.92** | 0.11  1.41 | 0.14  1.41 |
> > | PointCircle1 | **0.05  0.06** | **0.54  0.62** | 0.44  5.54 | **0.50  0.63** |
> > | PointCircle2 | **0.20  0.00** | **0.61  0.98** | 0.71  6.21 | **0.61  0.98** |
> > | PointGoal1 | **0.03  0.01** | **0.51  0.87** | 0.66  2.14 | **0.36  0.56** |
> > | PointGoal2 | **0.05  0.08** | **0.29  0.91** | 0.29  1.28 | 0.31  1.02 |
> > | PointPush1 | **0.31  0.89** | **0.19  0.88** | **0.31  0.89** | **0.19  0.88** |
> > | PointPush2 | **0.09  0.29** | **0.13  0.63** | 0.24  1.40 | 0.19  1.47 |
> > | CarButton1 | **-0.02  0.78** | **0.07  0.74** | **-0.06  0.16** | **0.07  0.74** |
> > | CarButton2 | **-0.02  0.40** | **-0.02  0.89** | **-0.02  0.40** | -0.02  1.33 |
> > | CarCircle1 | **0.21  0.24** | 0.49  2.96 | 0.69  5.35 | 0.51  3.34 |
> > | CarCircle2 | **0.40  0.42** | **0.28  0.98** | 0.51  4.13 | **0.28  0.98** |
> > | CarGoal1 | **0.43  0.72** | **0.39  0.75** | **0.43  0.72** | **0.33  0.47** |
> > | CarGoal2 | **0.07  0.27** | **0.19  0.81** | **0.07  0.27** | **0.19  0.81** |
> > | CarPush1 | **0.25  0.43** | **0.28  0.96** | **0.25  0.43** | **0.20  0.67** |
> > | CarPush2 | **0.13  0.59** | **0.09  0.88** | **0.13  0.59** | **0.07  0,73** |
> > | SwimmerVelocityV0 | **-0.04  0.31** | **0.62  0.98** | **-0.04  0.31** | 0.63  1.29 |
> > | SwimmerVelocityV1 | **-0.04  0.14** | **0.44  0.87** | **-0.04  0.14** | **0.44  0.87** |
> > | HopperVelocityV0 | **0.30  0.23** | **0.18  0.52** | **0.30  0.23** | 0.84  1.50 |
> > | HopperVelocityV1 | **0.16  0.86** | **0.18  0.86** | **0.16  0.86** | 0.35  1.17 |
> > | HalfCheetahVelocityV0 | **0.89  0.00** | **0.67  0.38** | **0.89  0.00** | **0.51  0.36** |
> > | HalfCheetahVelocityV1 | **0.89  0.00** | **0.84  1.00** | **0.89  0.00** | **0.84  1.00** |
> > | Walker2dVelocityV0 | **0.05  0.12** | 0.32  2.90 | 0.11  1.11 | 0.32  2.90 |
> > | Walker2dVelocityV1 | **0.53  0.80** | **0.78  0.12** | **0.53  0.80** | **0.73  0.42** |
> > | AntVelocityV0 | **0.77  0.00** | **0.90  0.84** | **0.77  0.00** | **0.90  0.84** |
> > | AntVelocityV1 | **0.89  0.00** | 0.97  1.58 | **0.89  0.00** | 0.98  1.75 |
> > | Average | 0.25  0.32 | 0.39  0.99 | 0.36  1.37 | 0.40  1.11 |

---

### Author Response · Authors · 2024-11-20
**General response to reviewers**

We appreciate valuable comments from all reviewers. We have revised our paper carefully according to your suggestions We summarize our modifications as follows.

1. We revise the definition of the task in **Section 3.1 (for Reviewer BUtY)**.
2. We add an explanation regarding the non-fully autoregressive nature of Decision Transformer (DT) in **Section 3.2 (for Reviewer BUtY)**.
3. We correct typos related to the experimental setup in **Section 5 (for Reviewer BUtY)**.
4. We add related theoretical analysis of the policy from the perspective of offline reinforcement learning in **Appendix A (for Reviewer eMX9)**.
5. We include a more detailed explanation of two safety-conditioned RL works in **Appendix B (for Reviewer BUtY)**.
6. We add a more detailed description of SMACOT’s (our method) task identification process in **Appendix C (for Reviewer BUtY)**.
7. We add the description of the SMACOT method for distinguishing tasks in an unknown environment in **Appendix D (for Reviewer GfAj)**.
8. We add more experiments in **Appendix G**:
    1. **G.1** Time complexity analysis **(for Reviewer GfAj)**.
    2. **G.2** Zero-shot generalization to different safety constraints and different safety thresholds **(for Reviewer eMX9)**.
    3. **G.5** Comparison with Trajectory Transformer (TT) **(for Reviewer xwJ3)**.
    4. **G.6** Comparison with FISOR **(for Reviewer BUtY)**.
    4. **G.7** Ablation on the hyperparameter $X$ **(for Reviewer xwJ3)**.
    5. **G.8** Discussion and ablation on the inverse dynamics model $g$ **(for Reviewer xwJ3)**.
    6. **G.9** Comparison with another multi-task baseline in task transfer and task transfer results to a dissimilar task **(for Reviewer BUtY and eMX9)**.

The major modifications are colored red for the sake of clarity in the recently submitted version. We hope that our response can address all your concerns of our paper. Please let us know if we miss anything. We are looking forward to further inspiring discussions.

---

### Meta-Review · Area_Chair_6AQJ · 2024-12-22

**Metareview:**

Summary: The paper introduces SMACOT, a method for safe offline multi-task reinforcement learning that uses a Constraint Prioritized Return-To-Go token and a specialized prompt encoder to balance safety constraints with reward maximization. The approach shows improved safety performance on the OSRL dataset, meeting safety constraints in more environments compared to baselines.

Strengths:

SMACOT demonstrates promising results in balancing safety and reward in multi-task reinforcement learning, with comprehensive experiments on the OSRL dataset.

The paper is well-structured, with clear motivation and a thorough explanation of the proposed method.


Drawbacks:

The paper's experiments and theoretical contributions are limited, with concerns about the generalizability of the approach to more diverse and complex task settings.

There is a lack of comparison with state-of-the-art methods like the Trajectory Transformer (TT), which could provide a more robust evaluation of SMACOT's performance.

The cost constraint is treated as a soft constraint, which may not fully guarantee meeting safety requirements, and there are concerns about the alignment of the Return-To-Go set with actual returns in safety-sensitive environments.

Given the above points, I must reject this work as it does not fully meet the acceptance criteria due to its limited theoretical foundation, lack of comparison with existing state-of-the-art methods, and concerns about the robustness of the safety constraints.

**Additional Comments On Reviewer Discussion:**

Concerns are not well-addressed.

---

### Decision · Program_Chairs · 2025-01-22

Reject